# Global Convergence in Training Large-Scale Transformers

**Cheng Gao**[1]* **Yuan Cao**[2]* **Zihao Li**[1] **Yihan He**[1] **Mengdi Wang**[1]
**Han Liu**[3] **Jason M. Klusowski**[1]† **Jianqing Fan**[1]†

[1]Princeton University   [2]The University of Hong Kong   [3]Northwestern University
{chenggao,zihaoli,yihan.he,mengdiw,jason.klusowski,jqfan}@princeton.edu
yuancao@hku.hk    hanliu@northwestern.edu

## Abstract

Despite the widespread success of Transformers across various domains, their optimization guarantees in large-scale model settings are not well-understood. This paper rigorously analyzes the convergence properties of gradient flow in training Transformers with weight decay regularization. First, we construct the mean-field limit of large-scale Transformers, showing that as the model width and depth go to infinity, gradient flow converges to the Wasserstein gradient flow, which is represented by a partial differential equation. Then, we demonstrate that the gradient flow reaches a global minimum consistent with the PDE solution when the weight decay regularization parameter is sufficiently small. Our analysis is based on a series of novel mean-field techniques that adapt to Transformers. Compared with existing tools for deep networks [47] that demand homogeneity and global Lipschitz smoothness, we utilize a refined analysis assuming only *partial homogeneity* and *local Lipschitz smoothness*. These new techniques are of independent interest.

## 1 Introduction

Transformers have revolutionized the field of deep learning since their introduction in [66]. These models are distinguished by their immense scales, often comprising billions of parameters to achieve state-of-the-art performance. Notably, this massive parameterization enables them to excel in a variety of domains, notably in natural language processing [21, 55, 65] and vision tasks [20, 36], where they have significantly advanced the frontiers of machine learning.

Despite the widespread adoption of Transformer models, our understanding of their optimization guarantees is still in its early stages. One particularly intriguing phenomenon is that as the size of model increases, training algorithms typically converges globally despite the highly nonconvex landscape of the training objective function. Remarkably, it remains somewhat enigmatic how gradient-based approaches can consistently succeed when training large-scale Transformers.

Notably, there have been several recent works showing the global convergence of training overparameterized neural networks [51, 16, 28, 14, 47, 22, 23, 35, 3, 25, 76]. In particular, several works [47, 22, 23] studied the setting with deep neural networks with skip connections. By studying the connections between the network with discretization in the parameter space and a corresponding ordinary differential equation system [71, 12, 43], these works demonstrated global convergence guarantees of wide and deep neural networks based on a mean-field analysis. However, these results are established based on certain homogeneity and/or global Lipschitz smoothness properties of the

---

*Equal Contribution.

38th Conference on Neural Information Processing Systems (NeurIPS 2024).

neural network, which are not applicable to Transformer models. Therefore, it remains an open question how gradient-based methods can effectively train large-scale Transformers.

## 1.1 Our contribution

In this work, we bridge the gap between Transformer theory and practice by demonstrating the global convergence of Transformer training optimization via gradient flow in a large-scale model regime. We analyze the mean-field limit of the Transformer model, which is characterized by the *distribution* of model parameters, shifting the focus from parameter space to distributional dynamics in the Wasserstein metric [16]. This approach yields two key theorems:

    i. We show the closeness between practical discrete Transformers trained by gradient flow and continuous Transformers whose parameter distribution follows a partial differential equation of the Wasserstein gradient flow (Theorem 3.1). Our result demonstrates that large-scale discrete Transformers can be approximated by its mean-field limit and the approximation error can be expressed in terms of the width and depth of the Transformer models.

    ii. This approximation facilitates our analysis of the global convergence (Theorem 4.1) of discrete Transformer models. By leveraging the universal approximation capabilities of either the self-attention or feed-forward layers, we demonstrate that a basic gradient flow method can reliably find a global optimum, despite the highly non-convex landscape of the training objective.

We also highlight our novel contributions to Transformer theory through the development of these two core results:

    i. The assumption on activation regularity conditions (Assumption 2) is less stringent compared to those usually found in studies of two-layer neural networks [51, 16, 28, 14] or deep ResNet networks [22, 23, 47]. In particular, many existing approximation guarantees reply on a Lipschitz continuity property of the network gradients, which limits the mean-filed study to neural networks with smooth activation functions. In comparison, our analysis relaxes this assumption and only requires local Lipschitz continuity of the gradient in expectation. This relaxation broadens the applicability of our approach and ensures that our result can cover more practical Transformer architectures.

    ii. Our model differs from the ResNet models in [47, 22, 23, 13], as those models incorporate only a single identical encoder within each evolutionary block. Unlike the typical theoretical configurations, our model employs two distinct encoders $f$ and $h$ that alternate throughout the network's depth. More importantly, despite the distinct encoders used, the continuous limit of our model uniformly interprets the encoder as an average of $f$ and $h$, providing a rigorous validation of concepts proposed in [47] and [67] from a new perspective.

    iii. Our global convergence guarantee for training Transformer models is also broadly applicable: our assumption (Assumption 4) ensures global convergence by relying on the universal approximation capabilities of *either* the self-attention or the feed-forward encoder. Additionally, we incorporate a more flexible framework by adopting partial 1-homogeneity for only a subset of the parameters, in contrast to the full parameter homogeneity required in studies such as [47]. This modification enables the use of softmax and sigmoid activation, expanding beyond the hardmax and ReLU restricted by full homogeneity.

**Additional related works.** See Appendix B for a detailed discussion.

**Notations.** For any $\alpha \in \mathbb{R}^d$, $\dim(\alpha)$ refers to its dimension $d$. For any $B \in \mathbb{R}^{d \times d}$, its trace is denoted by $\mathrm{Tr}(B)$. For any positive integer $n$, Let $[n] = \{1, 2, \ldots, n\}$. Let $0_d$ denote the $d$-dimension vector of all zeros. Let $W_p(\mu, \nu)$ denote the Wasserstein-$p$ distance between two probability measures $\mu, \nu \in \mathcal{P}(\mathbb{R}^d)$ for $p \geq 1$. For a matrix $A = (a_1, a_2, \ldots, a_n)$, define its vectorization version as $\mathrm{vec}[A] := (a_1^\top, a_2^\top, \ldots, a_n^\top)^\top$. Let $\delta(\cdot)$ denote the Dirac mass and $\mathbf{1}\{\cdot\}$ be the indicator function. Let $\mathrm{supp}(\cdot)$ denote the support of any distribution. Let $\|\cdot\| = \|\cdot\|_2$ denote the $l_2$ norm and $\|\cdot\|_{\max}$ denote the maximum norm. For any subsets $D_1, D_2$ in Euclidean space, define $\mathcal{C}(D_1, D_2)$ as the collection of functions that map $D_1$ to $D_2$ and are continuous over $D_1$. Define the Bounded Lipschitz norm for any measure $\mu \in \mathcal{M}(\mathbb{R}^d)$ as $\|\mu\|_{\mathrm{BL}} := \sup\{\int f d\mu : f : \mathbb{R}^d \to \mathbb{R}, \ \sup|f| \leq 1, \ f \text{ is } 1-\mathrm{Lipschitz}\}$.

## 2 Transformer model

In this section, we describe our deep Transformer model with each data input as a sequence, and the gradient flow algorithm used for training.

### 2.1 Data setting

In our paper, the data input is both general and straightforward: an input sequence $H \in \mathbb{R}^{D \times (N+1)}$ consisting of $N + 1$ tokens, each with dimension $D$. We consider the setting where each input sequence $H$ is associated with a label $y(H) \in \mathbb{R}$, where $y(H)$ is the target function we aim to learn. Furthermore, we assume that each instance $H$ is i.i.d. drawn from a population distribution $\mu$.

**Relation to in-context learning (ICL)**   Our data setting is versatile and applicable to any task involving sequential input. It particularly suits the in-context learning (ICL) scenario [6, 10, 75], where models are capable of making accurate predictions on new data when prompted with training examples from the same pool. For clarity, consider the input sequence $H \in \mathbb{R}^{D \times (N+1)}$ formatted as follows:

$$H = [h_1, h_2, \ldots, h_{N+1}] = \begin{bmatrix} x_1 & x_2 & \ldots & x_N & x_{N+1} \\ y_1 & y_2 & \ldots & y_N & 0 \\ p_1 & p_2 & \ldots & p_N & p_{N+1} \end{bmatrix} \overset{i.i.d.}{\sim} \mu, \quad y_{N+1} = y(H).$$

Here, $\{x_i\}_{i \in [N]}$ are the input vectors, each associated with a corresponding label $\{y_i\}_{i \in [N]}$. The last token, $x_{N+1}$ is the test input for which a prediction is made. The third row contains the customized and fixed positional encoding vectors $\{p_i\}_{i \in [N]}$, which typically include ones, zeros, and indicators denoting the token for prediction. The label for the query point $x_{N+1}$ is then given by $y_{N+1} = y(H)$ in our terminology. ICL operates in a zero-shot fashion, without any updates to the model's parameters, highlighting a unique and powerful capability of these systems to adapt and generalize based on the provided context alone. In [6], the authors demonstrate that fixed Transformers can approximate in-context penalized generalized linear regression to any desired degree.

### 2.2 Model

We follow a common configuration of Transformer architectures [6, 38, 40, 48, 73] where each Transformer block consists of two distinct layers: a self-attention mechanism layer and a token-wise feed-forward neural network layer, both equipped with skip connections. We assume that both layers consist of the average of $M$ heads, treated uniformly as the *width* across all blocks for simplicity. The formulation for a matrix input $Z \in \mathbb{R}^{D \times (N+1)}$ and a given residual step size $\eta > 0$ is as follows: Each residual self-attention layer is represented by

$$\text{Attn}_{\theta_1, \theta_2, \ldots, \theta_M}(Z, \eta) = Z + \eta M^{-1} \sum_{j=1}^{M} f(Z, \theta_j), \tag{2.1}$$

and each residual feed-forward neural network layer is defined by

$$\text{MLP}_{w_1, w_2, \ldots, w_M}(Z, \eta) = Z + \eta M^{-1} \sum_{j=1}^{M} h(Z, w_j) \tag{2.2}$$

for parameter vectors $\theta$ and $w$ in the Euclidean space. The encoders for the self-attention and feed-forward layers are denoted as $f : \mathbb{R}^{D \times (N+1)} \to \mathbb{R}^{D \times (N+1)}$ and $h : \mathbb{R}^{D \times (N+1)} \to \mathbb{R}^{D \times (N+1)}$, respectively. The self-attention encoder $f$ formulation, commonly adopting a *multiplicative* or *dot-product* approach as detailed in [8, 38, 48, 64, 66, 73], can be exemplified by

$$f(Z, \theta) = W_O W_V Z \sigma_{\text{A}} \Big[ (W_K Z)^\top W_Q Z \Big],$$

where $W_V, W_K, W_Q \in \mathbb{R}^{s \times D}$, and $W_O \in \mathbb{R}^{D \times s}$. This formulation can be reparametrized to

$$f(Z, \theta) = V Z \sigma_{\text{A}} \Big[ Z^\top W Z \Big], \tag{2.3}$$

where $V, W \in \mathbb{R}^{D \times D}$, $\theta = \text{vec}[V, W]$. The activation $\sigma_A$ typically uses column-wise softmax, but component-wise ReLU is also viable, as in [6]. For the feed-forward layer, an example of the encoder is $h(Z, w) = W_2 \sigma_M(W_1 Z)$, as detailed in [6, 38, 73], where $w = \text{vec}[W_1, W_2]$ and the activation $\sigma_M$ is component-wise ReLU. Alternatively, setting $h \equiv 0$ results in a Transformer block that comprises only the self-attention layer, referred to as "attention-only" Transformers, as discussed in [6, 46, 49, 66].

Next, we analyze a Transformer network composed of $L$ Transformer blocks, referring to $L$ as the *depth* of the model. In this paper, we introduce an additional term, $\eta$, in (2.1) and (2.2) to simulate the model's evolution in a residual manner. We set the step size $\eta$ as $\Delta t/2$, where $\Delta t = 1/L$. As $L$ increases, $\Delta t$ approaches zero, allowing Transformer blocks to incrementally contribute to the model's overall progression. The structure of the network is then defined as follows:

$$\begin{cases} \widehat{T}_\Theta(H, t + \Delta t/2) & = \text{Attn}_{\theta_{t,1}, \ldots, \theta_{t,M}}(\widehat{T}_\Theta(H, t), \Delta t/2) \\ \widehat{T}_\Theta(H, t + \Delta t) & = \text{MLP}_{w_{t,1}, \ldots, w_{t,M}}(\widehat{T}_\Theta(H, t + \Delta t/2), \Delta t/2) \end{cases} \tag{2.4}$$

for each $t = 0, \Delta t, \ldots, (L-1)\Delta t$ with $\widehat{T}_\Theta(H, 0) = H$. We abbreviate the subscript $t = 0, \Delta t, \ldots, (L-1)\Delta t$ by $t$ and $j = 1, 2, \ldots, M$ by $j$ for simplicity. Here, $\Theta = \{\theta_{t,j}, w_{t,j}\}_{t,j}$ denotes all parameters in the Transformer model.

Throughout this paper, we treat $D$ and $N$ as bounded finite values, while $M$ and $L$ are treated as diverging, aligning with the setting of large-scale Transformers.

## 2.3 Gradient flow

For the $l_2$ regularization with $\lambda > 0$, we consider training the constructed Transformer model using the following $\lambda$-regularized risk objective:

$$\widehat{Q}(\Theta) = \widehat{R}(\Theta) + \frac{\lambda}{2ML} \sum_t \sum_{j=1}^M (\|\theta_{t,j}\|_2^2 + \|w_{t,j}\|_2^2), \tag{2.5}$$

with the population squared risk function defined as

$$\widehat{R}(\Theta) = \mathbb{E}_\mu \left[ \frac{1}{2} \left( \text{Read}[\widehat{T}_\Theta(H, 1)] - y(H) \right)^2 \right].$$

In Section 3.2, we will show that $l_2$-regularization on the parameter norms is essential for the well-posedness of the (Wasserstein) gradient flow to control parameter growth under our mild assumptions, even with a very small $\lambda > 0$. Similar strategies that consider necessary $l_2$ regularization are employed in [23] and [70]. Then, drawing on the methodologies in [6, 32, 46], our model processes the final output through a simple *read-out* function, $\text{Read}[\cdot]$, extracting the $(d+1, N+1)$-th entry of its input. We propose that this read-out layer can be expanded to any linear mapping with bounded parameter norm without affecting the validity of our theoretical results.

To minimize the objective function (2.5), we implement the *standard gradient flow method* as follows:

Step 1. Initially, for each $t = 0, \Delta t, \ldots, (L-1)\Delta t$, we sample $M$ particles $\theta_{t,j}^{(0)}, w_{t,j}^{(0)}$ with $j \in [M]$ independently from $\rho_0(\theta, w|t)$, where $\rho_0$ is a pre-defined distribution with bounded support.

Step 2. Then, we update all parameters $\theta_{t,j}^{(\tau)}, w_{t,j}^{(\tau)}$ in the set $\Theta^{(\tau)} = \{\theta_{t,j}^{(\tau)}, w_{t,j}^{(\tau)}\}_{t,j}$ using gradient flow (scaled by $ML$), which is defined as follows:

$$\frac{d\theta_{t,j}^{(\tau)}}{d\tau} = -ML \nabla_{\theta_{t,j}}[\widehat{Q}(\Theta^{(\tau)})], \quad \frac{dw_{t,j}^{(\tau)}}{d\tau} = -ML \nabla_{w_{t,j}}[\widehat{Q}(\Theta^{(\tau)})]. \tag{2.6}$$

Define the function $\widehat{R}(H; \Theta) = \frac{1}{2}(\text{Read}[\widehat{T}_\Theta(H, 1)] - y(H))^2$, and the partial derivative $\widehat{p}_\Theta(H, t) = \partial\widehat{R}(H; \Theta)/\partial\widehat{T}_\Theta(H, t)^\top$ for each $t = 0, \Delta t/2, \Delta t, \ldots, (L-1)\Delta t, (L-1/2)\Delta t$. Refer to Appendix C.4 for the explicit formula of $\widehat{p}_\Theta(H, t)$. Using the chain rule, we derive the explicit form of the gradient flow as follows:

$$\frac{d\theta_{t,j}^{(\tau)}}{d\tau} = -\widehat{G}_f(\theta_{t,j}^{(\tau)}, \Theta^{(\tau)}, t), \quad \frac{dw_{t,j}^{(\tau)}}{d\tau} = -\widehat{G}_h(w_{t,j}^{(\tau)}, \Theta^{(\tau)}, t). \tag{2.7}$$

where

$$\widehat{G}_f(\theta, \Theta, t) = \frac{1}{2}\mathbb{E}_\mu\Big[\nabla_\theta \mathrm{Tr}\Big(f(\widehat{T}_\Theta(H, t), \theta)^\top \widehat{p}_\Theta(H, t + \Delta t/2)\Big)\Big] + \lambda\theta,$$

$$\widehat{G}_h(w, \Theta, t) = \frac{1}{2}\mathbb{E}_\mu\Big[\nabla_w \mathrm{Tr}\Big(h(\widehat{T}_\Theta(H, t + \Delta/2), w)^\top \widehat{p}_\Theta(H, t + \Delta t)\Big)\Big] + \lambda w$$

for $t = 0, \Delta t, \ldots, (L-1)\Delta t$.

## 3 Approximation by the mean-field limit

In this section, we present a rigorous approximation result that bridges Transformer models in (2.4) with their mean-field limit as continuous Transformers. Thus, the width $M$ and depth $L$ in our proposed model are treated as discretization of this continuous limit in the parameter space.

### 3.1 Assumptions

In addition, we introduce the norm $\|\cdot\|_{2-\mathrm{col}}$ as the maximum $l_2$ norm across all columns of a matrix. We proceed under several mild assumptions related to the data distribution and the encoders $f$ and $h$.

**Assumption 1** (Data regularity). *There exists some universal constant $B > 0$ such that, for any $H \in \mathrm{supp}(\mu)$, we have $\max\{\|H\|_{2-\mathrm{col}}, y(H)\} \leq B$. In addition, a universal constant $K_y > 0$ ensures that $y(H)$ is $K_y$-Lipschitz continuous for $\|\cdot\|_F$ over $H \in \mathrm{supp}(\mu)$.*

**Remark** Assumption 1 is irrelevant to the Transformer model, and is only a fairly mild assumption on the data.

**Assumption 2** (Transformer particle growth bound). *We assume that the gradient of $f(T, \theta)$ and $h(T, w)$ exists. Furthermore, we have*

  i. $\|f(T, \theta)\|_{2-\mathrm{col}} \leq K\|T\|_{2-\mathrm{col}}(1 + \|\theta\| + \|\theta\|^2)$.

  ii. *For every $i \in [N+1]$, we have $\|\nabla_\theta f(T, \theta)_{:,i}\|_2 \leq \phi_P(\|T\|_{2-\mathrm{col}})(1 + \|\theta\|)$.*

  iii. $\|\nabla_{\mathrm{vec}[T]}\mathrm{vec}[f(T, \theta)]\|_2 \leq \phi_T(N, D, \|T\|_F)(1 + \|\theta\| + \|\theta\|^2)$.

*for some continuous, monotonically increasing functions $\phi_P, \phi_T$ for every coordinate, and a universal constant $K > 0$. Similarly, if we replace $f$ with $h$ and $\theta$ with $w$, the same conditions apply.*

**Remark** There are three key observations for Assumption 2. Firstly, it incorporates the $\|\cdot\|_{2-\mathrm{col}}$ norm, which is particularly useful for handling sequential inputs where each column represents a token. Secondly, as we consider higher-order multiplications between data and parameters, this assumption accommodates a broader range of self-attention encoders, such as the one in (2.3) with softmax or ReLU activation (where the derivative is defined as $\mathrm{ReLU}'(x) = \mathbf{1}\{x > 0\}$). Lastly, a particularly interesting and frontier question is identifying the function $\phi_T$, and we have listed related literature in Appendix B.

**Assumption 3** (Locally Lipschitz continuous gradient in expectation). *Besides Assumption 2, for any $L_T > 0$ and any $L_T$-Lipschitz continuous functions $T_1 = T_1(H)$ and $T_2 = T_2(H)$, for every $i \in [N+1]$, we have*

  i. $\mathbb{E}_\mu\|\nabla_\theta f(T_1, \theta)_{:,i} - \nabla_\theta f(T_2, \theta)_{:,i}\|_2 \leq \phi_{PT}(\|\theta\|, K_T, L_T)\sup_H\|T_1 - T_2\|_{2-\mathrm{col}}$,

  ii. $\mathbb{E}_\mu\|\nabla_{\mathrm{vec}[T]}\mathrm{vec}[f(T_1, \theta)] - \nabla_{\mathrm{vec}[T]}\mathrm{vec}[f(T_1, \theta')]\|_2 \leq \phi_{TP}(N, D, \sup_H\|T_1\|_F, K_P, L_T)\|\theta - \theta'\|$

  iii. $\mathbb{E}_\mu\|\nabla_\theta f(T_1, \theta)_{:,i} - \nabla_\theta f(T_1, \theta')_{:,i}\|_2 \leq \phi_{PP}(K_P, \sup_H\|T_1\|_{2-\mathrm{col}}, L_T)\|\theta - \theta'\|$,

  iv. $\mathbb{E}_\mu\|\nabla_{\mathrm{vec}[T]}\mathrm{vec}[f(T_1, \theta)] - \nabla_{\mathrm{vec}[T]}\mathrm{vec}[f(T_2, \theta)]\|_2 \leq \phi_{TT}(N, D, K_T, \|\theta\|, L_T)\sup_H\|T_1 - T_2\|_F$

*for $K_T = \max\{\sup_H\|T_1\|_{2-\mathrm{col}}, \sup_H\|T_2\|_{2-\mathrm{col}}\}$, $K_P = \max\{\|\theta\|, \|\theta'\|\}$, and some continuous functions $\phi_{PT}, \phi_{TP}, \phi_{PP}, \phi_{TT}$ that are monotonically increasing for every coordinate. Similarly, if we replace $f$ with $h$ and $\theta$ with $w$, the same conditions apply.*

**Remark** Assumption 3 states that functions are locally Lipschitz continuous in expectation, suitable for encoders that utilize ReLU functions and have second-order derivatives almost everywhere. This assumption is naturally satisfied if the activation has a locally Lipschitz continuous gradient.

Define $\mathcal{P}^2$ as the set of probability measures endowed with the Wasserstein-2 distance, where the Lipschitz continuity with respect to the depth holds, i.e. there exists some universal constant $C_\rho > 0$ such that $\|\rho(\cdot, t) - \rho(\cdot, t')\|_{\mathrm{BL}} \leq C_\rho |t - t'|$ for any $t, t' \in [0, 1]$.

**Choice of $\rho_0$** Suppose $\rho_0 \in \mathcal{P}^2$ satisfies that for any $t \in [0, 1]$, the support of $\rho_0(\cdot, \cdot, t)$ is contained within the set $\{(\theta, w) : \|\theta\|^2 + \|w\|^2 \leq R^2\}$ for a universal constant $R$. Additionally, for each $t \in [0, 1]$, it holds that $\int_{\theta, w} \rho_0(\theta, w, t) d(\theta, w) = 1$. This condition suits common bounded support distributions, and a natural choice is a uniform distribution across a disk with radius $R$ for each $t \in [0, 1]$.

We would like to clarify that verifying Assumptions 2 and 3 for concrete examples of Transformer architectures with smooth activation functions is fairly intuitive, and the proof is mainly based on a series of tedious calculations. We give a concrete proposition with its brief proof in Appendix G.

## 3.2 Continuous Transformer and Wasserstein gradient flow

Drawing inspiration from [47] and [67], which suggest that deep residual networks behave like ensembles of residual networks locally, we apply a similar manipulation to formulate the continuous version of (2.4). Consider the following continuous version $T_\rho(H, t) \in \mathbb{R}^{D \times (N+1)}$, governed by the following continuous ODE that *averages the two encoders*:

$$\dot{T}_\rho(H, t) = \int_{\theta, w} \frac{f(T_\rho(H, t), \theta) + h(T_\rho(H, t), w)}{2} \rho(\theta, w, t) d(\theta, w), \quad T_\rho(H, 0) = H \qquad (3.1)$$

In (3.1), each encoder $f$ or $h$ is conceptualized as a particle, and we consider the distribution of these particles denoted as $\rho(\theta, w, t)$. For any $\rho \in \mathcal{P}^2$ that have a bounded support, the well-posedness of $T_\rho(H, t)$ that satisfies the Transformer ODE (3.1) is shown in Proposition C.1. Transitioning to the framework with continuous Transformers, our objective shifts to minimizing the $l_2$ risk function with regularization on the second moment of $\rho$ as follows:

$$Q(\rho) = R(\rho) + \frac{\lambda}{2} \int_0^1 \int_{\theta, w} (\|\theta\|_2^2 + \|w\|_2^2) \rho(\theta, w, t) d(\theta, w) dt, \qquad (3.2)$$

with

$$R(\rho) = \mathbb{E}_\mu \left[ \frac{1}{2} \Big( \mathrm{Read}[T_\rho(H, 1)] - y(H) \Big)^2 \right]. \qquad (3.3)$$

Define $p_\rho(H, t) \in \mathbb{R}^{D \times (N+1)}$, the partial derivative of $R(\rho)$ relative to $T_\rho(H, t)$ at a local query point $H$, as the solution derived in Appendix C.4 using the classical adjoint sensitivity method [58]:

$$\mathrm{vec}[p_\rho(H, t)]^\top = \Big( \mathrm{Read}[T_\rho(H, 1)] - y(H) \Big) \exp \Big( \int_t^1 \int_\beta \nabla_{\mathrm{vec}[T]} \mathrm{vec}[g(T_\rho(H, t), \beta) \rho(\beta, t) d\beta dt] \Big)_{DN+d+1,:}.$$

Using this, we can compute the functional derivative to $\rho$ as follows:

$$\frac{\delta Q}{\delta \rho}(\theta, w, t) = \mathbb{E}_\mu \left[ \mathrm{Tr} \Big( \left[ \frac{f(T_\rho(H, t), \theta) + h(T_\rho(H, t), w)}{2} \right]^\top p_\rho(H, t) \Big) \right] + \frac{\lambda}{2} (\|\theta\|_2^2 + \|w\|_2^2). \quad (3.4)$$

The following Proposition claims that $\frac{\delta Q}{\delta \rho}$ is indeed the derivative with respect to $\rho$ (specifically, the Fréchet derivative [30]) for the functional $Q(\rho)$.

**Proposition 3.1** (Functional derivative to $\rho$). *Under Assumptions 1 and 2, for any pair $\rho, \nu \in \mathcal{P}^2$ that have bounded supports, we have*

$$Q(\rho + \eta(\nu - \rho)) = Q(\rho) + \eta \Big\langle \frac{\delta Q}{\delta \rho}, \nu - \rho \Big\rangle + o(\eta),$$

*where $\frac{\delta Q}{\delta \rho}$ is defined in (3.4), and $\langle \frac{\delta Q}{\delta \rho}, \nu - \rho \rangle = \int_0^1 \int_{(\theta, w)} \frac{\delta Q}{\delta \rho} \cdot (\nu - \rho) d(\theta, w) dt \in \mathbb{R}$.*

Now, we are in a position to display the gradient flow of $\rho$ in the Wasserstein metric [16], given by a McKean-Vlasov type equation [4, 37, 54, 56]. Specifically, we study the following partial differential equation of the distribution $\rho^{(\tau)}(\theta, w, t)$:

$$
\begin{aligned}
\frac{d\rho^{(\tau)}(\theta, w, t)}{d\tau} &= \mathrm{div}_{(\theta,w)}\Big(\rho^{(\tau)}\nabla_{(\theta,w)}\frac{\delta Q}{\delta \rho}\Big|_{\rho=\rho^{(\tau)}}\Big) \\
&= \mathrm{div}_\theta\Big(\rho^{(\tau)}G_f(\theta, \rho^{(\tau)}, t)\Big) + \mathrm{div}_w\Big(\rho^{(\tau)}G_h(w, \rho^{(\tau)}, t)\Big),
\end{aligned}
\tag{3.5}
$$

where $\rho^{(0)} = \rho_0$, div is the divergence operator, and the gradient functions are defined as

$$
G_f(\theta, \rho, t) = \frac{1}{2}\mathbb{E}_\mu\Big[\nabla_\theta \mathrm{Tr}\Big(f(T_\rho(H, t), \theta)^\top p_\rho(H, t)\Big)\Big] + \lambda\theta,
$$

$$
G_h(w, \rho, t) = \frac{1}{2}\mathbb{E}_\mu\Big[\nabla_w \mathrm{Tr}\Big(h(T_\rho(H, t), w)^\top p_\rho(H, t)\Big)\Big] + \lambda w.
$$

Propositions D.1 and 3.2 provide the well-posedness of both gradient flow and Wasserstein gradient flow respectively. In both propositions, a $\lambda > 0$ is essential to stabilize the optimization process by controlling both the maximum and average norms across all parameters. If $\lambda$ is set to 0, it is only possible to establish the well-posedness of (3.5) over a finite maximal interval [47]. Similar adjustments to regularize the risk function are also noted in [23].

**Proposition 3.2** (Existence and uniqueness of Wasserstein gradient flow)**.** *Under Assumptions 1 and 2, there exists a unique solution $(\rho^{(\tau)})_{\tau \geq 0} \in \mathcal{P}^2 \times \mathbb{R}$ with $\rho^{(0)} = \rho_0$ for (3.5). Additionally, for any $\tau \geq 0$, we have*

  *i. $\rho^{(\tau)}$ has a bounded support $\{\theta, w : \|\theta\|^2 + \|w\|^2 \leq R_\tau\} \times [0,1]$, where $R_\tau = (R + 1)\exp(C_R\tau) - 1$ for some constant $C_R$ that only depends on $N, D, \lambda$ and the parameters of the assumptions.*

  *ii. $\int_0^1 (\|\theta\|^2 + \|w\|^2)\rho^{(\tau)}(\theta, w, t)d(\theta, w)dt \leq A_0^2$, where $A_0 := R^2 + \lambda^{-1}(2B^2 + 2B^2\exp(K(1 + R + R^2))^2)$.*

  *iii. $\int_{(\theta,w)} \rho^{(\tau)}(\theta, w, t)d(\theta, w) = 1$ for any $t \in [0,1]$.*

### 3.3   Approximation of large-scale Transformer

In this section, we discuss the general results associated with approximating our discrete Transformer model to its mean-field limit. First, we highlight that the minimization of the risk function with discretization, whether or not regularization is included, closely approximates the minimal risk achievable by continuous models.

**Proposition 3.3** (Global minimum approximation of discretization)**.** *Under Assumptions 1 and 2, we define $\mathcal{P}^{2,r}$ as the set of distributions in $\mathcal{P}^2$ concentrated on $\{(\theta, w) : \|\theta\|^2 + \|w\|^2 \leq r^2\} \times [0,1]$. for any $r > 0$. Then there exists a constant $C$ dependent on $N, D, r$ and the parameters of the assumptions such that*

$$
\inf_\Theta \widehat{R}(\Theta) \leq \inf_{\rho \in \mathcal{P}^{2,r}} R(\rho) + C\Big(L^{-1} + \sqrt{\frac{\log(L+1)}{M}}\Big),
$$

$$
\inf_\Theta \widehat{Q}(\Theta) \leq \inf_{\rho \in \mathcal{P}^{2,r}} Q(\rho) + C(1+\lambda)\Big(L^{-1} + \sqrt{\frac{\log(L+1)}{M}}\Big).
$$

Proposition 3.3 specifies that the distributions under consideration must have bounded support. While it is typically challenging to confirm whether the minimal risk is indeed achieved on a distribution with bounded support, this assumption is justified as $\lambda$ regulates parameter norms, implicitly encourages solutions residing in a compact region of the parameter space.

We now present the main theorem concerning the convergence of the gradient flow process to the Wasserstein gradient flow as outlined in (3.5). The proof with detailed explanation of the techniques used in Theorem 3.1 is provided in Appendix D.

**Theorem 3.1** (Gradient flow approximation of discretization)**.** *Define the empirical distribution as $\hat{\rho}^{(\tau)} := \frac{1}{ML}\sum_t \sum_{j=1}^M \delta(\theta_{t,j}^{(\tau)}, w_{t,j}^{(\tau)}, t)$ for any $\tau \geq 0$. Under Assumptions 1-3, we*

have that $(\hat{\rho}^{(\tau)})_{\tau \geq 0}$ *weakly converges to* $(\rho^{(\tau)})_{\tau \geq 0}$ *almost surely along any sequence such that* $L \to \infty, M/\log L \to \infty$. *Moreover, for any fixed* $\tau > 0$ *and any* $\delta > 0$, *with probability at least* $1 - 3\exp(-\delta)$ *with respect to the parameter initialization* $\Theta^{(0)}$, *we have*

  i. $\sup_{s \in [0,\tau]} |\text{Read}[\widehat{T}_{\Theta^{(s)}}(H,t)] - \text{Read}[T_{\rho^{(s)}}(H,t)]| \leq C\left(L^{-1} + \sqrt{\frac{\delta + \log(L+1)}{M}}\right)$

  ii. $\sup_{s \in [0,\tau]} |\widehat{R}(\Theta^{(s)}) - R(\rho^{(s)})| \leq C\left(L^{-1} + \sqrt{\frac{\delta + \log(L+1)}{M}}\right)$

  iii. $\sup_{s \in [0,\tau]} |\widehat{Q}(\Theta^{(s)}) - Q(\rho^{(s)})| \leq C\left(L^{-1} + \sqrt{\frac{\delta + \log(L+1)}{M}}\right)$

*for some constant* $C$ *that depends on on* $N, D, \tau, \lambda$ *and the parameters of the assumptions.*

Theorem 3.1 significantly advances our understanding by controlling the difference regarding both the Transformer output, the risk function, and the regularized risk function. It's noted that the difference bound in the model's approximation may increase, possibly exponentially [22, 23, 51], as the time horizon extends. As argued in [51], such behavior may be inherent to the systems being modeled.

Additionally, the technical uniqueness and innovation of this theorem contrast sharply with previous results from overparametrized ResNet models. Our analysis distinguishes itself in two ways. First, our discrete Transformer model (2.4) uniquely splits the averaged encoder $(f + h)/2$ into two distinct blocks with encoders $f$ and $h$. Second, we demonstrate uniform error control over any finite time interval $[0, \tau]$, enabling continuous monitoring of maximum error across the gradient flow's trajectory. In contrast, models in prior studies such as [22, 23] restricts the error analysis to a specific $s \in [0, \tau]$.

## 4 Global convergence of gradient flow

In this section, we explore the optimization problem for gradient flow in the context of the discrete Transformer model, focusing on our general global convergence results.

### 4.1 An additional assumption

To ensure the global convergence of gradient flow for our discrete Transformer model, we introduce the following assumption. While influenced by the work in [16, 22, 23, 47], our assumption is uniquely tailored to the context of Transformers:

**Assumption 4.** *There exists a pair* $(g, \alpha) \in \{(f, \theta), (h, w)\}$ *with a partition* $\alpha = (\alpha_1, \alpha_2)$ *such that*

  i. *(Partial 1-homogeneity) for any* $T \in \mathbb{R}^{D \times (N+1)}$ *and* $c \in \mathbb{R}$, *we have* $g(T, c\alpha_1, \alpha_2) = cf(T, \alpha_1, \alpha_2)$.

  ii. *(Universal kernel) a compact set* $\mathcal{K} \subset \mathbb{R}^{\dim(\alpha_2)}$ *ensures that the span of* $\left\{g(\cdot, \alpha) : \alpha \in \mathbb{R}^{\dim(\alpha_1)} \times \mathcal{K}\right\}$ *is dense in* $\mathcal{C}(\|T\|_{2-\text{col}} \leq B, \mathbb{R}^{D \times (N+1)})$ *for any* $B > 0$.

We emphasize that the universal kernel property, as discussed in [52], closely relates to the universal approximation abilities. Under our assumption, we require the universal approximation capabilities of *either* the self-attention encoder or the feed-forward encoder. In Appendix G, we provide a concrete example of Transformer architectures and verify the validity of Assumption 4.

The universal kernel property of the feed-forward layer encoder $h$ is well-established, particularly in two-layer neural network contexts [74]. Conversely, the universal approximation abilities of self-attention layers is a frontier research area, which, while not extensively covered in this paper, holds significant potential. Often labeled as "memorization capacity", this area is recently explored across multiple studies [27, 31, 38, 39, 49, 63, 73]. The interconnection between approximation abilities and memorization capacities is established in [38]. Notably, [49] investigated the expressive capabilities of one single multi-head softmax self-attention layer, thereby potentially validating our assumptions.

Finally, we posit that the universal kernel applies to $\alpha_2$ within a compact set, as the function's scale can be moderated by the homogeneous part $\alpha_1$. In scenarios where $\alpha_2$ and $\mathcal{K}$ are absent, our

assumption simplifies to that in [47], characterized by complete homogeneity. Conversely, in the absence of the $\alpha_1$ component, our framework aligns with [23] which necessitates a more stringent support condition for $\mathcal{K}$, as detailed later in Theorem 4.1.

## 4.2 Global convergence result

In this section, we establish the convergence properties of the optimization task for discrete Transformers through gradient flow dynamics.

**Theorem 4.1** (Global convergence up to $\lambda$). *Suppose that Assumptions 1-4 hold, and the Wasserstein gradient flow $(\rho^{(\tau)})_{\tau \geq 0}$ weakly converges to some $\rho_\infty \in \mathcal{P}^2$. If for some universal constant $R_\infty > 1$, the following two conditions hold:*

    *i. $(\rho^{(\tau)})_{\tau \geq 0}$ is concentrated on $\{\theta, w : \|\theta\|^2 + \|w\|^2 \leq R_\infty^2\} \times [0, 1]$ when $\tau$ is sufficiently large.*

    *ii. If Assumption 4 holds with $(g, \alpha) = (f, \theta)$, we assume there exists a $t^* \in [0, 1]$ such that the connected set $\mathrm{supp}(\rho_\infty(\cdot, t^*)) \supset \mathcal{D} \times \mathcal{K} \times \{w_0\}$, for some $w_0 \in \mathbb{R}^{\dim(w)}$ and $\mathcal{D} \subset \mathbb{R}^{\dim(\theta_1)}$ that separates $\{\theta_1 : \|\theta_1\| = 1/R_\infty\}$ and $\{\theta_1 : \|\theta_1\| = R_\infty\}$.*

    *ii'. If Assumption 4 holds with $(g, \alpha) = (h, w)$, we assume there exists a $t^* \in [0, 1]$ such that the connected set $\mathrm{supp}(\rho_\infty(\cdot, t^*)) \supset \times \{\theta_0\} \times \mathcal{K} \times \mathcal{D}$, for some $\theta_0 \in \mathbb{R}^{\dim(\theta)}$ and $\mathcal{D} \subset \mathbb{R}^{\dim(w_1)}$ that separates $\{w_1 : \|w_1\| = 1/R_\infty\}$ and $\{w_1 : \|w_1\| = R_\infty\}$.*

*Then, for any $\epsilon > 0$, there exists some $\tau_0 > 0$ such that*

$$\sup_{\tau \geq \tau_0} \widehat{R}(\Theta^{(\tau)}) \leq \epsilon + C_1 \left( L^{-1} + \sqrt{\frac{\delta + \log(L+1)}{M}} \right) + C_2 \lambda$$

*with probability at least $1 - 3\exp(-\delta)$ with respect to the parameter initialization $\Theta^{(0)}$ for any $\delta > 0$. Here, $C_1$ is some constant dependent only on $N, D, \tau_0, \lambda$ and the parameters of the assumptions, while $C_2$ depends only on $N, D, R_\infty$ and the parameters of the assumptions.*

Theorem 4.1 depicts the behavior of the risk function $\widehat{R}(\Theta^{(\tau)})$ as the training duration $\tau$ is sufficiently large. Specifically, $\widehat{R}(\Theta^{(\tau)})$ asymptotically approaches zero as both $L \to \infty$ and $M/\log L \to \infty$, with an additional term that scales with $\lambda$. This additional term attributes to the incorporation of a $\lambda$-weighted penalty on the norm of the parameters in our training objective $\widehat{Q}$. Consequently, by selecting an appropriately small $\lambda > 0$, the risk approximates zero, demonstrating global convergence to the minimum of $\widehat{R}$.

In addition, Theorem 4.1 posits some additional assumptions: the weak convergence of $\rho^{(\tau)}$, the long-time uniform boundedness, and the separation property for $\alpha_1$ with the support expansion of $\alpha_2$ to $\mathcal{K}$. Similar assumptions are made in the literature of deep model optimization theory [22, 23, 47]. While these types of assumptions are typically challenging to justify, we provide high-level justifications for them in Appendix C.5, deferring detailed verification to future research.

We then present a corollary that directly follows from Theorem 4.1:

**Corollary 4.1.** *Continuing with the notations and assumptions from Theorem 4.1, suppose $\lambda \leq C_\lambda \epsilon$ for some universal constant $C_\lambda > 0$. Then, for any $\delta > 0$, constants $\tau_0, L_0, K_0 > 0$ can be found such that:*

$$\sup_{\tau \geq \tau_0, L > L_0, M/\log L > K_0} \widehat{R}(\Theta^{(\tau)}) \leq (1/2 + C_2 C_\lambda)\epsilon \tag{4.1}$$

*with probability at least $1 - 3\exp(-\delta)$ with respect to the parameter initialization $\Theta^{(0)}$. Notably, if $C_\lambda \leq (2C_2)^{-1}$, the upper bound in (4.1) is less than or equal to $\epsilon$.*

Corollary 4.1 claims that with a fixed $\delta > 0$, for any $\epsilon > 0$, we can achieve an order of $\epsilon$-close approximation with sufficiently large $L$ and $M$. Though our result is asymptotic and does not involve an explicit rate, it is the first of its kind and lays the groundwork for future theoretical optimization guarantees for Transformers.

# 5 Proof ideas of main theorems

Given the technical nature of this paper, this section presents the key ideas behind the proof of our main novel results, along with an outline of the proof preparation.

**Idea for Theorem 3.1** This convergence is described in two parts. First, the finite-time result (points (i)-(iii)) uses *propagation of chaos* [62] to analyze how differences evolve over time, comparing the evolution of parameter particles in discrete and continuous dynamics. The approximation bound is derived using a third auxiliary dynamic ("nonlinear dynamics"), involving the triangle inequality and Grönwall's inequality, which allows us to bound output differences over time.

Second, weak convergence of the empirical distribution process relies on optimal transport theory and stability results for Wasserstein gradient flows [4], focusing on the convergence of *momentum fields* [4, 60]. This also requires bounding the gradient differences between discrete and continuous Transformers as they approach the mean-field limit. See Appendix D for a detailed illustration, including a description of each main step.

**Idea for Theorem 4.1** We first establish the continuity of the functional gradient $\frac{\delta Q}{\delta \rho}\big|_{\rho_\infty}$, ensuring it remains constant if the derivative with respect to $\beta$ is constant over a region. Next, we derive the key bound for $Q(\rho_\infty)$, which is proportional to $\lambda$, by analyzing the functional energy $Q$'s landscape through its derivatives.

Finally, we show that the finite-time risk can approach this bound. Achieving $\epsilon$-level loss requires $Q(\rho^{(\tau_0)}) \leq \epsilon$ for some large $\tau_0$. Applying Theorem 3.1, we show $\widehat{Q}(\tau_0)$ becomes sufficiently small, and since $\widehat{Q}(\rho^{(\tau)})$ is non-increasing, it remains small for $\tau \geq \tau_0$. See Appendix E for a detailed illustration, including a description of each main step.

**Proof preparation for the main theorems** Appendix C.3 lists several useful lemmas essential to the main results. Specifically, Lemmas C.1–C.6 ensure the boundedness of key components and bound the output differences between discrete and continuous Transformers under different parameter settings. This boundedness is non-trivial due to the mild Assumptions 2 and 3 that fit the Transformer architecture. The technical lemmas in Appendix C.3 form the foundation for all subsequent proofs. Before introducing the nonlinear dynamics used to bound parameter differences under non-i.i.d. settings, these lemmas first establish an important oracle approximation bound result (Lemma D.6) with i.i.d. parameter settings. Additionally, they serve as key tools for bounding the (functional) gradient differences between Transformer dynamics, as shown in Lemmas D.1–D.3, which are essential for proving the approximation bound in Theorem 3.1.

In Theorem 4.1, Lemma E.1 plays a key role by demonstrating that for any $\rho$, there is always a nearly descent direction around $\rho$ for $Q(\rho)$, implying that all local minima are nearly global. This motivates further landscape analysis for bounding $\frac{\delta Q}{\delta \rho}\big|_{\rho_\infty}$ in the main theorem.

# 6 Conclusion

We conclude by summarizing our key contributions and suggesting future research directions. This paper establishes the global convergence of large-scale Transformer models through gradient flow dynamics, providing a thorough theoretical foundation. Our analysis, focused on the mean-field limit with infinite width and depth, shifts optimization from parameter space to distributional probability measures. We present two main theorems: one confirming the close approximation between discrete and continuous gradient flows, and another demonstrating global convergence, highlighting that basic optimization methods can successfully navigate complex landscapes to find optimal solutions. The techniques and results from this study lay the groundwork for further exploration into Transformer optimization. Future work could explore direct gradient descent with specific focus on step sizes, and expand on the in-context learning approximation capabilities of Transformers, as initiated by [6]. Additionally, it's crucial to rigorously assess under what conditions can self-attention layers serve as universal kernels to enhance our theoretical understanding, and to determine the generalization error bounds of Transformers trained on finite samples. These directions promise to deepen the theoretical and practical insights into Transformer models.

## Acknowledgments and Disclosure of Funding

We thank the anonymous reviewers for their helpful comments. Yuan Cao is partially supported by NSFC 12301657 and Hong Kong RGC-ECS 27308624. Mengdi Wang acknowledges the support by NSF IIS-2107304, NSF CPS-2312093, ONR 1006977 and Genmab. Han Liu's research is partially supported by the NIH R01LM01372201. Jason M. Klusowski was supported in part by the National Science Foundation through CAREER DMS-2239448, DMS-2054808 and HDR TRIPODS CCF-1934924. Jianqing Fan's research was partially supported by NSF grants DMS-2210833, DMS-2053832, and ONR grant N00014-22-1-2340.

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

# A  Overview of Appendix

The appendix is organized as follows:

- **Appendix B:** Additional related works are discussed.
- **Appendix C:**
    - In Appendix C.1, additional notations and preliminary details are introduced.
    - In Appendix C.2, we show the proof of Proposition C.1, concerning the existence and uniqueness of the continuous Transformer ODE (3.1).
    - In Appendix C.3, useful lemmas for the main proofs are detailed.
    - In Appendix C.4, the explicit formulas for $p_\rho(H, t)$ and $\widehat{p}_\Theta(H, t)$ are explored via the adjoint sensitivity method.
    - In Appendix C.5, high-level explanations are provided to substantiate the assumptions made in Theorem 4.1.
- **Appendix D:** Includes proofs of main results from Section 3.
- **Appendix E:** Includes proofs of main results from Section 4.
- **Appendix F:** Includes proofs of all auxiliary technical results mentioned in Appendix C-E.
- **Appendix G:** The verification of Assumptions 2–4 for a concrete example of Transformer architectures is provided.
- **Appendix H:** We provide simple experiment results and discuss the widths and depths of Vision Transformers that guarantee convergence, achieving near-zero training loss and 100% training accuracy on the CIFAR-10 dataset.

# B  Additional related work

**Theory of Transformers.** Some very recent works have studied theoretical properties of Transformer models from different aspects. [75, 33] studied the in-context learning guarantees for single-layer Transformers to perform linear regression predictions after being trained with linear regression example tasks. [1, 6, 32] studied the in-context learning capability of Transformers through the function approximation point of view, and demonstrated that there exists Transformers with specific parameter configurations that can perform particular in-context learning tasks. [36, 45] investigated how single-layer Transformers can be trained to learn simple image models and topic models respectively.

The most closely related work to ours is [41], which is the only study we are aware of that addresses the general and universal in-context learning capability of large-scale Transformers through optimization dynamics. It shows that a two-layer MLP followed by a linear attention layer can approximate functions in a general Barron space sufficiently well as the Transformer width increases. Additionally, its corresponding mean-field dynamics, via Wasserstein gradient flow, converges to global minima for in-context feature learning. Another work exploring the mean-field limit of Transformers is [9], which examines the limit as the depth, key-query length, and number of heads increase to infinity.

In addition, we have noticed a lot of theoretical interest in identifying the optimal choice of $\phi_T$ in Assumption 2, i.e., the Lipschitz constant of the Jacobian matrix of the self-attention term. For instance, [19] suggest that $\phi_T$ can be bounded by $\sqrt{N/D} + \mathrm{poly}(\|T\|_F)$, where $\mathrm{poly}(\cdot)$ denotes a polynomial function. In the context of $l_2$ self-attention, [39] find $\phi_T$ to be $\sqrt{N \log N/D}$, which notably does not depend on $\|T\|_F$, and [69] demonstrate that for $l_1$ distance metrics in attention layers, $\phi_T$ could be $\sqrt{D \log N}$.

**Global convergence of fully connected neural networks.** A line of recent works have studied the global convergence of (stochastic) gradient descent in training overparameterized neural networks in the mean-field regime [16, 51, 50, 70, 28, 29]. They consider the limit of the neural network as the width of the network at each layer goes to infinity, and models the limit of the network as a functional of the distribution of network parameters. A separate line of works also established the global convergence guarantees for training overparameterized neural networks in the "neural tangent kernel" regime [35, 3, 25, 76, 17, 2, 5, 11], where the gradient descent training iterates are

asymptotically equivalent to the training iterates of kernel regression based on the neural tangent kernel.

**Connection between ordinary differential equation models and infinite-depth ResNets.** Our work is also closely related to the recent literature aiming to understand ResNets by analyzing their connections to ordinary differential equations [71, 12, 43, 42, 72, 26, 22, 47, 23, 44, 7, 15, 13]. Specifically, [71, 12, 44, 15] studied the approximation of flow-based networks via discrete networks. [43, 42, 72, 26, 22, 47, 23, 44, 7] studied the optimization of the infinite-depth and infinite-width ResNets. [13] studied the generalization properties of the ResNet trained in the mean-field regime.

# C   Proof setup

## C.1   Additional technical notations

Define

$$\beta := (\theta, w)^\top, \quad g(T, \beta) := \frac{f(T, \theta) + h(T, w)}{2}.$$

Thus, $\frac{\delta Q}{\delta \rho}$ could be expressed as

$$\frac{\delta Q}{\delta \rho}(\beta, t) = \mathbb{E}_\mu \Big[ \mathrm{Tr} \Big( \big[ g(T_\rho(H, t), \beta) \big]^\top p_\rho(H, t) \Big) \Big] + \frac{\lambda}{2} \|\beta\|_2^2. \tag{C.1}$$

Additionally, we can combine $G_f$ with $G_h$, and $\widehat{G}_f$ with $\widehat{G}_h$ to reformulate as

$$G(\beta, \rho, t) = \mathbb{E}_\mu \Big[ \nabla_\beta \mathrm{Tr} \Big( g(T_\rho(H, t), \beta)^\top p_\rho(H, t) \Big) \Big] + \lambda \beta, \tag{C.2}$$

and

$$\widehat{G}(\beta, \Theta, t) = \mathbb{E}_\mu \Big[ \nabla_\beta \mathrm{Tr} \Big( \Big\{ \begin{matrix} f(\widehat{T}_\Theta(H, t), \theta)/2 \\ h(\widehat{T}_\Theta(H, t + \Delta t/2), w)/2 \end{matrix} \Big\}^\top \Big\{ \begin{matrix} \widehat{p}_\Theta(H, t + \Delta t/2) \\ \widehat{p}_\Theta(H, t + \Delta t) \end{matrix} \Big\} \Big) \Big] + \lambda \beta. \tag{C.3}$$

**Remark 1.** *To facilitate the proof, we restate Assumptions 2 and 3 for $g(T, \beta)$. Under Assumption 2, the gradient of $g(T, \beta)$ respect to $T$ and $\beta$ exists. Additionally, we have*

*Under Assumption 2:*

   *i.* $\|g(T, \beta)\|_{2-\mathrm{col}} \leq K \|T\|_{2-\mathrm{col}} (1 + \|\beta\| + \|\beta\|^2)$

   *ii. For every $i \in [N + 1]$, we have $\|\nabla_\beta g(T, \beta)_{:,i}\|_2 \leq \phi_P(\|T\|_{2-\mathrm{col}})(1 + \|\beta\|)$*

   *iii.* $\|\nabla_{\mathrm{vec}[T]} \mathrm{vec}[g(T, \beta)]\|_2 \leq \phi_T(N, D, \|T\|_F)(1 + \|\beta\| + \|\beta\|^2)$

*Under Assumption 3: For any $L_T > 0$ and any $L_T$-Lipschitz continuous functions $T_1 = T_1(H)$ and $T_2 = T_2(H)$, for every $i \in [N + 1]$, we have*

   *i.* $\mathbb{E}_\mu \|\nabla_\theta g(T_1, \beta)_{:,i} - \nabla_\theta g(T_2, \beta)_{:,i}\|_2 \leq \phi_{PT}(\|\theta\|, K_T, L_T) \sup_H \|T_1 - T_2\|_{2-\mathrm{col}},$

   *ii.* $\mathbb{E}_\mu \|\nabla_{\mathrm{vec}[T]} \mathrm{vec}[g(T_1, \beta)] - \nabla_{\mathrm{vec}[T]} \mathrm{vec}[g(T_1, \beta')]\|_2 \leq \phi_{TP}(N, D, \sup_H \|T_1\|_F, K_P, L_T) \|\theta - \theta'\|$

   *iii.* $\mathbb{E}_\mu \|\nabla_\theta g(T_1, \beta)_{:,i} - \nabla_\theta g(T_1, \beta')_{:,i}\|_2 \leq \phi_{PP}(K_P, \sup_H \|T_1\|_{2-\mathrm{col}}, L_T) \|\theta - \theta'\|,$

   *iv.* $\mathbb{E}_\mu \|\nabla_{\mathrm{vec}[T]} \mathrm{vec}[g(T_1, \beta)] - \nabla_{\mathrm{vec}[T]} \mathrm{vec}[g(T_2, \beta)]\|_2 \leq \phi_{TT}(N, D, K_T, \|\theta\|, L_T) \sup_H \|T_1 - T_2\|_F$

*Verifying all these results above only needs the basic triangle inequality of general norms, so we omit the trivial proof. We will apply them directly throughout the proofs of results. Additionally, we omit writing $L_T$ for simplicity in the proof, as all functions applied to Assumption 3 will be Lipschitz continuous with some universally bounded Lipschitz constant.*

Next, we introduce some additional technical notations. Denote the identity matrix with $d$-dimension as $I_d$. Define the sample space $\Omega := \mathbb{R}^{\dim \beta} \times [0, 1]$, and $\mathcal{P}(\Omega)$ as the probability measure space

defined on $\Omega$. For any $\Theta = \{\beta_{t,j}\}_{t/\Delta t+1\in[L],j\in[M]}$ and $\widetilde{\Theta} = \{\widetilde{\beta}_{t,j}\}_{t/\Delta t+1\in[L],j\in[M]}$, define
$d(\Theta, \widetilde{\Theta}) = \sum_t \sum_{j=1}^M \|\beta_{t,j} - \widetilde{\beta}_{t,j}\|$. Define the local risk function as

$$R(H; \rho) = \frac{1}{2}\Big(\text{Read}[T_\rho(H,1)] - y(H)\Big)^2.$$

Define the nested family of compact subsets $(P_r)_{r>0}$ as

$$P_r := \{\beta : \|\beta\| \le r\} \times [0,1], \quad \forall r > 0.$$

For any $\rho, \nu \in \mathcal{P}(\Omega)$ and $p \ge 1$, define the $l_p$ distance $\|\rho - \nu\|_p$ as

$$\|\rho - \nu\|_p = \Big(\int_0^1 \int_\beta |\rho(x) - \nu(x)|^p d\beta dt\Big)^{1/p}.$$

Specifically, when $p = 1$, we have

$$W_1(\rho, \nu) = \sup\Big\{ \int_0^1 \int_\beta f(\rho - \nu)d\beta dt : f \text{ is } 1-\text{Lipschitz}, \text{f}(\mathbf{0}, 0) = 0\Big\}$$

$$\le \sup\Big\{ \int_0^1 \int_\beta |f||\rho - \nu|d\beta dt : f \text{ is } 1-\text{Lipschitz}, \text{f}(\mathbf{0}, 0) = 0\Big\}$$

$$\le (r+1)\|\rho - \nu\|_1$$

for any $\rho, \nu \in \mathcal{P}^2$ concentrated on $P_r$. For simplicity, any $H$ discussed throughout this paper is assumed to lie within $\text{supp}(\mu)$.

## C.2  Transformer ODE existence and uniqueness

In this section, we establish the existence and uniqueness of the solution $T_\rho(H,t)$ to the ODE presented in (3.1) for any $H$, given that $\rho \in \mathcal{P}^2$ is concentrated on a bounded support, specifically, $P_r$ for some $r > 0$. This following proposition forms the cornerstone of the subsequent technical analyses:

**Proposition C.1** (Existence and uniqueness of Transformer ODE)**.** *Under Assumptions 1 and 2, for any $\rho \in \mathcal{P}^2$ that has a bounded support, there exists a unique solution of* (3.1) *on $t \in [0,1]$ that is Lipschitz continuous with respect to $(H,t)$.*

Initially, we demonstrate that the integral $\int_\beta \rho(\beta, t)d\beta$ is bounded. According to the definition $\mathcal{P}^2$, it follows that

$$\text{(C.4)}$$

$$|\int_\beta \rho(\beta, t)d\beta - \int_\beta \rho(\beta, t')d\beta| \le C_\rho |t - t'|$$

for any $t, t' \in [0,1]$. Integrating (C.4) over $t_2 \in [0,1]$ obtains

$$\int_\beta \rho(\beta, t)d\beta = 1 + \int_\beta \rho(\beta, t)d\beta - \int_0^1 \int_\beta \rho(\beta, t')d\beta dt'$$

$$\le 1 + C_\rho \int_0^1 |t - t'|dt' \qquad \text{(C.5)}$$

$$\le 1 + C_\rho/2.$$

For the remainder of the technical proof, we will employ (C.5) without additional elaboration.

*Proof of Proposition C.1.* **Step I: Create a small neighboring area with local Lipschitz continuity**

Consider any vector $\beta$ such that $\|\beta\| \le r$. Define $F(T,t) := \int_\beta g(T, \beta)\rho(\beta, t)d\beta$. For $T$ within the rectangle $\{T : \|T - H\|_{\max} \le \delta\}$, where $\delta > 0$ is bounded, both $\|T\|_{2-\text{col}}$ and $\|T\|_F$ are also bounded. Given Assumption 2 (i), $g(T, \beta)$ is universally bounded by some constant $K_{\delta,r}$. Moreover,

under Assumption 2 (ii) and (iii), $g(T, \beta)$ is Lipschitz continuity with some constant $L_{\delta,r}$. Hence, within the rectangle $\{T : \|T - H\|_{\max} \leq \delta\} \times [0, 1]$ the following properties hold:

$$|F(T, t_1) - F(T, t_2)| \leq \max\{K_{\delta,r}, L_{\delta,r}\}\|\rho(\cdot, t_1) - \rho(\cdot, t_2)\|_{\mathrm{BL}} \leq C_\rho \max\{K_{\delta,r}, L_{\delta,r}\}|t_1 - t_2|. \tag{C.6}$$

which indicates that $F(T, t)$ is continuous with respect to $t$ within the rectangle $\{T : \|T - H\|_{\max} \leq \delta\} \times [0, 1]$.

Moreover, within the bounded region $\{T : \|T - H\|_{\max} \leq \delta\}$, Assumption 2 (iii) ensures that $\|\nabla_{\mathrm{vec}[T]}\mathrm{vec}[g(T, \beta)]\|_2$ bounded. Consequently, $g(T, \beta)$ is Lipschitz-continuous with respect to $T$ for $\|\cdot\|_F$. Denote this Lipschitz constant by $L'_{\delta,r}$. Therefore, for any $T, T' \in \{T : \|T - H\|_{\max} \leq \delta\}$ and $t \in [0, 1]$, it follows that

$$|F(T, t) - F(T', t)| \leq L'_{\delta,r}\|T - T'\|_F \int_\beta \rho(\beta, t)d\beta \leq L'_{\delta,r}(1 + C_\rho/2)\|T - T'\|_F, \tag{C.7}$$

which deduces the Lipschitz continuity of $F(T, t)$ with respect to $T$ for $\|\cdot\|_F$.

**Step II: Show that the maximal existence interval is infinite by repeatedly using the Picard-Lindelöf Theorem**

Invoking the Picard-Lindelöf Theorem, there exists some $\epsilon > 0$ such that the initial value problem

$$\dot{T}(H, t) = F(T, t), \quad T(H, 0) = H$$

has a unique solution on $t \in [0, \epsilon]$. Given that this claim holds for any $H$, the standard ODE Extensibility Theorem guarantees a continuation of $T(t)$ to a maximal interval of existence, denoted as $[0, t_{\max}]$.

Assume by contradiction that $t_{\max} < 1$. From (3.1) and Assumption 2(i), for any $t \in [0, t_{\max}]$ we see that

$$\begin{aligned}
\frac{d}{dt}\|T_\rho(H, t)\|_{2-\mathrm{col}} \leq \|\dot{T}_\rho(H, t)\|_{2-\mathrm{col}} &= \|\int_\beta g(T_\rho(H, t), \beta)\rho(\beta, t)d\beta\|_{2-\mathrm{col}} \\
&\leq \int_\beta \|g(T_\rho(H, t), \beta)\|_{2-\mathrm{col}}\rho(\beta, t)d\beta \\
&\leq \int_\beta K(1 + \|\beta\|_2 + \|\beta\|_2^2)\rho(\beta, t)\|T_\rho(H, t)\|_{2-\mathrm{col}}d\beta.
\end{aligned} \tag{C.8}$$

Therefore, by the Grönwall's inequality, we have

$$\begin{aligned}
\|T_\rho(H, t_{\max})\|_{2-\mathrm{col}} &\leq \|T_\rho(H, 0)\|_{2-\mathrm{col}} \exp\left(\int_0^\top \int_\beta K(1 + \|\beta\|_2 + \|\beta\|_2^2)\rho(\beta, s)d\beta ds\right) \\
&\leq \|H\|_{2-\mathrm{col}} \exp(K(1 + C_\rho/2 + r + r^2)t_{\max}) < \infty.
\end{aligned}$$

This presents a contradiction to the notion that $t_{\max} < 1$. This is because, By reapplying the local Picard-Lindelöf Theorem using the state $T(H, t_{\max})$ as the new initial condition, we can extend the interval of existence beyond $t_{\max}$. Consequently, we must conclude that $t_{\max} = 1$, and the existence and uniqueness follows.

**Step III: Show that the Lipschitz continuity with respect to $(H, t)$**

In the final part of our proof, we demonstrate that $T_\rho(H, t)$ is Lipschitz continuous with respect to $(H, t)$ for $H \in \mathrm{supp}(\mu)$ and any $t \in [0, 1]$. Given that $T_\rho(H, t)$ is universally bounded within $H \in \mathrm{supp}(\mu)$ and any $t \in [0, 1]$, we only need to focus on establishing its Lipschitz continuity with respect to $H$ and $t$ separately. The Lipschitz continuity with respect to $t$ is derived from

$$\|T_\rho(H, t_1) - T_\rho(H, t_2)\|_{2-\mathrm{col}} \leq \int_{t_1}^{t_2} \int_\beta \|g(T_\rho(H, t), \beta\|_{2-\mathrm{col}}\rho(\beta, t)d\beta dt \leq (1 + C_\rho/2)KC(1 + r + r^2)(t_2 - t_1), \tag{C.9}$$

for any $t_1, t_2 \in [0, 1]$. Given Assumption 2 (iii), we have

$$\|T_\rho(H,t) - T_\rho(H',t)\|_F \leq \int_0^\top \int_\beta \|g(T_\rho(H,s),\beta) - g(T_\rho(H',s),\beta)\|_F \rho(\beta,s)d\beta ds$$

$$\leq \int_0^\top \int_\beta \phi_T(N,D,B\exp(K(1+C_\rho/2+r+r^2)))(1+r+r^2)\|T_\rho(H,s) - T_\rho(H',s)\|_F \rho(\beta,s)d\beta ds$$

$$\text{(C.10)}$$

for any $H, H' \in \text{supp}(\mu)$. Define $L_H := \phi_T(N,D,B\exp(K(1+C_\rho/2+r+r^2)))(1+r+r^2)$. Utilizing Grönwall's inequality, we establish:

$$\|T_\rho(H,t) - T_\rho(H',t)\|_F \leq \|H - H'\|_F \exp\left(L_H \int_0^1 \int_\beta \rho(\beta,s)d\beta ds\right) = \exp(L_H)\|H - H'\|_F$$

given that $T_\rho(\cdot, 0)$ serves as the identity mapping. Consequently, $T_\rho(H,t)$ demonstrates Lipschitz continuity with respect to $H \in \text{supp}(\mu)$. □

## C.3 Useful technical lemmas

**Lemma C.1** (Continuous Transformer output bound). *Under Assumption 2, for any distribution $\rho \in \mathcal{P}(\Omega)$ where $\int_0^1 \int_\beta \|\beta\|_2^2 \rho(\beta,t)d\beta dt \leq A^2$ for some constant $A > 0$ and for any $t \in [0,1]$, we have*

$$\|T_\rho(H,t)\|_{2-\text{col}} \leq \|H\|_{2-\text{col}} \exp(K(1 + A + A^2)).$$

**Lemma C.2** (Continuous Transformer difference bound). *Under Assumption 2 and given $H$, for any $\rho, \nu \in \mathcal{P}^2$ that satisfy $\int_0^1 \int_\beta \|\beta\|_2^2 \rho(\beta,t)d\beta dt \leq A^2$ and have bounded supports $P_r$ for some constants $A, r > 0$, we have that*

$$\sup_{t \in [0,1]} \|T_\rho(H,t) - T_\nu(H,t)\|_F \leq C_r W_1(\rho,\nu).$$

*Here, the universal constant $C_r$ only depends on $N, D, r, A$, and the parameters of the assumptions.*

**Lemma C.3** (Continuous Transformer gradient component bound). *Under Assumption 1 and 2, for any $\rho \in \mathcal{P}(\Omega)$ where $\text{supp}(\rho) \subset P_r$ and $\int_0^1 \int_\beta \|\beta\|^2 \rho(\beta,t)d\beta dt \leq A^2$, we have*

$$\sup_{(\beta,t)\in P_r} \|g(T_\rho(H,t),\beta)\|_F \leq \sqrt{N+1}KB\exp(K(1+A+A^2))(1+r+r^2),$$

$$\sup_{(\beta,t)\in P_r} \|p_\rho(H,t)\|_F \leq (B + B\exp(K(1+A+A^2)))\exp\left(\phi_T(N,D,\sqrt{N+1}KB\exp(K(1+A+A^2))(1+A+A^2))\right),$$

$$\sup_{(\beta,t)\in P_r} \left|\frac{\delta Q}{\delta \rho}(\beta,t)\right| \leq \sqrt{N+1}KB\exp(K(1+A+A^2))(1+r+r^2)(B + B\exp(K(1+A+A^2)))$$

$$\exp\left(\phi_T(N,D,\sqrt{N+1}KB\exp(K(1+A+A^2))(1+A+A^2))\right) + \frac{\lambda}{2}r^2.$$

**Lemma C.4** (Discrete Transformer bound). *Under Assumptions 2, for any $\Theta$ where $\frac{1}{ML}\sum_t \sum_{j=1}^M \|\beta\|^2 \leq A^2$ for some universal constant $A > 0$ and at any $t = 0, \Delta t/2, \Delta t, \ldots, (L-1/2)\Delta t, 1$, we have*

$$\|\widehat{T}_\Theta(H,t)\|_{2-\text{col}} \leq \|H\|_{2-\text{col}} \exp(K(1 + A + A^2)).$$

**Lemma C.5** (Discrete Transformer difference bound). *Under Assumption 1 and 2, for any $H$, let $\Theta = \{\beta_{t,j}\}_{t/\Delta t+1\in[L],j\in[M]}, \widetilde{\Theta} = \{\widetilde{\beta}_{t,j}\}_{t/\Delta t+1\in[L],j\in[M]}$ such that $\max\{\|\beta_{t,j}\|, \|\widetilde{\beta}_{t,j}\|\} \leq r$ for any $t = 0, \ldots, (L-1)\Delta t, j = 1, \ldots, M$. Then, we have that*

$$\|\widehat{T}_\Theta(H,t) - \widehat{T}_{\widetilde{\Theta}}(H,t)\|_F \leq C_r \frac{1}{ML}d(\Theta, \widetilde{\Theta}).$$

*Here the universal constant $C_r$ only depends on $N, D, r$, and the parameters of the assumptions.*

**Lemma C.6** (Discrete Transformer gradient component bound). *Under Assumption 1 and 2, for any* $\Theta$ *such that* $\sup_{t,j}\|\beta_{t,j}\|^2 \le r^2$ *and* $\frac{1}{ML}\sum_t\sum_{j=1}^M\|\beta\|^2 \le A^2$, *we have, for any* $t = 0, \Delta t/2, \ldots, (L-1/2)\Delta t, 1$

$$\sup_{(\theta,t)\in P_r} \max\{\|f(\widehat{T}_\Theta(H,t),\theta)\|_F, \|h(\widehat{T}_\Theta(H,t),w)\|_F, \|g(\widehat{T}_\Theta(H,t),\beta)\|_F\} \le \sqrt{N+1}KB_T(1+r+r^2),$$

$$\sup_{(\beta,t)\in P_r} \sup_{i\in[N+1]} \max\left\{\left\|\nabla_\theta f(\widehat{T}_\Theta(H,t),\theta)_{:,i}\right\|, \left\|\nabla_w h(\widehat{T}_\Theta(H,t),w)_{:,i}\right\|, \left\|\nabla_\beta g(\widehat{T}_\Theta(H,t),\beta)_{:,i}\right\|\right\} \le \phi_P(B_T)(1+r),$$

$$\sup_{(\beta,t)\in P_r} \|\widehat{p}_\Theta(H,t)\|_F \le (B+B_T)\exp\left(\phi_T(N,D,\sqrt{N+1}KB_T)(1+A+A^2)\right),$$

*where* $B_T = B\exp(K(1+A+A^2))$.

**Lemma C.7** (Norm average concentration). *Under Assumption 2, consider a parameter setting* $\Theta = \{\beta_{t,j}\}_{t/\Delta t+1\in[L],j\in[M]}$ *i.i.d. drawn from* $\{\rho(\beta|t)\}_{t/\Delta t+1\in[L],j\in[M]}$ *where* $\rho \in \mathcal{P}^2$ *is concentrated on* $P_r$ *and satisfies* $\int_\beta \rho(\beta,t)d\beta = 1$ *for every* $t \in [0,1]$. *Then, with probability at least* $1 - \exp(-\delta)$ *with respect to the parameter initialization* $\Theta^{(0)}$, *we have*

$$\left|\frac{1}{ML}\sum_t\sum_{j=1}^M\|\beta_{t,j}\|^2 - \int_0^1\int_\beta\|\beta\|^2\rho(\beta,t)d\beta dt\right| \lesssim L^{-1} + \sqrt{\frac{\delta + \log(L+1)}{M}}.$$

*for any* $\delta > 0$. *Here, the* $\lesssim$ *notation hides the dependencies on* $r$ *and the parameters specified in the assumption.*

**Lemma C.8** (Matrix product difference bound). *Suppose that for some* $d > 0$, *the matrices* $A_1, A_2, \ldots, A_L$ *and* $B_1, B_2, \ldots, B_L$ *satisfy the following conditions:*

1. *For each* $l = 1, \ldots, L$, *the norms of the matrices are bounded as* $\|A_l\| \le 1 + a_l, \|B_l\| \le 1 + b_l$, *where* $a_l, b_l > 0$.

2. *The product of the increments for each matrix is bounded by* $\prod_{l=1}^L 1 + \max\{a_l, b_l\} \le C$ *for some constant* $C > 0$.

*Under these conditions, it holds that*

$$\left\|\prod_{l=1}^L A_l - \prod_{l=1}^L B_l\right\| \le C\sum_{l=1}^L \|A_l - B_l\|.$$

## C.4 Solution of adjoint ODE

In this section, we define the partial derivative

$$p_\rho(H,t) := \frac{\partial R(H;\rho)}{\partial T_\rho(H,t)^\top} \in \mathbb{R}^{D\times(N+1)}$$

without specifying its explicit formula. Denote the derivative of $T_\rho(H,1)$ to $T_\rho(H,t)$ (after vectorization) by the Jacobian $J_\rho(H,t) \in \mathbb{R}^{(N+1)D\times(N+1)D}$, and assume that $\dot{J}\rho(H,t)$ exists for any $t \in [0,1]$. Then [58] shows that $J_\rho(H,t)$ satisfies the adjoint equation of the ODE.

$$\dot{J}_\rho(H,t) = -J_\rho(H,t)\nabla_{\text{vec}[T]}\left\{\text{vec}[\int_\beta g(T_\rho(H,t),\beta)\rho(\beta,t)d\beta]\right\} \tag{C.11}$$

for any $t \in [0,1]$. By applying the chain rule and exchanging the order of the derivative and integral, we have, for any $t \in [0,1]$, that

$$\text{vec}[p_\rho(H,t)]^\top = \frac{\partial R(H;\rho)}{\partial\text{vec}[T_\rho(H,1)]}\frac{\partial\text{vec}[T_\rho(H,1)]}{\partial\text{vec}[T_\rho(H,t)]}$$

$$= \text{vec}[\frac{\partial R(H;\rho)}{\partial T_\rho(H,1)^\top}]^\top\frac{\partial\text{vec}[T_\rho(H,1)]}{\partial\text{vec}[T_\rho(H,t)]}$$

$$= \text{vec}[p_\rho(H,1)]^\top J_\rho(H,t)$$

Hence, by taking the derivative with respect to $t$, we obtain that

$$\text{vec}[\dot{p}_\rho(H,t)]^\top = -\text{vec}[p_\rho(H,t)]^\top \int_\beta \nabla_{\text{vec}[T]} \text{vec}[g(T_\rho(H,t),\beta)]\rho(\beta,t)d\beta$$

with the solution

$$\text{vec}[p_\rho(H,t)]^\top = \text{vec}[p_\rho(H,1)]^\top \exp\Big(\int_t^1 \int_\beta \nabla_{\text{vec}[T]} \text{vec}[g(T_\rho(H,s),\beta)]\rho(\beta,s)d\beta ds\Big). \quad \text{(C.12)}$$

On the other hand, we have

$$p_\rho(H,1) = \frac{\partial R(H;\rho)}{\partial T_\rho(H,1)} = (\text{Read}[T_\rho(H,1)] - y(H))E_{\text{read}}, \quad \text{(C.13)}$$

where $E_{\text{read}}$ is a $D \times (N+1)$ zero matrix except 1 at the $(d+1, N+1)-$th entry. Moreover, from (C.12) and (C.13), we see that

$$\text{vec}[p_\rho(H,t)]^\top = (\text{Read}[T_\rho(H,1)] - y(H)) \cdot \exp\Big(\int_t^1 \int_\beta \nabla_{\text{vec}[T]} \text{vec}[g(T_\rho(H,s),\beta)]\rho(\beta,s)d\beta ds\Big)_{DN+d+1,:}.$$

$$\text{(C.14)}$$

Additionally, we could explicitly derive the formula for $\widehat{p}_\Theta(H,t) = \frac{\partial \widehat{R}(H;\Theta)}{\partial \widehat{T}_\Theta(H,t)}$ for the discrete Transformer. By applying the chain rule multiple times across each layer with the encoder either $f$ or $h$, for any $t = 0, \Delta t, \ldots, (L-1)\Delta t, 1$, we have

$$\text{vec}[\widehat{p}_\Theta(H,t)] = (\text{Read}[\widehat{T}_\Theta(H,1)] - y(H))$$

$$\Big\{ \prod_{\substack{(s-t)/\Delta t+1 \in [(1-t)/\Delta t] \\ j \in [M]}} \Big(I_{\text{dimvec}[T]} + (\Delta t/2)M^{-1}\sum_{j=1}^M \nabla_{\text{vec}[T]} \text{vec}[f(\widehat{T}_\Theta(H,s), \theta_{s,j})]\Big)$$

$$\prod_{\substack{(s-t)/\Delta t+1 \in [(1-t)/\Delta t] \\ j \in [M]}} \Big(I_{\text{dimvec}[T]} + (\Delta t/2)M^{-1}\sum_{j=1}^M \nabla_{\text{vec}[T]} \text{vec}[h(\widehat{T}_\Theta(H,s+\Delta t/2), w_{s,j})]\Big)\Big\}_{DN+d+1,:},$$

$$\text{(C.15)}$$

and

$$\text{vec}[\widehat{p}_\Theta(H,t+\Delta t/2)] = (\text{Read}[\widehat{T}_\Theta(H,1)] - y(H))$$

$$\Big\{ \prod_{\substack{(s-t)/\Delta t+2 \in [(1-t)/\Delta t] \\ j \in [M]}} \Big(I_{\text{dimvec}[T]} + (\Delta t/2)M^{-1}\sum_{j=1}^M \nabla_{\text{vec}[T]} \text{vec}[f(\widehat{T}_\Theta(H,s), \theta_{s,j})]\Big)$$

$$\prod_{\substack{(s-t)/\Delta t+1 \in [(1-t)/\Delta t] \\ j \in [M]}} \Big(I_{\text{dimvec}[T]} + (\Delta t/2)M^{-1}\sum_{j=1}^M \nabla_{\text{vec}[T]} \text{vec}[h(\widehat{T}_\Theta(H,s+\Delta t/2), w_{s,j})]\Big)\Big\}_{DN+d+1,:}.$$

$$\text{(C.16)}$$

## C.5  Explanation for assumptions made in Theorem 4.1

The justification of the two assumptions outlined in Theorem 4 warrants careful consideration. While we provide only high-level justifications, they underpin significant aspects of our theoretical framework.

For the first assumption, we argue that the regularization parameter $\lambda$, which penalizes the magnitude of the parameter norms, implicitly promotes solutions that are confined to a compact subset of the parameter space. This rationale is conceptual and requires that regularization effectively constrains the growth of the parameter norms, thereby localizing the solutions.

The second assumption concerns the separation property. It is naturally satisfied as long as the origin $0_{\text{dim}\theta+\text{dim}w}$ remains an interior point of $\text{supp}(\rho_\infty(\cdot,t))$. This condition is relatively mild and

is generally satisfied. The challenge arises in verifying that $\text{supp}(\rho_\infty(\cdot, t))$ for the $\alpha_2$ component extends to encompass the entire space $\mathcal{K}$. While direct confirmation is elusive, it is suggested by [23] initially spans $\mathcal{K}$, this expansive support property is maintained at any finite time. Thus, we conjecture that the condition holds under these circumstances, providing a basis for this assumption.

# D  Proofs of main results in Section 3

## D.1  Proof of Theorem 3.1

This convergence is detailed in two parts. First, the finite time result, as stated in points (i)-(iii), utilizes a concept in probability theory known as *propagation of chaos* [62] to examine how differences evolve uniformly across a given time interval. In the context of our model, this involves comparing how parameter particles evolve under discrete versus continuous dynamics.

Specifically, the approximation bound is derived using a third auxiliary dynamic, termed the "nonlinear dynamics," by bounding the dynamic difference over the entire finite time interval. This process involves applying the triangle inequality to each component and concluding with a Grönwall's inequality. Since the Transformer output can be bounded by the dynamic difference, we can then bound the output difference at any specific time, along with the difference regarding different time for the same dynamic. By applying a probability union bound on a dense set of $L^2$ points, we can extend this to bound the maximal difference over any time interval.

Secondly, the weak convergence of the empirical distribution process leverages optimal transport theory alongside abstract stability results for Wasserstein gradient flows [4]. This argument involves detailed analysis of the discretization of particle distributions in space, particularly focusing on obtaining the convergence of the sequence of *momentum fields* [4, 60] that could directly leads to the result. To obtain the convergence of the momentum field sequence, we also need to bound the parameter gradient difference between discrete and continuous Transformers as the mean-field limit.

**Preparatory Step: Nonlinear dynamics**

We first define some auxiliary quantities and differential equations that are useful for the proof. For any gradient flow parameter setting $\Theta^{(\tau)} = \{\beta_{t,j}^{(\tau)}\}_{t,j}$, from its definition (2.7), we could rewrite the dynamics as

$$\beta_{t,j}^{(\tau)} = \beta^{(0)} - \int_0^\tau \widehat{G}(\beta_{t,j}^{(s)}, \Theta^{(s)}, t) ds \tag{D.1}$$

for any gradient flow time $\tau > 0$, depth index $t = 0, \Delta t, \ldots, (L-1)\Delta t$ and width index $j = 1, \ldots, M$. For simplicity, any mentioned constant only depends on $N, D, \tau, \lambda$ and the parameters of the assumptions, and we abbreviate the subscript $t = 0, \Delta t, \ldots, (L-1)\Delta t,\ j = 1, \ldots, M$ by $t, j$ throughout the proof.

Inspired by the "propagation of chaos" idea [62], we could define the "nonlinear dynamics" with the same initialization setting $\widetilde{\Theta}^{(\tau)} = \{\widetilde{\beta}_{t,j}^{(\tau)}\}_{t,j}$, i.e.

$$\begin{cases} \widetilde{\beta}_{t,j}^{(\tau)} = \widetilde{\beta}^{(0)} - \int_0^\tau G(\widetilde{\beta}_{t,j}^{(s)}, \rho^{(s)}, t) ds, \\ \widetilde{\beta}_{t,j}^{(0)} = \beta_{t,j}^{(0)} \end{cases} \tag{D.2}$$

for any $t, j$. Here, $(\rho^{(s)})_{s \geq 0}$ is the solution to the Wasserstein gradient flow (3.5), of which the uniqueness is implied by Proposition 3.2. Since (D.2) is just the particle flow of (3.5), its existence and uniqueness are guaranteed by Proposition 3.2.

Observing that $\{\beta_{t,j}\}_{t,j}$ are independent due to the dynamics only involving $(\rho^{(s)})_{s \geq 0}$ with i.i.d initialization over $\rho_0$, we can consider $\{\widetilde{\beta}_{t,j}^{(s)}\}_{t,j}$ as i.i.d. samples drawn from $\{\rho^{(s)}\}_{t,j}$. In addition, from Propositions 3.2 and D.1, for any $t, j$ we have $\max\{\|\beta_{t,j}\|, \|\widetilde{\beta}_{t,j}\|\} \leq R_\tau$, where $R_\tau$ is defined as in these propositions and does not depend on $M$ and $L$.

**Preparatory Step: Bound the gradient difference regarding parameter settings**

As the second preparatory step for the proof of Theorem 3.1, we present the following three lemmas that will be helpful:

**Lemma D.1** (Continuous gradient difference bound). *Suppose Assumptions 1-3 hold. If we have $\rho, \nu \in \mathcal{P}^2$ concentrated on $P_r$ for some $r > 0$, and $\beta, \widetilde{\beta}$ such that $\max\{\|\beta\|, \|\widetilde{\beta}\|\} \leq r$, then*

$$\left\| G(\beta, \rho, t) - G(\widetilde{\beta}, \nu, t) \right\| \leq C_G \Big( \exp(C_G W_1(\rho, \nu)) - 1 + (1 + \lambda) \|\beta - \widetilde{\beta}\| \Big).$$

*for any $t \in [0, 1]$. Here, the constant $C_G$ only depends on $N, D, r$, and the parameters of the assumptions.*

**Lemma D.2** (Discrete gradient difference bound). *Under Assumption 1-3, for any $\Theta = \{\beta_{t,j}\}_{t/\Delta t+1 \in [L], j \in [M]}, \widetilde{\Theta} = \{\widetilde{\beta}_{t,j}\}_{t/\Delta t+1 \in [L], j \in [M]}$ such that $\max\{\|\beta_{t,j}\|, \|\widetilde{\beta}_{t,j}\|\} \leq r$ for any $t = 0, \ldots, (L-1)\Delta t$, $j = 1, \ldots, M$. Then we have that*

$$\|\widehat{G}(\beta, \Theta, t) - \widehat{G}(\widetilde{\beta}, \widetilde{\Theta}, t)\| \leq C_G \Big( \frac{1}{ML} d(\Theta, \widetilde{\Theta}) + (1 + \lambda) \|\beta - \widetilde{\beta}\| \Big).$$

*for any $\beta, \widetilde{\beta} \in \{\beta : \|\beta\| \leq r\}$ and $t = 0, \Delta t, \ldots, (L-1)\Delta t$. Here, the constant $C_G$ only depends on $N, D, r$ and the parameters of the assumptions and $d(\Theta, \widetilde{\Theta})$ is defined as $\sum_t \sum_{j=1}^{M} \|\beta_{t,j} - \widetilde{\beta}_{t,j}\|$.*

**Lemma D.3** (Oracle gradient approximation with discretization). *Under Assumptions 1-3, suppose that the parameter setting $\Theta$ is i.i.d. drawn from $\{\rho(\beta|t)\}_{t/\Delta t+1 \in [L], j \in [M]}$ for some $\rho \in \mathcal{P}^2$ concentrated on $P_r$ and satisfies that $\int_\beta \rho(\beta, t)d\beta = 1$ for any $t \in [0, 1]$. Then with probability at least $1 - 4\exp(-\delta)$ with respect to the parameter initialization $\Theta^{(0)}$, we have*

$$\left\| \widehat{G}(\beta, \Theta, t) - G(\beta, \rho, t) \right\| \lesssim L^{-1} + \sqrt{\frac{\delta + \log(L+1)}{M}},$$

$$\|G(\beta, \hat{\rho}, t) - G(\beta, \rho, t)\| \lesssim L^{-1} + \sqrt{\frac{\delta + \log(L+1)}{M}},$$

$$\left\| \widehat{G}(\beta, \Theta, t) - G(\beta, \hat{\rho}, t) \right\| \lesssim L^{-1}$$

*for any $\beta \in \{\beta : \|\beta\| \leq r\}$, $t = 0, \Delta t, \ldots, (L-1)\Delta t, 1$ and any $\delta > 0$. Here, $\lesssim$ hides the dependencies on $N, D, r$ and the parameters of the assumptions.*

*Proof of Theorem 3.1.* Our proof consists of several steps outlined below:

**Step I: Show the $W_2$ continuity of parameter (sample) distributions**

Our analysis commences with the bound for $0 < s_1 < s_2 < \tau$, we have

$$W_2(\hat{\rho}^{(s_1)}, \hat{\rho}^{(s_2)})^2 \leq \frac{1}{ML} \sum_t \sum_{j=1}^{M} |\beta_{t,j}^{(s_1)} - \beta_{t,j}^{(s_2)}|^2 \leq \frac{(s_2 - s_1)}{ML} \sum_t \sum_{j=1}^{M} \int_{s_1}^{s_2} \|\widehat{G}(\beta_{t,j}^{(s)}, \Theta^{(s)}, t)\|^2 ds,$$

where each particle at time $s_1$ is paired with its position at time $s_2$, leveraging the Jensen's inequality. Recalling the identity

$$\frac{d\widehat{Q}(\Theta^{(s)})}{ds} = \frac{1}{ML} \sum_t \sum_{j=1}^{M} \|\widehat{G}(\beta_{t,j}^{(s)}, \Theta^{(s)}, t)\|^2$$

shown in Proposition D.1, it follows that

$$W_2(\hat{\rho}^{(s_1)}, \hat{\rho}^{(s_2)}) \leq (s_2 - s_1)^{1/2} \widehat{Q}^{1/2}(\Theta^{(0)}) \leq \frac{\lambda}{2} A_0^2,$$

where the last inequality uses (D.30). Since $A_0^2 \lesssim 1 + \lambda^{-1}$, we see that $W_2(\hat{\rho}^{(s_1)}, \hat{\rho}^{(s_2)}) \leq C(1 + \lambda)(s_2 - s_1)^{1/2}$ for some constant $C$ dependent on the parameters listed in the result. Similarly, we have

$$W_2(\rho^{(s_1)}, \rho^{(s_2)})^2 \leq \mathbb{E} \left\| \widetilde{\beta}^{(s_2)} - \widetilde{\beta}^{(s_1)} \right\|^2 \leq (s_2 - s_1) \int_{s_1}^{s_2} \int_0^1 \int_\beta \left\| G(\beta, \rho^{(s)}, t) \right\|^2 d\beta dt ds \leq (s_2 - s_1) Q(\rho_0) \lesssim (s_2 - s_1) C(1 + \lambda)$$

where $\widetilde{\beta}^{(s_1)} \sim \rho^{(s_1)}(\beta, t)$ is embedded with its future position at $\widetilde{\beta}^{(s_2)}$. The last step is feasible by setting $C$ large enough, noticing that $Q(\rho_0) \leq \lambda A_0^2/2$. To summarize, we have

$$\max \left\{ W_2(\hat{\rho}^{(s_1)}, \hat{\rho}^{(s_2)}), W_2(\rho^{(s_1)}, \rho^{(s_2)}) \right\} \leq C(1 + \lambda)\sqrt{s_2 - s_1} \tag{D.3}$$

for some constant $C > 0$ dependent on the parameters listed in the result.

**Step II: Bound the difference between gradient flow dynamics and non-linear dynamics**

Next, we aim to bound $\Delta(s) := \sup_{s' \in [0,s]} \sup_{t,j} \|\beta_{t,j}^{(s')} - \widetilde{\beta}_{t,j}^{(s')}\|$ for any $s \in [0, \tau]$. Taking the difference of (D.1) and (D.2), we obtain that

$$
\begin{aligned}
\|\beta_{t,j}^{(\tau)} - \widetilde{\beta}_{t,j}^{(\tau)}\| &\leq \int_0^\tau \left\| \widehat{G}(\beta_{t,j}^{(s)}, \Theta^{(s)}, t) - G(\widetilde{\beta}_{t,j}^{(s)}, \rho^{(s)}, t) \right\| ds \\
&\leq \int_0^\tau \left\| \widehat{G}(\beta_{t,j}^{(s)}, \Theta^{(s)}, t) - \widehat{G}(\widetilde{\beta}_{t,j}^{(s)}, \widetilde{\Theta}^{(s)}, t) \right\| ds \\
&\quad + \int_0^\tau \left\| \widehat{G}(\widetilde{\beta}_{t,j}^{(s)}, \widetilde{\Theta}^{(s)}, t) - G(\widetilde{\beta}_{t,j}^{(s)}, \rho^{(s)}, t) \right\| ds \\
&\leq C_G \int_0^\tau \left( \frac{1}{ML} d(\Theta^{(s)}, \widetilde{\Theta}^{(s)}) + (1+\lambda)\|\beta^{(s)} - \widetilde{\beta}^{(s)}\| \right) ds + \int_0^\tau \left\| \widehat{G}(\widetilde{\beta}_{t,j}^{(s)}, \widetilde{\Theta}^{(s)}, t) - G(\widetilde{\beta}_{t,j}^{(s)}, \rho^{(s)}, t) \right\| ds \\
&\leq C_G \int_0^\tau \left( \frac{1}{ML} d(\Theta^{(s)}, \widetilde{\Theta}^{(s)}) + (1+\lambda)\|\beta^{(s)} - \widetilde{\beta}^{(s)}\| \right) ds + \sup_{s \in [0,\tau]} \left\| \widehat{G}(\widetilde{\beta}_{t,j}^{(s)}, \widetilde{\Theta}^{(s)}, t) - G(\widetilde{\beta}_{t,j}^{(s)}, \rho^{(s)}, t) \right\|
\end{aligned}
\tag{D.4}
$$

where the final inequality stems from Lemma D.2, employing a constant $C_G$ dependent on the parameters listed in the result. By taking the supremacy over $t, j$ in (D.4), and considering $\frac{1}{ML} d(\Theta^{(s)}, \widetilde{\Theta}^{(s)}) \leq \sup_{t,j} \|\beta_{t,j}^{(\tau)} - \widetilde{\beta}_{t,j}^{(\tau)}\|$ for any $s \geq 0$, we derive

$$
\sup_{t,j} \|\beta_{t,j}^{(\tau)} - \widetilde{\beta}_{t,j}^{(\tau)}\| \leq C_G(2+\lambda) \int_0^\tau \sup_{t,j} \|\beta^{(s)} - \widetilde{\beta}^{(s)}\| ds + \sup_{t,j} \sup_{s \in [0,\tau]} \left\| \widehat{G}(\widetilde{\beta}_{t,j}^{(s)}, \widetilde{\Theta}^{(s)}, t) - G(\widetilde{\beta}_{t,j}^{(s)}, \rho^{(s)}, t) \right\|
$$

Further supremacy taken over $s \in [0, \tau]$ yields:

$$
\Delta(\tau) \leq \widetilde{\Delta}(\tau) + C_G(2 + \lambda) \int_0^\tau \Delta(s) ds,
\tag{D.5}
$$

where we define $\widetilde{\Delta}(\tau) := \sup_{t,j} \sup_{s \in [0,\tau]} \left\| \widehat{G}(\widetilde{\beta}_{t,j}^{(s)}, \widetilde{\Theta}^{(s)}, t) - G(\widetilde{\beta}_{t,j}^{(s)}, \rho^{(s)}, t) \right\|$ for simplicity of notation. Apply the Grönwall's inequality to (D.5) yields

$$
\Delta(\tau) \leq \exp\left( C_G(2 + \lambda)\tau \right) \widetilde{\Delta}(\tau).
\tag{D.6}
$$

It remains to bound $\widetilde{\Delta}(\tau)$ to bound $\Delta(\tau)$. It's worth noting that by Lemmas D.1 and D.2, for any $s_1, s_2 \in [0, \tau]$ and $t, j$, we have

$$
\begin{aligned}
&\left| \left\| \widehat{G}(\widetilde{\beta}_{t,j}^{(s_2)}, \widetilde{\Theta}^{(s_2)}, t) - G(\widetilde{\beta}_{t,j}^{(s_2)}, \rho^{(s_2)}, t) \right\| - \left\| \widehat{G}(\widetilde{\beta}_{t,j}^{(s_1)}, \widetilde{\Theta}^{(s_1)}, t) - G(\widetilde{\beta}_{t,j}^{(s_1)}, \rho^{(s_1)}, t) \right\| \right| \\
&\leq \left\| \widehat{G}(\widetilde{\beta}_{t,j}^{(s_1)}, \widetilde{\Theta}^{(s_1)}, t) - \widehat{G}(\widetilde{\beta}_{t,j}^{(s_2)}, \widetilde{\Theta}^{(s_2)}, t) \right\| + \left\| G(\widetilde{\beta}_{t,j}^{(s_1)}, \rho^{(s_1)}, t) - G(\widetilde{\beta}_{t,j}^{(s_2)}, \rho^{(s_2)}, t) \right\| \\
&\leq C_\Delta(\exp(C_\Delta W_1(\rho^{(s_1)}, \rho^{(s_2)})) - 1) + C_\Delta \frac{1}{ML} d(\widetilde{\Theta}^{(s_1)}, \widetilde{\Theta}^{(s_2)}) + C_\Delta(1+\lambda)\|\widetilde{\beta}_{t,j}^{(s_1)} - \widetilde{\beta}_{t,j}^{(s_2)}\| \\
&\lesssim \exp(C_\Delta C \sqrt{s_2 - s_1}) - 1 + (1+\lambda) \sup_{t,j} \|\widetilde{\beta}_{t,j}^{(s_1)} - \widetilde{\beta}_{t,j}^{(s_2)}\| \\
&\lesssim \sqrt{s_2 - s_1} + \sup_{t,j} \|\widetilde{\beta}_{t,j}^{(s_1)} - \widetilde{\beta}_{t,j}^{(s_2)}\|
\end{aligned}
\tag{D.7}
$$

for some constant $C_\Delta$ dependent on the parameters listed in the result. Moreover, for any $t, j$, we have

$$
\|\widetilde{\beta}_{t,j}^{(s_1)} - \widetilde{\beta}_{t,j}^{(s_2)}\| \leq \int_{s_1}^{s_2} \|G(\widetilde{\beta}_{t,j}^{(s)}, \widetilde{\Theta}^{(s)}, t)\| \leq \sqrt{s_2 - s_1} \int_{s_1}^{s_2} \|G(\widetilde{\beta}_{t,j}^{(s)}, \widetilde{\Theta}^{(s)}, t)\|^2 \lesssim (1+\lambda^2)\sqrt{s_2 - s_1},
\tag{D.8}
$$

where the universal boundedness of $\|G(\widetilde{\beta}_{t,j}^{(s)}, \widetilde{\Theta}^{(s)}, t) - \lambda\widetilde{\beta}_{t,j}^{(s)}\|$ can be readily derived from the third result of Lemma C.3 alongside Assumption 2 (ii). Therefore, we obtain

$$
\left\| \widehat{G}(\widetilde{\beta}_{t,j}^{(s_2)}, \widetilde{\Theta}^{(s_2)}, t) - G(\widetilde{\beta}_{t,j}^{(s_2)}, \rho^{(s_2)}, t) \right\| - \left\| \widehat{G}(\widetilde{\beta}_{t,j}^{(s_1)}, \widetilde{\Theta}^{(s_1)}, t) - G(\widetilde{\beta}_{t,j}^{(s_1)}, \rho^{(s_1)}, t) \right\| \lesssim (1+\lambda+\lambda^2+\lambda^3)\sqrt{s_2 - s_1}
\tag{D.9}
$$

for any $s_1, s_2 \in [0, \tau]$.

Define $\tau_n = n\tau/L^2$ for $n = 1, 2, \ldots, L^2$. Leveraging Lemma D.3 and applying the union bound over $n \in [L^2]$ (updating the $\delta$ in the lemma to $\delta + 2\log L$), we obtain, with a probability of at least $1 - \exp(-\delta)$ with respect to the parameter initialization $\Theta^{(0)}$:

$$\sup_{s \in \{\tau_n\}_{n \in [L^2]}} \left\| \widehat{G}(\widetilde{\beta}_{t,j}^{(s)}, \widetilde{\Theta}^{(s)}, t) - G(\widetilde{\beta}_{t,j}^{(s)}, \rho^{(s)}, t) \right\| \lesssim (1 + \lambda + \lambda^2 + \lambda^3) L^{-1} + \sqrt{\frac{\delta + \log L + \log(L+1)}{M}}$$

$$\lesssim L^{-1} + \sqrt{\frac{\delta + \log(L+1)}{M}}$$

Furthermore, leveraging (D.9), we deduce

$$\sup_{t,j} \sup_{s \in [0,\tau]} \left\| \widehat{G}(\widetilde{\beta}_{t,j}^{(s)}, \widetilde{\Theta}^{(s)}, t) - G(\widetilde{\beta}_{t,j}^{(s)}, \rho^{(s)}, t) \right\| \lesssim \sup_{t,j} \sup_{s \in \{\tau_n\}_{n \in [L^2]}} \left\| \widehat{G}(\widetilde{\beta}_{t,j}^{(s)}, \widetilde{\Theta}^{(s)}, t) - G(\widetilde{\beta}_{t,j}^{(s)}, \rho^{(s)}, t) \right\|$$

$$+ \sup_{|s_1 - s_2| \leq \tau/L^2} (1 + \lambda + \lambda^2 + \lambda^3) \sqrt{s_2 - s_1}$$

$$\lesssim L^{-1} + \sqrt{\frac{\delta + \log(L+1)}{M}}.$$

Returning to (D.6), we establish that with a probability of at least $1 - \exp(-\delta)$ with respect to the parameter initialization $\Theta^{(0)}$,

$$\sup_{s \in [0,\tau]} \sup_{t,j} \|\beta_{t,j}^{(s)} - \widetilde{\beta}_{t,j}^{(s)}\| = \Delta(\tau) \lesssim L^{-1} + \sqrt{\frac{\delta + \log(L+1)}{M}}.$$

We denote the event where the above inequality holds as $E_1$, thus we have $\mathbb{P}(E_1) \geq 1 - \exp(-\delta)$.

**Step III: Prove the finite time results**

Now, we are poised to demonstrate the results in Theorem 3.1 that concern supremacy over $s \in [0, \tau]$. The verification of Lemmas C.4 reveals the existence of a universal constant $B_\tau := B \exp(K(1 + R_\tau + R_\tau^2))$ such that

$$\max\{\|T_{\rho^{(\tau)}}(H, t)\|_{2-\text{col}}, \|\widehat{T}_{\Theta^{(\tau)}}(H, t)\|_{2-\text{col}}, \|\widehat{T}_{\widetilde{\Theta}^{(\tau)}}(H, t)\|_{2-\text{col}}\} \leq B_\tau$$

for any $H$ and $t \in [0, 1]$.

Utilizing Lemma D.6 and applying the union bound over $n \in [L^2]$, we observe

$$\sup_{s \in \{\tau_n\}_{n \in [L^2]}} \|\widehat{T}_{\widetilde{\Theta}^{(s)}}(H, t) - T_{\rho^{(s)}}(H, t)\|_F \lesssim L^{-1} + \sqrt{\frac{\delta + \log(L+1)}{M}}.$$

Additionally, note that for any $s_1, s_2 \in [0, \tau]$, we derive from Lemmas C.2 and C.5 that

$$\left| \|\widehat{T}_{\widetilde{\Theta}^{(s_1)}}(H, t) - T_{\rho^{(s_1)}}(H, t)\|_F - \|\widehat{T}_{\widetilde{\Theta}^{(s_2)}}(H, t) - T_{\rho^{(s_2)}}(H, t)\|_F \right|$$

$$\leq \|\widehat{T}_{\widetilde{\Theta}^{(s_1)}}(H, t) - \widehat{T}_{\widetilde{\Theta}^{(s_2)}}(H, t)\|_F + \|T_{\rho^{(s_1)}}(H, t) - T_{\rho^{(s_2)}}(H, t)\|_F \tag{D.10}$$

$$\lesssim \sup_{t,j} \|\widetilde{\beta}_{t,j}^{(s_1)} - \widetilde{\beta}_{t,j}^{(s_2)}\| + W_1(\rho^{(s_1)}, \rho^{(s_2)})$$

$$\lesssim \sqrt{s_2 - s_1}$$

where the last inequality is derived utilizing (D.3) and (D.8). Consequently, we have

$$\sup_{s \in [0,\tau]} \|\widehat{T}_{\widetilde{\Theta}^{(s)}}(H, t) - T_{\rho^{(s)}}(H, t)\|_F \lesssim \sup_{s \in \{\tau_n\}_{n \in [L^2]}} \|\widehat{T}_{\widetilde{\Theta}^{(s)}}(H, t) - T_{\rho^{(s)}}(H, t)\|_F + \sup_{|s_2 - s_1| \leq \tau/L^2} \sqrt{s_2 - s_1}$$

$$\lesssim L^{-1} + \sqrt{\frac{\delta + \log(L+1)}{M}}$$

(D.11)

with probability at by $1 - \exp(-\delta)$. We denote the event where the above inequality holds as $E_2$, thus we have $\mathbb{P}(E_2) \geq 1 - \exp(-\delta)$. Considering that $\{\widetilde{\beta}_{t,j}^{(s)}\}_{t,j}$ could be regarded as i.i.d. samples

drawn from $\{\rho^{(s)}\}_{t,j}$, employing a similar method with the concentration guarantee from Lemma C.7, we can readily deduce the existence of an event $E_3$ with $\mathbb{P}(E_3) \geq 1 - \exp(-\delta)$ such that, under $E_3$, we have

$$\sup_{s \in [0,\tau]} |\frac{1}{ML} \sum_t \sum_{j=1}^M \|\widetilde{\beta}_{t,j}^{(s)}\|^2 - \int_0^1 \int_\beta \|\beta\|^2 \rho^{(s)}(\beta, t) d\beta dt| \lesssim L^{-1} + \sqrt{\frac{\delta + \log(L+1)}{M}}.$$

Now, let's analyze the scenario under the probability event $E_1 \cap E_2 \cap E_3$ with $\mathbb{P}(E_1 \cap E_2 \cap E_3) \geq 1 - 3\exp(-\delta)$. Lemma C.5 demonstrates that

$$\sup_{s \in [0,\tau]} |\mathrm{Read}[\widehat{T}_{\Theta^{(s)}}(H, t)] - \mathrm{Read}[\widehat{T}_{\widetilde{\Theta}^{(s)}}(H, t)]| \leq \sup_{s \in [0,\tau]} \|\widehat{T}_{\Theta^{(s)}}(H, t) - \widehat{T}_{\widetilde{\Theta}^{(s)}}(H, t)\|_F$$

$$\lesssim \sup_{s \in [0,\tau]} \sup_{t,j} \|\beta_{t,j}^{(s)} - \widetilde{\beta}_{t,j}^{(s)}\| \lesssim L^{-1} + \sqrt{\frac{\delta + \log(L+1)}{M}}. \tag{D.12}$$

This further implies, by (D.11), that

$$\sup_{s \in [0,\tau]} |\mathrm{Read}[\widehat{T}_{\Theta^{(s)}}(H, t)] - \mathrm{Read}[T_{\rho^{(s)}}(H, t)]|$$

$$\leq \sup_{s \in [0,\tau]} |\mathrm{Read}[\widehat{T}_{\Theta^{(s)}}(H, t)] - \mathrm{Read}[\widehat{T}_{\widetilde{\Theta}^{(s)}}(H, t)]| + \sup_{s \in [0,\tau]} \|\widehat{T}_{\widetilde{\Theta}^{(s)}}(H, t) - T_{\rho^{(s)}}(H, t)\|_F \tag{D.13}$$

$$\lesssim L^{-1} + \sqrt{\frac{\delta + \log(L+1)}{M}},$$

Since $\max\{\|T_{\rho^{(\tau)}}(H, t)\|_{2-\mathrm{col}}, \|\widehat{T}_{\Theta^{(\tau)}}(H, t)\|_{2-\mathrm{col}}, \|\widehat{T}_{\widetilde{\Theta}^{(\tau)}}(H, t)\|_{2-\mathrm{col}}\}$ is universally bounded, (D.13) immediately indicates that

$$\sup_{s \in [0,\tau]} |\widehat{R}(\Theta^{(s)}) - R(\rho^{(s)})| \lesssim \sup_{s \in [0,\tau]} |\mathrm{Read}[\widehat{T}_{\Theta^{(s)}}(H, t)] - \mathrm{Read}[T_{\rho^{(s)}}(H, t)]| \lesssim L^{-1} + \sqrt{\frac{\delta + \log(L+1)}{M}},$$
$$\tag{D.14}$$

and

$$\sup_{s \in [0,\tau]} |\widehat{Q}(\Theta^{(s)}) - Q(\rho^{(s)})| \leq \sup_{s \in [0,\tau]} |\widehat{R}(\Theta^{(s)}) - R(\rho^{(s)})| + \sup_{s \in [0,\tau]} |\frac{1}{ML} \sum_t \sum_{j=1}^M \|\widetilde{\beta}_{t,j}^{(s)}\|^2 - \int_0^1 \int_\beta \|\beta\|^2 \rho^{(s)}(\beta, t) d\beta dt|$$

$$\lesssim L^{-1} + \sqrt{\frac{\delta + \log(L+1)}{M}}. \tag{D.15}$$

**Step IV: Prove the weakly convergence**

For the remainder of the proof, we adopt a similar approach as in the proof of Theorem 2.6 in [16]. We denote $\hat{\rho}$ as $\hat{\rho}_{M,L}$ for any given $M$ and $L$. It's essential to note that we treat $\hat{\rho}_{M,L}$ as probability measures in this step of the proof.

Recalling (D.3), for any $s_1, s_2 \in [0,\tau]$, we have

$$W_2(\hat{\rho}_{M,L}^{(s_1)}, \hat{\rho}_{M,L}^{(s_2)}) \leq C(1+\lambda)\sqrt{s_2 - s_1}$$

for some constant $C$ dependent on the parameters listed in the result. We observe that the family of curves $(s \mapsto \hat{\rho}_{M,L}^{(s)})_{M,L}$ is equicontinuous in $W_2$ on $[0,\tau]$, uniformly in $M, L$. Additionally, the family $(\hat{\rho}_{M,L})_{M,L}$ lies within a $W_2$ ball, thus weakly precompact. As the weak topology is weaker than the topology induced by $W_2$, according to the Arzelà–Ascoli theorem, along any sequence where $L \to \infty$ and $\log L/M \to \infty$, we can identify a subsequence that converges weakly to a certain process $(\nu^{(s)})_{s \geq 0} \in \mathcal{P}^2 \times \mathbb{R}$, concentrated on $P_{R_\tau}$ at all times. In the subsequent analysis, we solely focus on this subsequence, still denoted as $(\hat{\rho}_{M,L})_{M,L}$.

For any $t \in [0,1]$, let's define the sequence $(E_{M,L}^\top)_{M,L}$ of momentum fields, which is a vector-valued measure on $[0,\tau] \times \Omega$, denoted by $E_{M,L} := \widehat{G}(\beta, \Theta_{M,L}^{(s)}, t)\hat{\rho}^{(s)}(\beta, t)ds$. We also define $E := G(\beta, \nu^{(s)}, t)\nu^{(s)}(\beta, t)ds$.

Considering that both $\hat{\rho}_{M,L}$ and $\nu$ are concentrated on $P_{R_\tau}$, we also have uniform convergence in the Bounded Lipschitz metric. Hence, for any bounded and Lipschitz function $\varphi : [0,\tau] \times \mathbb{R}^{\dim\beta} \to \mathbb{R}^{\dim\beta}$, it holds

$$\|\hat{\rho}^{(s)} - \nu^s\|_{\mathrm{BL}} \to 0$$

uniformly among $s \in [0,\tau]$ along the sequence.

Note that

$$
\left| \int_0^\tau \int_0^1 \int_\beta \varphi \cdot d(E_{M,L} - E) \right| \leq \|\varphi\|_{\max} \int_0^\tau \int_0^1 \int_\beta \left\| \widehat{G}(\beta, \Theta_{M,L}^{(s)}, t) - G(\beta, \nu^{(s)}, t) \right\| \hat{\rho}^{(s)}(\beta, t) d\beta dt ds
$$

$$
+ \left| \int_0^\tau \int_0^1 \int_\beta \varphi \cdot (\hat{\rho}^{(s)} - \nu^s)(\beta, t) d\beta dt ds \right|
$$

$$
\lesssim \|\varphi\|_{\max} \int_0^\tau \int_0^1 \int_\beta \left\| \widehat{G}(\beta, \Theta_{M,L}^{(s)}, t) - G(\beta, \hat{\rho}_{M,L}^{(s)}, t) \right\| \hat{\rho}^{(s)}(\beta, t) d\beta dt ds
$$

$$
+ \|\varphi\|_{\max} \int_0^\tau \int_0^1 \int_\beta \left\| G(\beta, \hat{\rho}^{(s)}, t) - G(\beta, \nu^{(s)}, t) \right\| \hat{\rho}^{(s)}(\beta, t) d\beta dt ds
$$

$$
+ \sup_{s \in [0,\tau]} \|\hat{\rho}^{(s)} - \nu^s\|_{\mathrm{BL}}
$$

$$
\lesssim L^{-1} + \sup_{s \in [0,\tau]} \|\hat{\rho}^{(s)} - \nu^s\|_{\mathrm{BL}} + \sup_{s \in [0,\tau], \|\beta\| \leq R_\tau} \left\| G(\beta, \hat{\rho}^{(s)}, t) - G(\beta, \nu^{(s)}, t) \right\|
$$

$$
\lesssim L^{-1} + \sqrt{\frac{\delta + \log(L+1)}{M}} + \sup_{s \in [0,\tau]} \|\hat{\rho}^{(s)} - \nu^s\|_{\mathrm{BL}}
$$

(D.16)

for some constant $C_E$ dependent on the parameters listed in the result and with probability at least $1 - \exp(-\delta)$ with respect to the parameter initialization $\Theta^{(0)}$ for any $\delta > 0$. Here, the third inequality of (D.16) utilizes the third result of Lemma D.3, and the fourth inequality uses the second result of Lemma D.3, following a similar process in Step II to achieve supremacy over $s \in [0,\tau]$.

From (D.16), we infer that $\left| \int_0^\tau \int_0^1 \int_\beta \varphi \cdot d(E_{M,L} - E) \right| \to 0$ almost surely along the sequence. Hence, $E_{M,L}$ converges weakly to $E$ almost surely along the sequence, and the particle gradient flow for $(\nu^{(\tau)})_{\tau \geq 0}$ almost surely satisfies (2.7) on $[0,\tau]$ for any arbitrarily given $\tau > 0$. According to the Fokker-Planck equation without noise involved [59], we conclude that $(\nu^{(\tau)})_{\tau \geq 0}$ almost surely satisfies (3.5). Consequently, the uniqueness stated in Proposition 3.2 ensures that $(\nu^{(\tau)})_{\tau \geq 0} = (\rho^{(\tau)})_{\tau \geq 0}$ almost surely. $\qquad\square$

## D.2   Proof of Proposition 3.1

*Proof.* Suppose that the Fréchet derivative $\frac{\delta R}{\delta \rho}$ indeed exists, we establish

$$\frac{\delta Q}{\delta \rho}(\theta, w, t) = \frac{\delta R}{\delta \rho}(\theta, w, t) + \frac{\lambda}{2}(\|\theta\|_2^2 + \|w\|_2^2).$$

Therefore, it suffices to show that the Fréchet derivative of $L$ with respect to $\rho$ is

$$\frac{\delta R}{\delta \rho}(\beta, t) = \mathbb{E}_\mu \left[ \mathrm{Tr}\Big( g(T_\rho(H,t), \beta)^\top p_\rho(H,t) \Big) \right]. \tag{D.17}$$

Denote $\rho_\eta = \rho + \eta(\nu - \rho)$. We provide the following lemma to bound $T_{\rho_\eta}(H,1) - T_\rho(H,1)$ by expanding the first-order derivative as follows

**Lemma D.4** (First-order derivative of Transformer output). *Under Assumption 2, for any $H$ and $\rho, \nu \in \mathcal{P}^2$ that have bounded supports, we have*

$$
\mathrm{vec}[T_{\rho_\eta}(H,1) - T_\rho(H,1)] = \eta \int_0^1 \int_\beta \exp\Big( \int_t^1 \int_\beta \nabla_{\mathrm{vec}[T]} \mathrm{vec}[g(T_\rho(H,s), \beta)] \rho(\beta, s) d\beta \Big)
$$

$$
\cdot \mathrm{vec}[g(T_\rho(H,t), \beta)](\nu - \rho)(\beta, t) d\beta dt + o(\eta),
$$

(D.18)

*where $\rho_\eta := \rho + \eta(\nu - \rho)$.*

Given (D.18), we observe from the solution of $p_\rho$ (C.13) and (C.14) that

$$
\begin{aligned}
&\Big(\mathrm{Read}[T_\rho(H,1)] - y(H)\Big)\mathrm{Read}[T_{\rho_\eta}(H,1) - T_\rho(H,1)]\\
=&\,\mathrm{vec}[p_\rho(H,1)]^\top \mathrm{vec}[T_{\rho_\eta}(H,1) - T_\rho(H,1)]\\
=&\,\eta \int_0^1 \int_\beta \mathrm{vec}[p_\rho(H,t)]^\top \mathrm{vec}[g(T_\rho(H,t),\beta)](\rho - \nu)(\beta,t)d\beta dt + o(\eta)\qquad\text{(D.19)}\\
=&\,\eta \int_0^1 \int_\beta \mathrm{Tr}\Big(g(T_\rho(H,t),\beta)^\top p_\rho(H,t)\Big)(\rho - \nu)(\beta,t)d\beta dt + o(\eta),
\end{aligned}
$$

Hence, by applying (D.19) to the risk function, we obtain

$$
\begin{aligned}
R(\rho_\eta) - R(\rho) &= \frac{1}{2}\mathbb{E}_\mu\Big[\Big(\mathrm{Read}[T_{\rho_\eta}(H,1)] - y(H)\Big)^2 - \Big(\mathrm{Read}[T_\rho(H,1)] - y(H)\Big)^2\Big]\\
&= \mathbb{E}_\mu\Big[\Big(\mathrm{Read}[T_\rho(H,1)] - y(H)\Big)\mathrm{Read}[T_\rho(H,1) - T_{\rho_\eta}(H,1)]\Big]\\
&\quad + \mathrm{Read}[T_\rho(H,1) - T_{\rho_\eta}(H,1)]O(\mathrm{Read}[T_{\rho_\eta}(H,1) - T_\rho(H,1)])\\
&= \eta\Big\langle \frac{\delta R}{\delta \rho}, \nu - \rho\Big\rangle + o(\eta),
\end{aligned}
$$

which indicates (D.17) and concludes the proof. $\qquad\square$

### D.3 Proof of well-posedness of Wasserstein gradient flow

*Proof.* Following a similar idea as Proposition 2.5 of [16], we leverage the general theory of Wasserstein gradient flow developed in [4]. Define the functional family $Q_r(\rho)$ as

$$
Q_r(\rho) = \begin{cases} Q(\rho) & \rho(P_r) = 1,\\ \infty & \text{otherwise.} \end{cases}
$$

For any $r > 0$, let's consider any *admissible transport* $\gamma \in \mathcal{P}^{\Omega\times\Omega}$ concentrated on $P_r$. By definition, both of its marginals, denoted by $\rho_1$ and $\rho_2$, are concentrated on $P_r$. We define the transport cost for $\gamma$ as

$$
C_p(\gamma) := \Big(\int |x - y|^p d\gamma(x,y)\Big)^{1/p}
$$

for $p \geq 1$. Additionally, we denote the transport interpolation as $\rho_\alpha^\gamma := ((1-\alpha)\rho_1 + \alpha\rho_2)_{\#\gamma}$. Our proof consists of several steps outlined below.

**Step I: Show that $Q_r$ is proper and continuous for $W_2$ on its closed domain:**

Note that the parameters $r, D, N, \lambda$ remain fixed throughout this proof step, so we hide the constant dependencies on them. For any $(\beta, t) \in P_r$, we have $Q_r(\delta_{(\beta,t)}) = \frac{1}{2}\mathbb{E}_\mu[(\mathrm{Read}[H + \Delta t g(H,\beta)] - y(H))^2] + \frac{\lambda}{2}\|\beta\|^2 < \infty$. This indicates that $Q_r$ is proper. Moreover, for any $\rho, \nu \in \mathcal{P}^2$ whose bounded support belong to $P_r$, Lemma C.1 ensures that

$$
\|T_\rho(H,t) + T_\nu(H,t)\|_F = O(1),
$$

and Lemma C.2 guarantees that

$$
\|T_\rho(H,t) - T_\nu(H,t)\|_F = O(W_1(\rho,\nu)) = O(W_2(\rho,\nu))
$$

for any $H$. Therefore, we have

$$
\begin{aligned}
R(\nu) - R(\rho) &= \frac{1}{2}\mathbb{E}_\mu\Big[\Big(\mathrm{Read}[T_\nu(H,1)] - y(H)\Big)^2 - \Big(\mathrm{Read}[T_\rho(H,1)] - y(H)\Big)^2\Big]\\
&= \mathbb{E}_\mu\Big[\Big(\mathrm{Read}[T_\rho(H,1)] - y(H)\Big)\mathrm{Read}[T_\rho(H,1) - T_\nu(H,1)]\Big]\qquad\text{(D.20)}\\
&\quad + \mathrm{Read}[T_\rho(H,1) - T_\nu(H,1)]O(\mathrm{Read}[T_\nu(H,1) - T_\rho(H,1)])\\
&= O(W_2(\rho,\nu)).
\end{aligned}
$$

Furthermore, since both $\rho$ and $\nu$ have bounded support, $\|\beta\|^2$ is Lipschitz continuous with respect to $(\beta, t)$. Therefore, by the Kantorovich-Rubinstein Theorem (see Theorem 5.10 of [68], for example), we have

$$\left| \frac{\lambda}{2} \int_\beta \|\beta\|^2 (\rho - \nu) d\beta dt \right| = O(W_1(\rho, \nu)) = O(W_2(\rho, \nu)). \tag{D.21}$$

Combining (D.20) and (D.21), we obtain that $Q(\rho) - Q(\nu) = O(W_2(\rho, \nu))$. Therefore, $Q_r$ is continuous for $W_2$ on its closed domain.

**Step II: Show that $\alpha \mapsto Q(\rho_\alpha^\gamma)/C_2^2(\gamma)$ is differentiable and has a Lipschitz continuous derivative**

Let's denote $h(\alpha) := Q_r(\rho_\alpha^\gamma)$. Lemma C.3 ensures that for any $\rho \in \mathcal{P}^2$ with bounded support belonging to $P_r$, we have bounded $\|\frac{\delta Q}{\delta \rho}(\cdot, \cdot)\|_F$ on $P_r$. Therefore, $Q_r(\rho_\alpha^\gamma)$ is differentiable with respect to $t$, and the derivative reads

$$h'(\alpha) = \langle \frac{\delta Q}{\delta \rho} \Big|_{\rho = \rho_\alpha^\gamma}, \frac{d}{d\alpha} \rho_\alpha^\gamma \rangle$$
$$= \int d \frac{\delta Q}{\delta \rho} \Big|_{\rho = \rho_\alpha^\gamma} \Big( (1 - \alpha)(\beta_1, t_1) + \alpha(\beta_2, t_2) \Big) \Big[ (\beta_1, t_1) - (\beta_2, t_2) \Big] d\gamma((\beta_1, t_1), (\beta_2, t_2)) \Big\}. \tag{D.22}$$

Then, it suffices to show that $h'(\alpha)$ is Lipschitz continuous. To accomplish this, we first propose the following lemma for later use:

**Lemma D.5** (Locally Lipschitz of $\rho$ for the gradient). *Under Assumptions 1 and 2, for any $\rho, \nu \in \mathcal{P}^2$ concentrated on $P_r$, there exists some constant $L_r$ depending on $r, N, D$ and parameters of the assumptions such that*

$$\sup_{t \in [0,1]} \|p_\rho(H, t) - p_\nu(H, t)\|_F \leq L_r \|\rho - \nu\|_1,$$

$$\sup_{(\beta, t) \in P_r} \left| \frac{\delta Q}{\delta \rho} \Big|_\rho (\beta, t) - \frac{\delta Q}{\delta \rho} \Big|_\nu (\beta, t) \right| \leq L_r \|\rho - \nu\|_1.$$

Returning to the lemma proof, for $\alpha_1, \alpha_2 \in [0, 1]$, by the triangle inequality we have $|h'(\alpha_1) - h'(\alpha_2)| \leq J_1 + J_2$ where

$$J_1 := \left| \int d \frac{\delta Q}{\delta \rho} \Big|_{\rho = \rho_{\alpha_1}^\gamma} \Big( (1 - \alpha_1)(\beta_1, t_1) + \alpha_1(\beta_2, t_2) \Big) \Big[ (\beta_1, t_1) - (\beta_2, t_2) \Big] d\gamma((\beta_1, t_1), (\beta_2, t_2)) \right.$$
$$\left. - \int d \frac{\delta Q}{\delta \rho} \Big|_{\rho = \rho_{\alpha_2}^\gamma} \Big( (1 - \alpha_1)(\beta_1, t_1) + \alpha_1(\beta_2, t_2) \Big) \Big[ (\beta_1, t_1) - (\beta_2, t_2) \Big] d\gamma((\beta_1, t_1), (\beta_2, t_2)) \right|$$
$$\leq \sup_{(\beta, t) \in P_r} \left| \frac{\delta Q}{\delta \rho} \Big|_\rho (\beta, t) - \frac{\delta Q}{\delta \rho} \Big|_\nu (\beta, t) \right| \int \|(\beta_1, t_1) - (\beta_2, t_2)\|_1 d\gamma((\beta_1, t_1), (\beta_2, t_2))$$
$$\leq L_r \|\rho_{\alpha_1}^\gamma - \rho_{\alpha_2}^\gamma\|_1 \cdot C_1(\gamma)$$
$$\leq L_r C_1^2(\gamma) |\alpha_1 - \alpha_2|$$
$$\leq L_r C_2^2(\gamma) |\alpha_1 - \alpha_2|$$
$$\tag{D.23}$$

by Lemma D.5. The final inequality of (D.23) applies Hölder's inequality to obtain that $C_1^2(\gamma) \leq C_2^2(\gamma)$. Furthermore,

$$J_2 := \left| \int d \frac{\delta Q}{\delta \rho} \Big|_{\rho = \rho_{\alpha_2}^\gamma} \Big\{ \Big[ (1 - \alpha_1)(\beta_1, t_1) + \alpha_1(\beta_2, t_2) \Big] \Big[ (\beta_1, t_1) - (\beta_2, t_2) \Big] d\gamma((\beta_1, t_1), (\beta_2, t_2)) \Big\} \right.$$
$$\left. - \int d \frac{\delta Q}{\delta \rho} \Big|_{\rho = \rho_{\alpha_2}^\gamma} \Big\{ \Big[ (1 - \alpha_2)(\beta_1, t_1) + \alpha_2(\beta_2, t_2) \Big] \Big[ (\beta_1, t_1) - (\beta_2, t_2) \Big] d\gamma((\beta_1, t_1), (\beta_2, t_2)) \Big\} \right|$$
$$\leq \sup_{(\beta, t) \in P_r} \left\| \frac{\delta Q}{\delta \rho} \Big|_{\rho = \rho_{\alpha_2}^\gamma} (\beta, t) \right\| |\alpha_1 - \alpha_2| \int \|(\beta_1, t_1) - (\beta_2, t_2)\|^2 d\gamma((\beta_1, t_1), (\beta_2, t_2))$$
$$\leq L_r' C_2^2(\gamma) |\alpha_1 - \alpha_2|$$
$$\tag{D.24}$$

where $L_r' := \sup_{(\beta,t)\in P_r} \|\frac{\delta Q}{\delta \rho}|_{\rho=\rho_{\alpha_2}^\gamma}(\beta,t)\| < \infty$ from Lemma C.3. Combining (D.23) and (D.24) leads us to the result that $h'(\alpha)/C_2^2(\gamma)$ is Lipschitz continuous.

**Step III: Show the well-posedness of Wasserstein gradient flow at some finite time**

We follow a similar approach to the proof of Proposition 2.5 in [16]. Since $h'(\alpha)$ is $\lambda_h \times C_2^2(\gamma)$-Lipschitz continuous with respect to $\alpha$ for some $\lambda_h$, the well-posedness of the Wasserstein gradient flow for $Q_r$ with the velocity field constrained on $P_r$ is a corollary of Theorem 11.2.2 of [4]. Specifically, there exists a unique curve $(\rho_r^{(\tau)})_{\tau \geq 0}$ continuous in $\mathcal{P}^2$ such that:

$$\frac{d\rho_r^{(\tau)}}{d\tau} = \operatorname{div}_\beta(\rho_r^{(\tau)} v_r^{(\tau)})$$

where

$$v_r^{(\tau)}(\beta,t) = \begin{cases} G(\beta, \rho_r^{(\tau)}, t), & (\beta,t) \in P_r, \\ 0, & \text{otherwise.} \end{cases}$$

for $\rho_r^{(\tau)}$-a.e. Given the initialization $\rho_0$ concentrated on $P_R$, for any $r > R$, the unique $\rho_r^{(\tau)}$ exhibits a first exit time denoted as

$$\tau_r := \inf\{\tau > 0 : \rho_r^{(\tau)}(P_r) < 1\}.$$

By defining this exit time, for any $\bar{r} > r$ and $\tau \in [0, \tau_r]$, we observe $v_r(\tau)(\beta,t) = G(\beta, \rho_r^{(\tau)}, t)$ and $v_{\bar{r}}(\tau)(\beta,t) = G(\beta, \rho_{\bar{r}}^{(\tau)}, t)$. Due to uniqueness, we infer $\rho_r^{(\tau)} = \rho_{\bar{r}}^{(\tau)}$ on $\tau \in [0, \tau_r]$. Considering $\rho_r^{(\tau)}$ as the solution to (3.5), we establish the existence and uniqueness of the Wasserstein gradient flow for $Q$ over $[0, \tau_r]$.

**Step IV: Show the well-posedness of Wasserstein gradient flow at all time**

To establish the Wasserstein gradient flow's definition for $\tau \geq 0$, it's necessary to demonstrate that $\lim_{r\to\infty} \tau_r = \infty$. For any $r > R$, according to the energy identity in Theorem 11.2.1 of [4], on $[0, t_r]$, we observe that $\tau \mapsto Q(\rho^{(\tau)})$ is non-increasing. Specifically, this represents

$$\begin{aligned}
\frac{dQ(\rho^{(\tau)})}{d\tau} &= \int_0^1 \int_\beta \Big\langle \frac{dQ}{d\rho}\Big|_{\rho=\rho^{(\tau)}}, \operatorname{div}_\beta(\rho^{(\tau)} G(\beta, \rho^{(\tau)}, t)) \Big\rangle \\
&= \int_0^1 \int_\beta \Big\langle G(\beta, \rho^{(\tau)}, t), \operatorname{div}_\beta(\rho^{(\tau)} G(\beta, \rho^{(\tau)}, t)) \Big\rangle d\beta dt \qquad \text{(D.25)} \\
&= \int_0^1 \int_\beta \rho^{(\tau)} \|G(\beta, \rho^{(\tau)}, t)\|_2^2 d\beta dt \leq 0.
\end{aligned}$$

Therefore, for any $\tau \in [0, \tau_r]$, utilizing Lemma C.1, we have

$$\begin{aligned}
Q(\rho^{(\tau)}) \leq Q(\rho_0) &= \mathbb{E}_\mu\Big[\frac{1}{2}\Big(\operatorname{Read}[T_{\rho_0}(H, 1)] - y(H)\Big)^2\Big] + \frac{\lambda}{2} \int_0^1 \int_\beta \|\beta\|^2 \rho_0(\beta, t) d\beta dt \\
&\leq \frac{1}{2}\mathbb{E}_\mu[(\|T_{\rho_0}(H, 1)\|_{2-\text{col}} + B)^2] + \frac{\lambda R^2}{2} \\
&\leq \mathbb{E}_\mu[(\|T_{\rho_0}(H, 1)\|_{2-\text{col}}^2 + B^2] + \frac{\lambda R^2}{2} \\
&\leq B^2 + B^2 \exp\Big(K(1 + R + R^2)\Big)^2 + \frac{\lambda R^2}{2}
\end{aligned} \qquad \text{(D.26)}$$

Thus, we have $\int_0^1 \int_\beta \|\beta\|^2 \rho^{(\tau)}(\beta, t) \le \frac{2}{\lambda} Q(\rho^{(\tau)}) \le R^2 + \lambda^{-1}\Big(2B^2 + 2B^2 \exp\Big(K(1 + R + R^2)\Big)\Big)^2\Big) = A_0^2$. According to Assumption (ii) and Lemma C.3, for any $(\beta, t) \in P_r$, we have

$$\|v_r^{(\tau)}(\beta, t) - \lambda\beta\| = \|G(\beta, \rho^{(\tau)}, t) - \lambda\beta\| \le \sum_{i=1}^{N+1} \Big\| \nabla_\beta \Big\{ g(T_\rho(H, t), \beta)_{:,i} \Big\} p_\rho(H, t)_{:,i} \Big\|$$

$$\le \sup_{i \in [N+1]} \|\nabla_\beta g(T_\rho(H, t), \beta)_{:,i}\| \sum_{i=1}^{N+1} \|p_\rho(H, t)_{:,i}\|$$

$$\le \sqrt{N+1} \sup_{i \in [N+1]} \|\nabla_\beta g(T_\rho(H, t), \beta)_{:,i}\| \|p_\rho(H, t)\|_F$$

$$\le \sqrt{N+1} \phi_P(\|T_\rho(H, t)\|_{2-\mathrm{col}}) \|p_\rho(H, t)\|_F (1 + \|\beta\|)$$

$$\le C_R(1 + \|\beta\|),$$

(D.27)

where $C_R := \sqrt{N+1} \phi_P(B \exp(K(1 + A_0 + A_0^2)))(B + B \exp(K(1 + A_0 + A_0^2))) \exp\Big(\phi_T(N, D, \sqrt{N+1}KB \exp(K(1 + A_0 + A_0^2))(1 + A_0 + A_0^2)\Big)$. Applying (D.27) to the gradient flow equation

$$\frac{d\beta^{(\tau)}}{d\tau} = -v_r^{(\tau)}(\beta, t), \quad \beta^{(0)} = \beta$$

for $\tau \ge 0$, we obtain $\frac{d\|\beta^{(\tau)}\|}{d\tau} = \frac{\langle -v_r^{(\tau)}(\beta,t), \|\beta^{(\tau)}\|\rangle}{\|\beta^{(\tau)}\|} \le \|v_r^{(\tau)}(\beta, t) - \lambda\beta\| \le C_R(1 + \|\beta^{(\tau)}\|)$. This indicates

$$\|\beta^{(\tau)}\| \le (\|\beta\| + 1) \exp(C_R\tau) - 1 \le (R+1) \exp(C_R\tau) - 1 \tag{D.28}$$

by the Grönwall's inequality. Therefore, for any $T > 0$, $\rho^{(T)}$ is concentrated on $P_{(R+1)\exp(C_RT)-1}$, implying that for $r > (R+1)\exp(C_RT)$, we have $\tau_r > T$. Hence, we conclude $\lim_{r \to \infty} \tau_r = \infty$, establishing the existence of a unique Wasserstein gradient flow from (3.5) over $\tau > 0$.

Eventually, we establish the three properties listed in Proposition 3.2 for $\rho^{(\tau)}$. By (3.5), for any $t \in [0, 1]$, we have

$$\int_\beta \rho^{(\tau)}(\beta, t) d\beta = \int_\beta \rho^{(0)}(\beta, t) d\beta + \int_0^\tau \Big( \int_\beta \mathrm{div}_\beta(\rho^{(s)} G^{(s)}(\beta, \rho^{(s)}, t)) d\beta \Big) ds$$

$$= 1 + \int_0^\tau 0 \cdot ds = 1$$

indicated by the Divergence Theorem as $\rho^{(s)}$ has bounded support. Next, for any $\tau \ge 0$, (D.28) shows that $\rho^{(\tau)}$ is concentrated on $P_{R_\tau}$. Moreover, (D.25) now holds for any $\tau > 0$, implying $\int_0^1 \int_\beta \|\beta\|^2 \rho^{(\tau)}(\beta, t) \le A_0^2$ for any $\tau \ge 0$. □

## D.4 Proof of well-posedness of gradient flow

**Proposition D.1** (Existence and uniqueness of gradient flow). *Under Assumptions 1-3, for any initialization of $\Theta^{(0)}$ i.i.d. drawn from $\{\rho_0(\theta, w|t)\}_{t,j}$, there exists a unique solution $(\Theta^{(\tau)})_{\tau \ge 0}$ for (2.7). Additionally, for any $\tau \ge 0$, we have*

    *i. $\Theta^{(\tau)}$ has a bounded support, meaning $\sup_{t,j}(\|\theta_{t,j}^{(\tau)}\|_2^2 + \|w_{t,j}^{(\tau)}\|_2^2) \le R_\tau$.*

    *ii. $\frac{1}{ML} \sum_t \sum_{j=1}^M (\|\theta_{t,j}^{(\tau)}\|_2^2 + \|w_{t,j}^{(\tau)}\|_2^2) \le A_0^2$.*

*Here, $R_\tau$ and $A_0$ are defined as in Proposition 3.2.*

*Proof.* The local Lipschitz continuity established in Lemma D.2 directly implies the continuity of $\widehat{G}_\beta(\beta_{t,j}, \Theta, t)$ with respect to $\Theta$. Since $\{ML \cdot \widehat{G}_\beta(\beta_{t,j}, \Theta, t)\}_{t/\Delta t+1 \in [L], j \in [M]}$ serves as the gradient of $\widehat{Q}(\Theta)$, it follows that $\widehat{Q}(\Theta)$ is continuously differentiable, indicating the local semiconvexity of $\widehat{Q}(\Theta)$. Specifically, for any $\Theta$, there exists some $\kappa > 0$ such that $\widehat{Q}(\Theta) + \kappa \sum_t \sum_{j=1}^M (\|\theta_{t,j}\|_2^2 +$

$\|w_{t,j}\|_2^2)$ is convex within a small neighborhood of $\Theta$. The existence and uniqueness of a gradient flow over the maximal interval $[0, \tau_{\max}]$ is a standard result (see Section 2.1 of [61]).

For any $\tau \in [0, \tau_{\max}]$, it holds that

$$\widehat{Q}(\Theta^{(0)}) \geq \widehat{Q}(\Theta^{(0)}) - \widehat{Q}(\Theta^{(\tau)}) = \int_0^\tau \sum_t \sum_{j=1}^M \langle \frac{d\widehat{Q}(\Theta)}{d\beta_{t,j}}\Big|_{\Theta=\Theta^{(\tau)}}, ML\frac{d\widehat{Q}(\Theta)}{d\beta_{t,j}}\Big|_{\Theta=\Theta^{(\tau)}} \rangle d\tau$$

$$= \int_0^\tau \frac{1}{ML} \sum_t \sum_{j=1}^M \|\widehat{G}_\beta(\beta_{t,j}^{(\tau)}, \Theta^{(\tau)}, t)\|^2 d\tau$$

$$\geq \frac{1}{ML\tau} \sum_t \sum_{j=1}^M \left( \int_0^\tau \|\widehat{G}_\beta(\beta_{t,j}^{(\tau)}, \Theta^{(\tau)}, t)\| d\tau \right)^2,$$

(D.29)

where the last inequality follows from Jensen's inequality. (D.29) establishes that $\widehat{Q}(\Theta^{(\tau)})$ is both upper and lower bounded, and $\Theta^{(\tau)}$ exhibits a bounded curve length over any the time interval $[0, \tau_{\max}]$. By compactness, if $\tau_{\max}$ is finite, then $\Theta^{(\tau_{\max})}$ exists and thus must exist beyond $\tau_{\max}$, which leads to contradiction. Therefore, $\tau_{\max} = \infty$, and the well-posedness of the gradient flow for $\tau \geq 0$ consequently follows. Additionally, (D.29) shows that for any $\tau \geq 0$,

$$\frac{1}{ML} \sum_t \sum_{j=1}^M \|\beta_{t,j}\|_2^2 \leq 2\lambda^{-1}\widehat{Q}(\Theta^{(\tau)}) \leq 2\lambda^{-1}\widehat{Q}(\Theta^{(0)}) = \lambda^{-1}\mathbb{E}_\mu\left[\left(\text{Read}[\widehat{T}_{\Theta^{(0)}}(H,1)] - y(H)\right)^2\right]$$

$$+ \frac{1}{ML} \sum_t \sum_{j=1}^M \|\beta_{t,j}^{(0)}\|_2^2$$

$$\leq \lambda^{-1}\mathbb{E}_\mu[(\|\widehat{T}_{\Theta^{(0)}}(H,1)\|_{2-\text{col}} + B)^2] + R^2$$

$$\leq 2\lambda^{-1}\mathbb{E}_\mu[(\|\widehat{T}_{\Theta^{(0)}}(H,1)\|_{2-\text{col}}^2 + B^2] + R^2$$

$$\leq R^2 + \lambda^{-1}\left(2B^2 + 2B^2\exp\left(K(1+R+R^2)\right)^2\right)$$

$$= A_0^2$$

(D.30)

The last inequality of (D.30) follows from Lemma C.4, thereby showing that $\frac{1}{ML}\sum_t \sum_{j=1}^M \|\beta_{t,j}\|_2^2 \leq A_0^2$ for any $\tau \geq 0$.

As the final part of our proof, we demonstrate that the norm of any entry of $\Theta$ is bounded at any given time $\tau \geq 0$. Note that

$$\|\widehat{G}(\beta, \Theta^{(\tau)}, t) - \lambda\beta\|$$

$$\leq \sum_{i=1}^{N+1} \left\|\nabla_\theta\{f(\widehat{T}_\Theta(H,t),\theta)_{:,i}\}\widehat{p}_\Theta(H,t)_{:,i}\right\|/2 + \left\|\nabla_w\{h(\widehat{T}_\Theta(H,t+\Delta/2),w)_{:,i}\}\widehat{p}_\Theta(H,t+\Delta t/2)_{:,i}\right\|/2$$

$$\leq \sup_{i\in[N+1]}(\|\nabla_\theta f(T_\rho(H,t),\theta)_{:,i}\|\sum_{i=1}^{N+1}\|p_\rho(H,t)_{:,i}\|/2 + \|\nabla_w h(T_\rho(H,t),w)_{:,i}\|\sum_{i=1}^{N+1}\|p_\rho(H,t+\Delta t/2)_{:,i}\|/2)$$

$$\leq \sqrt{N+1}\sup_{i\in[N+1]}\Big(\|\nabla_\theta f(T_\rho(H,t),\theta)_{:,i}\|\,\|p_\rho(H,t)\|_F/2$$

$$+ \|\nabla_w h(T_\rho(H,t+\Delta t/2),w)_{:,i}\|\,\|p_\rho(H,t+\Delta t/2)\|_F/2\Big)$$

$$\leq \sqrt{N+1}\max\{\phi_P(\|T_\rho(H,t)\|_{2-\text{col}}), \phi_P(\|T_\rho(H,t+\Delta t/2)\|_{2-\text{col}})\}$$

$$\max\{\|p_\rho(H,t)\|_F, \|p_\rho(H,t+\Delta t/2)\|_F\}(1+\|\beta\|)$$

$$\leq C_R(1+\|\beta\|),$$

(D.31)

Applying (D.31) to the gradient flow

$$\frac{d\beta_{t,j}^{(\tau)}}{d\tau} = -\widehat{G}(\beta_{t,j}^{(\tau)}, \Theta^{(\tau)}, t)$$

for $\tau \geq 0$, we have $\frac{d\|\beta_{t,j}^{(\tau)}\|}{d\tau} = \frac{\langle -\widehat{G}(\beta_{t,j}^{(\tau)}, \Theta^{(\tau)}, t), \|\beta^{(\tau)}\|\rangle}{\|\beta^{(\tau)}\|} \leq \|\widehat{G}(\beta_{t,j}^{(\tau)}, \Theta^{(\tau)}, t) - \lambda\beta\| \leq C_R(1 + \|\beta^{(\tau)}\|)$. This indicates

$$\|\beta^{(\tau)}\| \leq (\|\beta\| + 1)\exp(C_R\tau) - 1 \leq (R+1)\exp(C_R\tau) - 1 = R_\tau. \qquad \text{(D.32)}$$

$\square$

## D.5   Proof of Proposition 3.3

Before commencing the proof, we introduce two proxy Transformer procedures in addition to $\bar{T}_\Theta$ and $\widetilde{T}_\rho$. The first proxy, denoted as $\bar{T}_\Theta$, involves moving the layers with the encoder $h$ slightly forward by $\Delta t/2$ in the depth index. This adjustment results in a discrete Transformer with only $L$ layers, where each layer has a step size of $\Delta t$ and an encoder of $(f + h)/2$, represented by $g$. Specifically, $\bar{T}_\Theta$ can be written as

$$\begin{aligned}
\bar{T}_\Theta(H, t + \Delta t) =& \bar{T}_\Theta(H, t) + \Delta t M^{-1} \sum_{j=1}^{M} \left( f(\bar{T}_\Theta(H, t), \theta_{t,j}) + \sum_{j=1}^{M} h(\bar{T}_\Theta(H, t), w_{t,j}) \right) \\
=& \bar{T}_\Theta(H, t) + \Delta t M^{-1} \sum_{j=1}^{M} g(\bar{T}_\Theta(H, t), \beta_{t,j}).
\end{aligned} \qquad \text{(D.33)}$$

The second proxy, denoted as $\widetilde{T}_\rho$, extends the width to infinity by letting $M \to \infty$, effectively replacing the average with an integral:

$$\widetilde{T}_\rho(H, t + \Delta t) = \widetilde{T}_\rho(H, t) + \Delta t \int_\beta g(\widetilde{T}_\rho(H, t), \beta)\rho(\beta|t)d\beta. \qquad \text{(D.34)}$$

We let all four Transformers share the same initial state $\widehat{T}_\Theta(H, 0) = \bar{T}_\Theta(H, 0) = \widetilde{T}_\rho(H, 0) = T_\rho(H, 0) = H$.

We first present the following lemma, considering parameters i.i.d. drawn from some distribution $\rho \in \mathcal{P}^2$ with bounded support:

**Lemma D.6** (Oracle approximation of discretization). *Under Assumptions 1 and 2, suppose that the parameter setting $\Theta$ is i.i.d. drawn from $\{\rho(\theta, w|t)\}_{t,j}$ for some $\rho \in \mathcal{P}^2$ concentrated on $\{(\theta, w) : \|\theta\|^2 + \|w\|^2 \leq r^2\} \times [0, 1]$ and satisfies that $\int_{\theta,w} \rho(\theta, w, t)d(\theta, w) = 1$ for any $t \in [0, 1]$. Then with probability at least $1 - \exp(-\delta)$ with respect to the parameter initialization $\Theta^{(0)}$, we have*

$$\|\widehat{T}_\Theta(H, t) - T_\rho(H, t)\|_F \lesssim L^{-1} + \sqrt{\frac{\delta + \log(L+1)}{M}}.$$

*for any $H$, $t = 0, \Delta t, \ldots, (L-1)\Delta t, 1$ and any $\delta > 0$. Here, $\lesssim$ hides the dependencies on $N, D, r$ and the parameters of the assumptions.*

*Proof of Proposition 3.3.* Since $\rho \in \mathcal{P}^{2,r}$ has a bounded support for any $\rho$, there exists some $\rho^* \in \mathcal{P}^{2,r}$ such that $R(\rho^*) = \inf_{\rho \in \mathcal{P}^{2,r}} R(\rho)$. According to Lemma D.6, we can find a specific $\Theta$ such that

$$\|\widehat{T}_\Theta(H, t) - T_{\rho^*}(H, t)\|_F \leq C\left(L^{-1} + \sqrt{\frac{\log(L+1)}{M}}\right),$$

where $C$ depends on $N, D, r$, and the parameters of the assumptions. Moreover, from Lemma D.6, we ensure that each entry $\beta_{t,j}$ of $\Theta$ satisfies $\|\beta_{t,j}\| \leq r$. Verification of Lemmas C.1 and C.4 on $T_\rho$ and $\widehat{T}_\Theta$ respectively leads to their uniform boundedness, i.e., $\sup_t T_\rho(H, t) \lesssim 1$ and $\sup_t \widehat{T}_\Theta(H, t) \lesssim 1$. Therefore, we have

$$\begin{aligned}
|\widehat{R}(\Theta) - R(\rho^*)| \leq& \mathbb{E}_\mu[|\text{Read}[\widehat{T}_\Theta(H, 1) - T_\Theta(H, 1)]| \cdot |\text{Read}[\widehat{T}_\Theta(H, 1) + T_\Theta(H, 1)] + 2y(H)|] \\
\lesssim& L^{-1} + \sqrt{\frac{\log(L+1)}{M}}.
\end{aligned}$$

Here, $\lesssim$ hides the dependencies on $N, D, r$ and the parameters of the assumptions. The result then follows.

The proof for the energy functional $Q$ (and $\widehat{Q}$) follows a similar approach. There exists some $\rho^* \in \mathcal{P}^{2,r}$ such that $Q(\rho^*) = \inf_{\rho \in \mathcal{P}^{2,r}} Q(\rho)$. From Lemmas D.6 and C.7, we can find a specific $\Theta$ such that

$$\begin{cases} \|\widehat{T}_\Theta(H,t) - T_{\rho^*}(H,t)\|_F \leq C\left(L^{-1} + \sqrt{\frac{\log(L+1)}{M}}\right), \\ |\frac{1}{ML}\sum_t \sum_{j=1}^M \|\beta_{t,j}\|^2 - \int_0^1 \int_\beta \|\beta\|^2 \rho(\beta,t)d\beta dt| \leq C\left(L^{-1} + \sqrt{\frac{\log(L+1)}{M}}\right). \end{cases}$$

by setting $C$ large enough. Verification of Lemmas C.1 and C.4 on $T_\rho$ and $\widehat{T}_\Theta$ respectively leads to their uniform boundedness. Hence, we have

$$|\widehat{Q}(\Theta) - Q(\rho^*)| \leq |\widehat{R}(\Theta) - R(\rho^*)| + \lambda \frac{1}{ML}\sum_t \sum_{j=1}^M \|\beta_{t,j}\|^2 - \int_0^1 \int_\beta \|\beta\|^2 \rho(\beta,t)d\beta dt| \leq C(1+\lambda)\left(L^{-1} + \sqrt{\frac{\log(L+1)}{M}}\right).$$

The result thus follows. $\qquad\qquad\qquad\qquad\qquad\qquad\qquad\qquad\qquad\qquad\qquad\qquad\qquad$ $\square$

# E   Proofs of main results in Section 4

For simplicity, we assume that Assumption 4 holds with $(g,\alpha) = (f,\theta)$. The proof for the case of $(g,\alpha) = (h,w)$ is symmetric, involving a simple substitution of $f$ with $h$ and $\theta$ with $w$.

## E.1   Proofs of Theorem 4.1 and Corollary 4.1

Our proof of Theorem 4.1 consists of three parts, each focusing on bounding differences related to the energy functional $Q$ or $\widehat{Q}$.

The first step establishes the continuity of the functional gradient $\frac{\delta Q}{\delta \rho}|_{\rho_\infty}$. This ensures that if the derivative with respect to $\beta$ for the functional gradient is constant over a region, then the functional gradient remains constant within that region.

The second step provides the key bound for $Q(\rho_\infty)$, which is proportional to $\lambda$. This involves a detailed analysis of $Q$'s landscape by bounding its derivatives.

After obtaining the bound for $Q(\rho_\infty)$, the final steps are to show that the finite-time risk can approach this bound. Achieving a loss as small as $\epsilon$ requires $Q(\rho^{(\tau_0)}) \leq \epsilon$ for some sufficiently large $\tau_0$. We then apply Theorem 3.1, with constant dependency on $\tau_0$, to show that $\widehat{Q}(\tau_0)$ becomes sufficiently small. Since $\widehat{Q}(\rho^{(\tau)})$ is non-increasing, $\widehat{Q}(\tau)$ remains small for all $\tau \geq \tau_0$.

### Preparatory Step: Landscape analysis

First, the following lemma suggests that as long as the risk $R(\rho)$ remains positive, a descent direction for $Q(\rho)$ can be constructed at any depth index, provided that $\lambda$ is sufficiently small. This implies that by adjusting $\lambda$, one can influence the gradient flow to effectively reduce $Q(\rho)$.

**Lemma E.1** (Landscape of $Q(\rho)$). *Suppose that Assumptions 1-4 hold. For any $\rho \in \mathcal{P}^2$ concentrated on $P_r$ with some $r > 0$, any $w_0 \in \mathbb{R}^{\dim w}$, and any $t^* \in [0,1]$ such that $\int_\beta \rho(\beta, t^*) \geq 1/2$, there exists a $\nu \in \mathcal{P}(\mathbb{R}^{\dim \beta})$ such that*

    *i. for any $\beta \in \text{supp}(\nu)$, we have $1/B_r \leq \|\beta\| \leq B_r$*

    *ii. for any $(\theta_1, \theta_2, w)$, we have $\theta_2 \in \mathcal{K}$ and $w = w_0$.*

    *iii. $\int_\beta \frac{\delta Q}{\delta \rho}(\beta, t^*)\left(\nu(\beta) - \rho(\beta|t^*)\right)d\beta \leq C_1 \lambda - C_2 R(\rho)$*

*Here, $B_r, C_1, C_2$ are constants that depends on $N, d, r$ and the parameters of the assumptions.*

Given the $t^*$ and $w_0$ specified in the theorem, Lemma E.1 indicates that there exists some

$$\nu \in \mathcal{P}\left(\left(B_{\dim \theta_1}(0, B_{R_\infty})/B_{\dim \theta_1}(0, 1/B_{R_\infty})\right) \times \mathcal{K} \times \{w_0\}\right)$$

such that $\int_\beta \frac{\delta Q}{\delta \rho}|_{\rho_\infty}(\beta, t^*)\left(\nu(\beta) - \rho_\infty(\beta|t^*)\right)d\beta \leq C_1 \lambda - C_2 R(\rho)$, where $B_{R_\infty}, C_1, C_2$ are constants dependent on $N, d, R_\infty$ and the parameters of the assumptions.

In addition, for any $\rho \in \mathcal{P}^2$, we define the following two functional derivatives:

$$\frac{\delta Q_f}{\delta \rho}(\theta, w, t) = \mathbb{E}_\mu \Big[ \mathrm{Tr} \Big( \big[ f(T_\rho(H, t), \theta) \big]^\top p_\rho(H, t) \Big) \Big] + \lambda \|\theta\|_2^2$$

$$\frac{\delta Q_h}{\delta \rho}(\theta, w, t) = \mathbb{E}_\mu \Big[ \mathrm{Tr} \Big( \big[ h(T_\rho(H, t), w) \big]^\top p_\rho(H, t) \Big) \Big] + \lambda \|w\|_2^2.$$

It is obvious that $\frac{\delta Q}{\delta \rho} \equiv (\frac{\delta Q_f}{\delta \rho} + \frac{\delta Q_h}{\delta \rho})/2$.

*Proof of Theorem 4.1.* **Step I: Show that $\frac{\delta Q}{\delta \rho}|_{\rho_\infty}(\beta, t)$ is continuous with respect to $(\beta, t)$**

In Step I of the proof of Lemma E.1, we establish that $\rho_\infty \in \mathcal{P}^2$, with a bounded support $P_{R_\infty}$, implies $p_{\rho_\infty}(H, t)$ is $C_p$-Lipschitz continuous, where $C_p$ is a constant dependent solely on $N, D, R_\infty$, and the parameters in our assumptions.

Next, we would like to show that $\frac{\delta Q}{\delta \rho}_{\rho_\infty}(\beta, t)$ is continuous with respect to $(\beta, t) \in \mathbb{R}^{\mathrm{div}\beta} \times [0, 1]$. Let's focus on the region $(\beta, t) \in P_r$ for any $r > R_\infty$, so that $\rho_\infty$ is also concentrated on $P_r$. It's noteworthy that for any bounded support $(\beta, t) \in P_r$, Lemma C.1 and Assumption 2 (i) ensure the universal boundedness of $g(T_\rho(H, t), \beta)$, and Lemma C.3 ensures the universal boundedness of $p_\rho(H, t)$ for any $H \in \mathrm{supp}(\mu)$, with the constants depending solely on $N, D, r$, and the parameters of the assumptions.

Combining the Lipschitz continuity of $p_{\rho_\infty}(H, t)$ and $T_{\rho_\infty}(H, t)$ with respect to $(H, t)$, as shown in Proposition C.2, along with the Lipschitz continuity of $g(T, \beta)$ with respect to $(T, \beta)$ when $\|T\|_F$ is universally bounded (as guaranteed by Assumption 2 (ii) and (iii)), and their universal boundedness, we derive that $\mathrm{Tr}\Big( \big[ g(T_{\rho_\infty}(H, t), \beta) \big]^\top p_{\rho_\infty}(H, t) \Big)$ is $C_G$-Lipschitz continuous for $\|\cdot\|_2$ with respect to $(\beta, t) \in P_r$ for some constant $C_G$ that depends only on $N, D, r$ and the parameters of the assumptions. Since the Lipschitz constant $C_G$ is independent of the choice of $H$, we see that $\mathrm{Tr}\Big( \big[ g(T_{\rho_\infty}(H, t), \beta) \big]^\top p_{\rho_\infty}(H, t) \Big)$ is uniformly continuous across all $H \in \mathrm{supp}(\mu)$ with respect to $(\beta, t) \in P_r$. Consequently, we have that

$$\frac{\delta Q}{\delta \rho}(\beta, t)|_{\rho_\infty} = \mathbb{E}_\mu \Big[ \mathrm{Tr} \Big( \big[ g(T_{\rho_\infty}(H, t), \beta) \big]^\top p_{\rho_\infty}(H, t) \Big) \Big] + \frac{\lambda}{2} \|\beta\|_2^2$$

is continuous with respect to $(\beta, t) \in P_r$. Since the choice of $r$ is arbitrary, we conclude that $\frac{\delta Q}{\delta \rho}|_{\rho_\infty}(\beta, t)$ is continuous with respect to $(\beta, t) \in \mathbb{R}^{\mathrm{div}\beta} \times [0, 1]$.

**Step II: Show that $Q(\rho_\infty) \lesssim \lambda$ with further landscape analysis**

In the first part of the proof, we will adopt a similar approach to Theorem 3.9 of [47] to demonstrate that the stationary point of the Wasserstein gradient flow, denoted $\rho_\infty$, satisfies $Q(\rho_\infty) \lesssim \lambda$. It's worth noting that in [47], the authors assume $\lambda = 0$ and conclude $R(\rho_\infty) = Q(\rho_\infty) = 0$, but this claim relies on assuming the global existence of the Wasserstein gradient flow rather than proving it directly.

Based on the pivotal findings from [53] regarding the stationary points in the Wasserstein space, we infer that the stationary point $\rho_\infty$ of the Wasserstein gradient flow (3.5), i.e.

$$\frac{d\rho(\beta, t)}{d\tau} = \mathrm{div}_\beta \Big( \rho \nabla_\beta \frac{\delta Q}{\delta \rho} \Big),$$

must satisfy $\nabla_\beta \frac{\delta Q}{\delta \rho}|_{\rho_\infty} = 0$ almost everywhere over $\mathrm{supp}(\rho_\infty)$. This further indicates that $\nabla_\beta \frac{\delta Q}{\delta \rho}|_{\rho_\infty}(\theta_1, \theta_2, w, t^*) = 0$ almost everywhere over $\mathrm{supp}(\rho_\infty(\cdot, t^*))$. The fact $\rho(\cdot, t^*)$ is a connected set, coupled with the continuity of the Frechét differential $\frac{\delta Q}{\delta \rho}|_{\rho_\infty}$ with respect to $\beta$, implies that, $\frac{\delta Q}{\delta \rho}|_{\rho_\infty}(\theta_1, \theta_2, w, t^*) = C$ for some constant $C$ over $(\beta, t^*) \in \mathrm{supp}(\rho(\cdot, t^*))$.

Given the separation assumption on the support of $\rho_\infty(\cdot, t^*)$, we ensure that for any $(\theta_1, \theta_2) \in \Big( B_{\dim\theta_1}(0, B_{R_\infty})/B_{\dim\theta_1}(0, 1/B_{R_\infty}) \Big) \times \mathcal{K}$, there exists $c \in \mathbb{R}$, $1/R_\infty B_{R_\infty} \leq |c| \leq R_\infty B_{R_\infty}$

such that $(c\theta_1, \theta_2, w_0, t^*) \in \text{supp}(\rho_\infty)$. Combined with Assumption 4 (i), which implies the 1-homogeneity of $f(T, \theta_1, \theta_2)$ with respect to $\theta_1$, we have

$$
\begin{aligned}
\frac{\delta Q_f}{\delta \rho}\bigg|_{\rho_\infty} (c\theta_1, \theta_2, w_0, t^*) &= c\frac{\delta Q_f}{\delta \rho}\bigg|_{\rho_\infty} (\theta_1, \theta_2, w_0, t^*) + (|c|-1)\lambda\|\theta_1\|^2. \\
\frac{\delta Q_h}{\delta \rho}\bigg|_{\rho_\infty} (c\theta_1, \theta_2, w_0, t^*) &= \frac{\delta Q_h}{\delta \rho}\bigg|_{\rho_\infty} (\theta_1, \theta_2, w_0, t^*).
\end{aligned}
\tag{E.1}
$$

Hence, given that $\nabla_\beta \frac{\delta Q}{\delta \rho}|_{\rho_\infty}(\cdot, t^*) = 0$ almost everywhere over $\text{supp}(\rho_\infty(\cdot, t^*))$, it also holds that

$$
\begin{aligned}
\nabla_{(\theta_1, \theta_2, w)} \frac{\delta Q}{\delta \rho}\bigg|_{\rho_\infty} (\theta_1, \theta_2, w_0, t^*) &= \left(\nabla_{(\theta_1, \theta_2, w)} \frac{\delta Q_f}{\delta \rho}\bigg|_{\rho_\infty} (\theta_1, \theta_2, w_0, t^*) + \nabla_{(\theta_1, \theta_2, w)} \frac{\delta Q_h}{\delta \rho}\bigg|_{\rho_\infty} (\theta_1, \theta_2, w_0, t^*)\right)/2 \\
&= \left(-\frac{|c|-1}{c}\lambda\theta_1, 0_{\dim\theta_2}, 0_w\right),
\end{aligned}
$$

which implies

$$
\left\|\nabla_{(\theta_1, \theta_2, w)} \frac{\delta Q}{\delta \rho}\bigg|_{\rho_\infty} (\theta_1, \theta_2, w_0, t^*)\right\| \leq (R_\infty^2 B_{R_\infty} + R_\infty)\lambda.
\tag{E.2}
$$

Given the condition that $\|(c\theta_1, \theta_2, w_0, t^*) - (\theta_1, \theta_2, w_0, t^*)\| \leq R_\infty^2 B_{R_\infty}$, and recalling that $\frac{\delta Q}{\delta \rho}|_{\rho_\infty}(\theta_1, \theta_2, w, t^*) \equiv C$ across $(\beta, t^*) \in \text{supp}(\rho(\cdot, t^*))$, (E.2) further indicates that

$$
\left|\frac{\delta Q}{\delta \rho}\bigg|_{\rho_\infty} (\theta_1, \theta_2, w_0, t^*) - C\right| \leq (R_\infty^4 B_{R_\infty}^2 + R_\infty^3 B_{R_\infty})\lambda.
\tag{E.3}
$$

for any $(\theta_1, \theta_2) \in \left(B_{\dim\theta_1}(0, B_{R_\infty})/B_{\dim\theta_1}(0, 1/B_{R_\infty})\right) \times \mathcal{K}$. Hence, by Lemma E.1 we have

$$
\begin{aligned}
C_1\lambda - C_2 R(\rho_\infty) &\geq \int_\beta \frac{\delta Q}{\delta \rho}\bigg|_{\rho_\infty} (\beta, t^*)\left(\nu(\beta) - \rho_\infty(\beta|t^*)\right)d\beta \\
&= \int_\beta \left(\frac{\delta Q}{\delta \rho}\bigg|_{\rho_\infty} (\beta, t^*) - C\right)\left(\nu(\beta) - \rho_\infty(\beta|t^*)\right)d\beta \\
&\geq -\int_\beta (R_\infty^4 B_{R_\infty}^2 + R_\infty^3 B_{R_\infty})\lambda\left(\nu(\beta) + \rho_\infty(\beta|t^*)\right)d\beta \\
&\geq 2(R_\infty^4 B_{R_\infty}^2 + R_\infty^3 B_{R_\infty})\lambda.
\end{aligned}
\tag{E.4}
$$

Therefore, we have $R(\rho_\infty) \leq \frac{C_1 + 2(R_\infty^4 B_{R_\infty}^2 + R_\infty^3 B_{R_\infty})}{C_2}\lambda$, and

$$
Q(\rho_\infty) \leq R(\rho_\infty) + \int_0^1 \int_\beta \|\beta\|^2 \rho(\beta, t)d\beta dt \leq \left(\frac{C_1 + 2(R_\infty^4 B_{R_\infty}^2 + R_\infty^3 B_{R_\infty})}{C_2} + R_\infty^2\right)\lambda, \tag{E.5}
$$

which completes the first part of our proof.

**Step III: Bound the difference between $Q(\rho^{(\tau)})$ and $Q(\rho_\infty)$ when $\tau$ is large**

Proposition 3.2 establishes that the second moment for $\rho^{(\tau)}$ is uniformly bounded across all $\tau \geq 0$:

$$
\int_0^1 \int_\beta \|\beta\|^2 \rho^{(\tau)}(\beta, t)d\beta dt \leq A_0^2,
$$

where $A_0$ is defined as in Proposition 3.2. Therefore, the weak convergence of probability measures $(\rho^{(\tau)})_{\tau \geq 0}$ is equivalent to the convergence in the Wasserstein-2 distance, i.e.

$$
\lim_{\tau \to \infty} W_2(\rho^{(\tau)}, \rho_\infty) = 0.
\tag{E.6}
$$

When $\tau$ is sufficiently large, $\rho^{(\tau)}$ concentrates on $P_{R_\infty}$. Therefore, according to Lemma C.2, there exists a constant $C_0$ depending solely on $N$, $D$, $R_\infty$, and the parameters of the assumptions that

$$
\|T_{\rho_\infty}(H, t) - T_{\rho^{(\tau)}}(H, t)\|_F \leq C_0 W_2(\rho^{(\tau)}, \rho_\infty)
\tag{E.7}
$$

for any $H$ and $t \in [0, 1]$ when $\tau$ is sufficiently large. Note that Lemma C.1 shows that

$$\max\{\|T_{\rho^{(\tau)}}(H, t)\|_{2-\text{col}}, \|T_{\rho_\infty}(H, t)\|_{2-\text{col}}\} \leq B \exp(K(1 + R_\infty + R_\infty^2)) =: B_T$$

for any $H$ and $t \in [0, 1]$. Thus, from (E.7), we have

$$|Q(\rho_\infty) - Q(\rho^{(\tau)})| \leq \frac{1}{2} \mathbb{E}_\mu \Big[ \big| \text{Read}[|T_{\rho^{(\tau)}}(H, 1)|] + |T_{\rho_\infty}(H, 1)|] + 2|y(H)| \big| \Big] \|T_{\rho_\infty}(H, 1) - T_{\rho^{(\tau)}}(H, 1)\|_F$$

$$+ \frac{\lambda}{2} \int_0^1 \int_\beta \|\beta\|^2 (\rho_\infty - \rho^{(\tau)})(\beta, t) d\beta dt$$

$$\leq (B_T + B) C_0 W_2(\rho^{(\tau)}, \rho_\infty) + \frac{\lambda}{2} R_\infty W_1(\rho^{(\tau)}, \rho_\infty)$$

$$\leq ((B_T + B) C_0 + \frac{\lambda}{2} R_\infty) W_2(\rho^{(\tau)}, \rho_\infty),$$

(E.8)

where the second inequality incorporates the Kantorovich-Rubinstein Theorem (see Theorem 5.10 of [68], for example) and the $2R_\infty$-Lipschitz continuity of $\|\beta\|^2$ over the region $(\beta, t) \in P_{R_\infty}$. Combining equations (E.6) and (E.8), we deduce that for any $\epsilon > 0$, there exists some $\tau_0 > 0$ such that $|Q(\rho_\infty) - Q(\rho^{(\tau_0)})| \leq \epsilon$.

## Step IV: Complete the proof by bounding the difference between $\widehat{Q}(\Theta^{(\tau)})$ and $Q(\rho^{(\tau)})$ when $\tau$ is large

The final step can be seen as a direct corollary of the approximation result in Theorem 3.1. According to Theorem 3.1, there exists a constant $C_1$ dependent on $N, D, \tau_0, \lambda$, and the parameters specified in the assumptions, such that

$$|\widehat{Q}(\Theta^{(\tau_0)}) - Q(\rho^{(\tau_0)})| \leq C_1 \Big( L^{-1} + \sqrt{\frac{\delta + \log(L + 1)}{M}} \Big)$$

with probability at least $1 - 3\exp(-\delta)$ with respect to the parameter initialization $\Theta^{(0)}$ for any $\delta > 0$. Combining the outcomes from the preceding steps, we obtain

$$\widehat{Q}(\Theta^{(\tau_0)}) \leq \epsilon + C_1 \Big( L^{-1} + \sqrt{\frac{\delta + \log(L + 1)}{M}} \Big) + C_2 \lambda. \qquad \text{(E.9)}$$

Note that

$$\frac{d}{d\tau} \widehat{Q}(\Theta^{(\tau)}) = \sum_t \sum_{j=1}^M \langle \frac{d\widehat{Q}(\Theta)}{d\beta_{t,j}} \Big|_{\Theta=\Theta^{(\tau)}}, -ML \frac{d\widehat{Q}(\Theta)}{d\beta_{t,j}} \Big|_{\Theta=\Theta^{(\tau)}} \rangle = -\frac{1}{ML} \sum_t \sum_{j=1}^M \|\widehat{G}_\beta(\beta_{t,j}^{(\tau)}, \Theta^{(\tau)}, t)\|^2 \leq 0,$$

so the sequence $(\widehat{Q}(\Theta^{(\tau)}))_{\tau \geq 0}$ is non-decreasing. Hence, for any $\tau \geq \tau_0$,

$$\widehat{R}(\Theta^{(\tau)}) \leq \widehat{Q}(\Theta^{(\tau)}) \leq \widehat{Q}(\Theta^{(\tau_0)}) \leq \epsilon + C_1 \Big( L^{-1} + \sqrt{\frac{\delta + \log(L + 1)}{M}} \Big) + C_2 \lambda,$$

which completes the proof, recalling that $C_1$ depends only on $N, D, \tau_0, \lambda$ and the parameters of the assumptions, and $C_2$ depends only on $N, D, R_\infty$, and the parameters of the assumptions.

$\square$

*Proof of Corollary 4.1.* Given the choice of $\lambda$ By Theorem 4.1, there exists some $\tau_0 > 0$ such that

$$\sup_{\tau \geq \tau_0} \widehat{R}(\Theta^{(\tau)}) \leq \epsilon/2 + C_1 \Big( L^{-1} + \sqrt{\frac{\delta + \log(L + 1)}{M}} \Big) + C_2 C_\lambda \lambda$$

$$\leq (1/4 + C_2 C_\lambda) \epsilon + C_1 \Big( L^{-1} + \sqrt{\frac{\delta + \log(L + 1)}{M}} \Big).$$

The result holds by setting $L$ and $M/\log L$ sufficiently large to ensure that

$$C_1 \Big( L^{-1} + \sqrt{\frac{\delta + \log(L + 1)}{M}} \Big) \leq C_1 \Big( L^{-1} + \sqrt{\frac{2(1 + \delta) \log(L + 1)}{M}} \Big) \leq \epsilon/4.$$

$\square$

## F  Proofs of auxiliary results

### F.1  Proof of Lemma D.1

*Proof.* Lemma C.1 confirms that $\|T_\rho(H,t)\|_{2-\mathrm{col}}$ and $\|T_\nu(H,t)\|_{2-\mathrm{col}}$ are bounded uniformly by $B_T = B \exp(K(1 + r + r^2))$. Considering the definition (C.2), it suffices to demonstrate that

$$
\left\| \mathbb{E}_\mu \Big[ \nabla_\beta \mathrm{Tr}\Big( g(T_\rho(H,t),\beta)^\top p_\rho(H,t) \Big) - \nabla_\beta \mathrm{Tr}\Big( g(T_\nu(H,t),\widetilde{\beta})^\top p_\nu(H,t) \Big) \Big] \right\| \leq J_1 + J_2,
$$

where

$$
J_1 := \left\| \mathbb{E}_\mu \Big[ \nabla_\beta \mathrm{Tr}\Big( (g(T_\nu(H,t),\widetilde{\beta}) - g(T_\rho(H,t),\beta))^\top p_\nu(H,t) \Big) \Big] \right\| \lesssim \|\beta - \widetilde{\beta}\|,
$$

$$
J_2 := \left\| \mathbb{E}_\mu \Big[ \nabla_\beta \mathrm{Tr}\Big( g(T_\rho(H,t),\beta)^\top (p_\nu(H,t) - p_\rho(H,t)) \Big) \Big] \right\| \lesssim \exp(C_G W_1(\rho,\nu)) - 1.
$$

Here, the symbol $\lesssim$ hides dependencies on $N$, $D$, $r$, and the parameters of the assumptions. To bound $J_1$, consider that

$$
\begin{aligned}
J_1 &\leq \sup_{i \in [N+1]} \mathbb{E}_\mu \bigg[ \left\| g(T_\nu(H,t),\widetilde{\beta})_{:,i} - g(T_\rho(H,t),\beta)_{:,i} \right\| \sum_{i=1}^{N+1} \|p_\nu(H,t)_{:,i}\| \bigg] \\
&\leq \sqrt{N+1} \sup_{i \in [N+1]} \mathbb{E}_\mu \bigg[ \left\| g(T_\nu(H,t),\widetilde{\beta})_{:,i} - g(T_\rho(H,t),\beta)_{:,i} \right\| \|p_\nu(H,t)\|_F \bigg] \\
&\lesssim \sup_{i \in [N+1]} \mathbb{E}_\mu \bigg[ \left\| g(T_\nu(H,t),\widetilde{\beta})_{:,i} - g(T_\rho(H,t),\beta)_{:,i} \right\| \bigg] \\
&\leq \phi_{PT}(r,B_T) \|T_\rho(H,t) - T_\rho(H,t)\|_{2-\mathrm{col}} + \phi_P P(r,B_T) \|\beta - \widetilde{\beta}\| \\
&\lesssim W_1(\rho,\nu) + \|\beta - \widetilde{\beta}\|.
\end{aligned}
\tag{F.1}
$$

The third inequality in Equation (F.1) is derived from Lemma C.3, while the fourth inequality relies on Assumption 3 (i) and (iii). Lastly, bounding $\|T_\rho(H,t) - T_\rho(H,t)\|_{2-\mathrm{col}}$ by $W_1(\rho,\nu)$ is achieved with Lemma C.2.

On the other hand, to bound $J_2$, we have

$$
\begin{aligned}
J_2 &\leq \sqrt{N+1} \sup_{i \in [N+1]} \mathbb{E}_\mu \Big[ \|g(T_\rho(H,t),\beta)_{:,i}\| \, \|p_\nu(H,t) - p_\rho(H,t)\|_F \Big] \\
&\leq \sqrt{N+1} \|(T_\rho(H,t),\beta)\|_{2-\mathrm{col}} \|p_\nu(H,t) - p_\rho(H,t)\|_F \\
&\lesssim \|p_\nu(H,t) - p_\rho(H,t)\|_F.
\end{aligned}
\tag{F.2}
$$

In (F.2), the third inequality relies on Assumption 2 (i). Consequently, to establish $J_2 \lesssim \exp(C_G W_1(\rho,\nu)) - 1$, it is adequate to demonstrate that $\|p_\nu(H,t) - p_\rho(H,t)\|_F \leq I_1 + I_2 \lesssim W_1(\rho,\nu)$, where

$$
I_1 = |\mathrm{Read}[T_\rho(H,1) - T_\nu(H,1)]| \left\| \exp\Big( \int_t^1 \int_\beta \nabla_{\mathrm{vec}[T]} \mathrm{vec}[g(T_\nu(H,s),\beta)] \rho(\beta,s) d\beta ds \Big) \right\|,
$$

$$
\begin{aligned}
I_2 = &\, |\mathrm{Read}[T_\rho(H,1)] - y(H)| \\
&\left\| \exp\Big( \int_t^1 \int_\beta \nabla_{\mathrm{vec}[T]} \mathrm{vec}[g(T_\rho(H,s),\beta)] \rho(\beta,s) d\beta ds \Big) - \exp\Big( \int_t^1 \int_\beta \nabla_{\mathrm{vec}[T]} \mathrm{vec}[g(T_\nu(H,s),\beta)] \rho(\beta,s) d\beta ds \Big) \right\|.
\end{aligned}
$$

From Lemma C.2, it is trivial that $|\mathrm{Read}[T_\rho(H,1) - T_\nu(H,1)]| \leq \|T_\rho(H,1) - T_\nu(H,1)\|_F \lesssim W_1(\rho,\nu)$. Thus, $I_1 \lesssim W_1(\rho,\nu)$ given the boundedness of $\|\nabla_{\mathrm{vec}[T]} \mathrm{vec}[g(T_\nu(H,t),\beta)]\|$ as provided

in Assumption 2 (iii). To bound $I_2$, we have

$$I_2 \lesssim \left\| \exp\left(\int_t^1 \int_\beta \nabla_{\mathrm{vec}[T]}\mathrm{vec}[g(T_\rho(H,s),\beta)]\rho(\beta,s)d\beta ds\right) - \exp\left(\int_t^1 \int_\beta \nabla_{\mathrm{vec}[T]}\mathrm{vec}[g(T_\nu(H,s),\beta)]\rho(\beta,s)d\beta ds\right) \right\|$$

$$\lesssim \left\| \exp\left(\int_t^1 \int_\beta \nabla_{\mathrm{vec}[T]}\mathrm{vec}[g(T_\rho(H,s),\beta)]\rho(\beta,s)d\beta ds - \int_t^1 \int_\beta \nabla_{\mathrm{vec}[T]}\mathrm{vec}[g(T_\nu(H,s),\beta)]\rho(\beta,s)d\beta ds\right) - I_{\mathrm{dimvec}[T]} \right\|$$

$$\lesssim \exp\left(\int_t^1 \int_\beta \left\| \nabla_{\mathrm{vec}[T]}\mathrm{vec}[g(T_\rho(H,s),\beta)] - \nabla_{\mathrm{vec}[T]}\mathrm{vec}[g(T_\nu(H,s),\beta)] \right\| \rho(\beta,s)d\beta ds\right) - 1$$

$$\lesssim \exp\left(\phi_{TT}(N,D,B_T,r)\|g(T_\rho(H,t),\beta) - g(T_\nu(H,t),\beta)\|_F\right) - 1$$

$$\lesssim \exp\left(C_r\phi_{TT}(N,D,B_T,r)\phi_T(N,D,\sqrt{N+1}B_T)(1+r+r^2)W_1(\rho,\nu)\right) - 1$$
(F.3)

for some constant $C_r$ dependent on the parameters listed in the result setting. Here, the first inequality in (F.3) stems from $|\mathrm{Read}[T_\rho(H,1)] - y(H)| \le B_T + B$, the second inequality is ensured by the boundedness of $\|\nabla_{\mathrm{vec}[T]}\mathrm{vec}[g(T_\nu(H,t),\beta)]\|$ as stated in Assumption 2 (iii), and the fourth inequality is provided by Assumption 3 (iv). The last inequality in (F.3) arises from Assumption 2 (iii) and Lemma C.2. By combining Equation (F.3) with the bounds of $J_1$ and $I_1$, we deduce that $J_1 + J_2 \lesssim \exp(C_G W_1(\rho,\nu)) - 1 + \|\beta - \widetilde{\beta}\|$ for some constant $C_G$ dependent on the parameters listed in the result, thereby completing the proof. $\qquad\square$

## F.2 Proof of Lemma D.2

*Proof.* Lemma C.4 demonstrates that $\|\widehat{T}_\Theta(H,t)\|_{2-\mathrm{col}}$ and $\|\widehat{T}_{\widetilde{\Theta}}(H,t)\|_{2-\mathrm{col}}$ are bounded by $B_T = B\exp(K(1+A+A^2))$ for any $H$ and $t \in [0,1]$. We begin by bounding

$$\|\widehat{G}(\beta,\Theta,t) - \widehat{G}(\beta,\widetilde{\Theta},t)\| = \Bigg\{ \frac{1}{2}\mathbb{E}_\mu\Big[\nabla_\theta\mathrm{Tr}\Big(f(\widehat{T}_\Theta(H,t),\theta) - f(\widehat{T}_{\widetilde{\Theta}}(H,t),\theta)\Big)^\top \widehat{p}_\Theta(H,t+\Delta t/2)\Big]^\top$$

$$+\frac{1}{2}\mathbb{E}_\mu\Big[\nabla_\theta\mathrm{Tr} f(\widehat{T}_{\widetilde{\Theta}}(H,t),\theta)^\top\Big(\widehat{p}_\Theta(H,t+\Delta t/2) - \widehat{p}_{\widetilde{\Theta}}(H,t+\Delta t/2)\Big)\Big]^\top,$$

$$\frac{1}{2}\mathbb{E}_\mu\Big[\nabla_w\mathrm{Tr}\Big(h(\widehat{T}_\Theta(H,t+\Delta t/2),w) - h(\widehat{T}_{\widetilde{\Theta}}(H,t+\Delta t/2),w)\Big)^\top \widehat{p}_\Theta(H,t)\Big]^\top$$

$$+\frac{1}{2}\mathbb{E}_\mu\Big[\nabla_w\mathrm{Tr} h(\widehat{T}_{\widetilde{\Theta}}(H,t+\Delta t/2),w)^\top\Big(\widehat{p}_\Theta(H,t) - \widehat{p}_{\widetilde{\Theta}}(H,t)\Big)\Big]^\top\Bigg\}^\top.$$

To demonstrate that $\|\widehat{G}(\beta,\Theta,t) - \widehat{G}(\beta,\widetilde{\Theta},t)\| \le C_G\frac{1}{ML}d(\Theta,\widetilde{\Theta})$, it suffices to show

$$J_1 := \left\|\mathbb{E}_\mu\Big[\nabla_\theta\mathrm{Tr}\Big(f(\widehat{T}_\Theta(H,t),\theta) - f(\widehat{T}_{\widetilde{\Theta}}(H,t),\theta)\Big)^\top \widehat{p}_\Theta(H,t+\Delta t/2)\Big]\right\| \le C_G\frac{1}{ML}d(\Theta,\widetilde{\Theta}),$$
(F.4)

and

$$J_2 := \left\|\mathbb{E}_\mu\Big[\nabla_\theta\mathrm{Tr} f(\widehat{T}_{\widetilde{\Theta}}(H,t),\theta)^\top\Big(\widehat{p}_\Theta(H,t+\Delta t/2) - \widehat{p}_{\widetilde{\Theta}}(H,t+\Delta t/2)\Big)\Big]\right\| \le C_G\frac{1}{ML}d(\Theta,\widetilde{\Theta}),$$
(F.5)

as the other part for $h$ and $w$ follows a similar proof approach.

To bound $J_1$, by Assumption 3 (i) we have

$$J_1 \le \sum_{i=1}^{N+1}\mathbb{E}_\mu\Big[\Big\|\nabla_\theta f(\widehat{T}_\Theta(H,t),\theta)_{:,i} - f(\widehat{T}_{\widetilde{\Theta}}(H,t),\theta)_{:,i}\Big\| \|\widehat{p}_\Theta(H,t+\Delta t/2)_{:,i}\|\Big]$$

$$\le \mathbb{E}_\mu\Big[\sup_{i\in[N+1]}\Big\|\nabla_\theta f(\widehat{T}_\Theta(H,t),\theta)_{:,i} - f(\widehat{T}_{\widetilde{\Theta}}(H,t),\theta)_{:,i}\Big\| \sum_{i=1}^{N+1}\|\widehat{p}_\Theta(H,t+\Delta t/2)_{:,i}\|\Big]$$
(F.6)

$$\le \phi_T(r,B_T)\|\widehat{T}_\Theta(H,t) - \widehat{T}_{\widetilde{\Theta}}(H,t)\|_{2-\mathrm{col}}\sqrt{N+1}\|\widehat{p}_\Theta(H,t+\Delta t/2)\|_F$$

$$\lesssim \phi_T(r,B_T)\sqrt{N+1}\|\widehat{p}_\Theta(H,t+\Delta t/2)\|_F\frac{1}{ML}d(\Theta,\widetilde{\Theta})$$

$$\lesssim \frac{1}{ML}d(\Theta,\widetilde{\Theta}),$$

where the fourth inequality utilizes Lemma C.5 and the last inequality uses Lemma C.6.
To bound $J_2$, by Assumption 2 (ii), we have

$$
\begin{aligned}
J_2 \leq & \sqrt{N+1}\mathbb{E}_\mu\left[\sup_{i\in[N+1]}\left\|f(\widehat{T}_{\widetilde{\Theta}}(H,t),\theta)_{:,i}\right\|\left\|\widehat{p}_\Theta(H,t+\Delta t/2)-\widehat{p}_{\widetilde{\Theta}}(H,t+\Delta t/2)\right\|_F\right]\\
= & \sqrt{N+1}\phi(B_T)(1+r)\mathbb{E}_\mu\left[\left\|\widehat{p}_\Theta(H,t+\Delta t/2)-\widehat{p}_{\widetilde{\Theta}}(H,t+\Delta t/2)\right\|_F\right].
\end{aligned}
\tag{F.7}
$$

Hence, it suffices to show that $\|\widehat{p}_\Theta(H,t+\Delta t/2)-\widehat{p}_{\widetilde{\Theta}}(H,t+\Delta t/2)\|_F \lesssim \frac{1}{ML}d(\Theta,\widetilde{\Theta})$ to establish $J_2 \leq \frac{1}{ML}d(\Theta,\widetilde{\Theta})$. Recalling the formula $\widehat{p}_\Theta$ in (C.16), we have $\|\widehat{p}_\Theta(H,t+\Delta t/2)-\widehat{p}_{\widetilde{\Theta}}(H,t+\Delta t/2)\|_F \leq I_1 + I_2$, where

$$
\begin{aligned}
I_1 = & (\text{Read}[\widehat{T}_\Theta(H,1)-\widehat{T}_{\widetilde{\Theta}}(H,1)]\\
& \left\{\prod_{\substack{(s-t)/\Delta t+2\in[(1-t)/\Delta t]\\ j\in[M]}}\left(I_{\dim\text{vec}[T]}+(\Delta t/2)M^{-1}\sum_{j=1}^{M}\nabla_{\text{vec}[T]}\text{vec}[f(\widehat{T}_\Theta(H,s),\theta_{s,j})]\right)\right.\\
& \left.\prod_{\substack{(s-t)/\Delta t+1\in[(1-t)/\Delta t]\\ j\in[M]}}\left(I_{\dim\text{vec}[T]}+(\Delta t/2)M^{-1}\sum_{j=1}^{M}\nabla_{\text{vec}[T]}\text{vec}[h(\widehat{T}_\Theta(H,s+\Delta t/2),w_{s,j})]\right)\right\}_{DN+d+1,:}\\
\leq & \frac{1}{ML}d(\Theta,\widetilde{\Theta})\\
& \left\|\prod_{\substack{(s-t)/\Delta t+2\in[(1-t)/\Delta t]\\ j\in[M]}}\left(I_{\dim\text{vec}[T]}+(\Delta t/2)M^{-1}\sum_{j=1}^{M}\nabla_{\text{vec}[T]}\text{vec}[f(\widehat{T}_\Theta(H,s),\theta_{s,j})]\right)\right.\\
& \left.\prod_{\substack{(s-t)/\Delta t+1\in[(1-t)/\Delta t]\\ j\in[M]}}\left(I_{\dim\text{vec}[T]}+(\Delta t/2)M^{-1}\sum_{j=1}^{M}\nabla_{\text{vec}[T]}\text{vec}[h(\widehat{T}_\Theta(H,s+\Delta t/2),w_{s,j})]\right)\right\|\\
\leq & \frac{1}{ML}d(\Theta,\widetilde{\Theta})\\
& \exp\left((\Delta t/2)\sum_{(s-t)/\Delta t+1\in[(1-t)/\Delta t]}M^{-1}\sum_{j=1}^{M}\left\|\nabla_{\text{vec}[T]}\text{vec}[f(\widehat{T}_\Theta(H,s),\theta_{s,j})]\right\|\right.\\
& \left.+(\Delta t/2)\sum_{(s-t)/\Delta t+1\in[(1-t)/\Delta t]}M^{-1}\sum_{j=1}^{M}\left\|\nabla_{\text{vec}[T]}\text{vec}[h(\widehat{T}_\Theta(H,t),w_{s,j})]\right\|\right)\\
\leq & \frac{1}{ML}d(\Theta,\widetilde{\Theta})\exp\left(\phi_T(N,D,\sqrt{N+1}KB_T)(1+r+r^2)\right)\\
\lesssim & \frac{1}{ML}d(\Theta,\widetilde{\Theta}),
\end{aligned}
$$

and
$$I_2 = |\mathrm{Read}[\widehat{T}_{\widetilde{\Theta}}(H,1)] + y(H)|$$

$$\Bigg\{ \prod_{\substack{(s-t)/\Delta t+2\in[(1-t)/\Delta t] \\ j\in[M]}} \Big( I_{\mathrm{dimvec}[T]} + (\Delta t/2)M^{-1}\sum_{j=1}^{M} \nabla_{\mathrm{vec}[T]}\mathrm{vec}[f(\widehat{T}_{\Theta}(H,s),\theta_{s,j})] \Big)$$

$$\prod_{\substack{(s-t)/\Delta t+1\in[(1-t)/\Delta t] \\ j\in[M]}} \Big( I_{\mathrm{dimvec}[T]} + (\Delta t/2)M^{-1}\sum_{j=1}^{M} \nabla_{\mathrm{vec}[T]}\mathrm{vec}[h(\widehat{T}_{\Theta}(H,s+\Delta t/2),w_{s,j})] \Big) -$$

$$\prod_{\substack{(s-t)/\Delta t+2\in[(1-t)/\Delta t] \\ j\in[M]}} \Big( I_{\mathrm{dimvec}[T]} + (\Delta t/2)M^{-1}\sum_{j=1}^{M} \nabla_{\mathrm{vec}[T]}\mathrm{vec}[f(\widehat{T}_{\widetilde{\Theta}}(H,t),\widetilde{\theta}_{s,j})] \Big)$$

$$\prod_{\substack{(s-t)/\Delta t+1\in[(1-t)/\Delta t] \\ j\in[M]}} \Big( I_{\mathrm{dimvec}[T]} + (\Delta t/2)M^{-1}\sum_{j=1}^{M} \nabla_{\mathrm{vec}[T]}\mathrm{vec}[h(\widehat{T}_{\widetilde{\Theta}}(H,t+\Delta/2),\widetilde{w}_{s,j})] \Big) \Bigg\}_{DN+d+1,:}$$

$$\leq (B+B_T)$$

$$\cdot \exp\Big( (\Delta t/2) \sum_{(s-t)/\Delta t+1\in[(1-t)/\Delta t]} M^{-1}\sum_{j=1}^{M} \max\{\big\|\nabla_{\mathrm{vec}[T]}\mathrm{vec}[f(\widehat{T}_{\Theta}(H,s),\theta_{s,j})]\big\|, \big\|\nabla_{\mathrm{vec}[T]}\mathrm{vec}[f(\widehat{T}_{\widetilde{\Theta}}(H,t),\widetilde{\theta}_{s,j})]\}\big\|$$

$$+ (\Delta t/2) \sum_{(s-t)/\Delta t+1\in[(1-t)/\Delta t]} M^{-1}\sum_{j=1}^{M} \max\{\big\|\nabla_{\mathrm{vec}[T]}\mathrm{vec}[h(\widehat{T}_{\Theta}(H,t),w_{s,j})]\big\|, \big\|\nabla_{\mathrm{vec}[T]}\mathrm{vec}[h(\widehat{T}_{\widetilde{\Theta}}(H,t),\widetilde{w}_{s,j})]\}\big\| \Big)$$

$$\cdot (\Delta t/2) \sum_{\substack{(s-t)/\Delta t+1\in[(1-t)/\Delta t] \\ j\in[M]}} \Big( \Big\| M^{-1}\sum_{j=1}^{M} \nabla_{\mathrm{vec}[T]}\mathrm{vec}[f(\widehat{T}_{\Theta}(H,s),\theta_{s,j})] - M^{-1}\sum_{j=1}^{M} \nabla_{\mathrm{vec}[T]}\mathrm{vec}[f(\widehat{T}_{\widetilde{\Theta}}(H,t),\widetilde{\theta}_{s,j})] \Big\|$$

$$+ \Big\| M^{-1}\sum_{j=1}^{M} \nabla_{\mathrm{vec}[T]}\mathrm{vec}[h(\widehat{T}_{\Theta}(H,t),w_{s,j})] - M^{-1}\sum_{j=1}^{M} \nabla_{\mathrm{vec}[T]}\mathrm{vec}[h(\widehat{T}_{\widetilde{\Theta}}(H,t),\widetilde{w}_{s,j})] \Big\| \Big)$$

$$\leq (B+B_T)\exp\Big( \phi_T(N,D,\sqrt{N+1}KB_T)(1+r+r^2) \Big)\phi_{TP}(N,D,\sqrt{N+1}B_T,r)\frac{1}{ML}d(\Theta,\widetilde{\Theta})$$

$$\lesssim \frac{1}{ML}d(\Theta,\widetilde{\Theta}).$$

where the first inequality applies Lemma C.8, and the second inequality relies on Assumption 2 (iii) and Assumption 3 (ii). Therefore, we conclude that $I_2 \lesssim \frac{1}{ML}d(\Theta,\widetilde{\Theta})$. By combining the bounds of $I_1$ and $I_2$, we observe that Equation (F.5) holds, thereby establishing $\|\widehat{G}(\beta,\Theta,t) - \widehat{G}(\beta,\widetilde{\Theta},t)\| \leq C_G\frac{1}{ML}d(\Theta,\widetilde{\Theta})$ for some $C_G$ dependent on $N$, $d$, $r$, and the parameters of the assumptions.

It remains to prove that $\|\widehat{G}(\beta,\widetilde{\Theta},t) - \widehat{G}(\widetilde{\beta},\widetilde{\Theta},t)\| \leq C_G(1+\lambda)\|\beta-\widetilde{\beta}\|$. Note that
$$\|\widehat{G}(\beta,\widetilde{\Theta},t) - \widehat{G}(\widetilde{\beta},\widetilde{\Theta},t)\| \leq \lambda\|\beta-\widetilde{\beta}\|$$
$$+ \Big\{ \frac{1}{2}\mathbb{E}_\mu\Big[\nabla_\theta\mathrm{Tr}\Big(f(\widehat{T}_{\widetilde{\Theta}}(H,t),\theta) - f(\widehat{T}_{\widetilde{\Theta}}(H,t),\widetilde{\theta})\Big)^\top \widehat{p}_\Theta(H,t+\Delta t/2)\Big]^\top,$$
$$\frac{1}{2}\mathbb{E}_\mu\Big[\nabla_w\mathrm{Tr}\Big(h(\widehat{T}_{\widetilde{\Theta}}(H,t+\Delta t/2),w) - h(\widehat{T}_{\widetilde{\Theta}}(H,t+\Delta t/2),\widetilde{w})\Big)^\top \widehat{p}_\Theta(H,t)\Big]^\top \Big\}^\top.$$

Therefore, we only need to show that
$$\Big\| \mathbb{E}_\mu\Big[\nabla_\theta\mathrm{Tr}\Big(f(\widehat{T}_{\widetilde{\Theta}}(H,t),\theta) - f(\widehat{T}_{\widetilde{\Theta}}(H,t),\widetilde{\theta})\Big)^\top \widehat{p}_\Theta(H,t+\Delta t/2)\Big] \Big\| \leq C_G\|\theta-\widetilde{\theta}\|,$$

and
$$\Big\| \mathbb{E}_\mu\Big[\nabla_w\mathrm{Tr}\Big(h(\widehat{T}_{\widetilde{\Theta}}(H,t),w) - h(\widehat{T}_{\widetilde{\Theta}}(H,t),\widetilde{w})\Big)^\top \widehat{p}_\Theta(H,t)\Big] \Big\| \leq C_G\|w-\widetilde{w}\|,$$

to obtain $\|\widehat{G}(\beta, \widetilde{\Theta}, t) - \widehat{G}(\widetilde{\beta}, \widetilde{\Theta}, t)\| \leq C_G(1+\lambda)\|\beta - \widetilde{\beta}\|$. Here, we only establish the inequality above for $f$ and $\theta$, as the proof of the other inequality follows a similar pattern. Note that by Assumption 3 (iii), we have

$$\left\| \mathbb{E}_\mu \Big[ \nabla_\theta \mathrm{Tr}\Big( f(\widehat{T}_{\widetilde{\Theta}}(H,t), \theta) - f(\widehat{T}_{\widetilde{\Theta}}(H,t), \widetilde{\theta}) \Big)^\top \widehat{p}_\Theta(H, t + \Delta t/2) \Big] \right\|$$

$$\leq \sup_{i \in [N+1]} \mathbb{E}_\mu \Big[ \Big\| \nabla_\theta \Big( f(\widehat{T}_{\widetilde{\Theta}}(H,t), \theta) - f(\widehat{T}_{\widetilde{\Theta}}(H,t), \widetilde{\theta}) \Big)_{:,i} \Big\| \Big] \sqrt{N+1} \|\widehat{p}_\Theta(H, t + \Delta t/2)\|_F$$

$$\leq \phi_{PP}(r, B_T) \|\theta - \widetilde{\theta}\| \sqrt{N+1} \|\widehat{p}_\Theta(H, t + \Delta t/2)\|_F$$

$$\lesssim \|\theta - \widetilde{\theta}\|.$$

where the last inequality applies Lemma C.6. Therefore, we conclude that $\|\widehat{G}(\beta, \widetilde{\Theta}, t) - \widehat{G}(\widetilde{\beta}, \widetilde{\Theta}, t)\| \leq C_G(1+\lambda)\|\beta - \widetilde{\beta}\|$, completing the proof. $\qquad \square$

### F.3  Proof of Lemma D.3

*Proof.* Lemmas C.1 and C.4 establish that $\max \|T_\rho(H,t)\|_{2-\mathrm{col}}, \|\widehat{T}_\Theta(H,t)\|_{2-\mathrm{col}} \leq B_T := B \exp(K(1+r+r^2))$. According to Lemma D.6, there exists an event $E$ with $\mathbb{P}(E) \geq 1 - \exp(-\delta)$ such that under $E$, we have

$$\|\widehat{T}_\Theta(H,t) - T_\rho(H,t)\|_F \lesssim L^{-1} + \sqrt{\frac{\delta + \log(L+1)}{M}}. \tag{F.8}$$

for any $H$ and $t = 0, \Delta t, \ldots, (L-1)\Delta t, 1$. Following the same proof procedure as in Lemma D.6, with $\rho$ replaced by $\widehat{\rho}$, and bounding only $J_1$ and $J_3$ in the proof (as there is no need to utilize Hoeffding's inequality to bridge the difference due to a finite width $M$), we could obtain the bound

$$\|\widehat{T}_\Theta(H,t) - T_{\widehat{\rho}}(H,t)\|_F \lesssim L^{-1}. \tag{F.9}$$

We present the proof only for the case involving $\rho$. The bounding of $\|\widehat{G}(\beta, \Theta, t) - G(\beta, \widehat{\rho}, t)\|_F$ can be derived analogously by substituting $\rho$ with $\widehat{\rho}$ using Equation (F.9), and skipping the process of bounding $\|D_1 - D_2\|$ where $D_1$ and $D_2$ will be defined later. The bounding of $\|G(\beta, \widehat{\rho}, t) - G(\beta, \rho, t)\|$ can be straightforwardly achieved by combining the results obtained from the other two cases.

By the definitions of the gradients in Equations (C.2) and (C.3), we observe that $\left\| \widehat{G}(\beta, \Theta, t) - G(\beta, \rho, t) \right\| \lesssim \left\| \widehat{G}_f(\theta, \Theta, t) - G_f(\theta, \rho, t) \right\| + \left\| \widehat{G}_h(w, \Theta, t) - G_h(w, \rho, t) \right\|$. We will focus on showing that $\left\| 2(\widehat{G}_f(\theta, \Theta, t) - G_f(\theta, \rho, t)) \right\| \lesssim L^{-1} + \sqrt{\frac{\delta + \log(L+1)}{M}}$, as the other part of the proof follows a similar approach.

Let's define the following quantities $A_1, A_2, B_1, B_2, C_1, C_2$:

$$A_1 := \nabla_\theta \mathrm{vec}[f(\widehat{T}_\Theta(H,t), \theta)], \quad A_2 := \nabla_\theta \mathrm{vec}[f(T_\rho(H,t), \theta)]$$

$$B_1 := \mathrm{Read}[\widehat{T}_\Theta(H,1) - y(H)], \quad B_2 := \mathrm{Read}[T_\rho(H,1) - y(H)]$$

$$C_1 := \Big\{ \prod_{\substack{(s-t)/\Delta t + 2 \in [(1-t)/\Delta t] \\ j \in [M]}} \Big( I_{\mathrm{dimvec}[T]} + (\Delta t/2) M^{-1} \sum_{j=1}^M \nabla_{\mathrm{vec}[T]} \mathrm{vec}[f(\widehat{T}_\Theta(H,s), \theta_{s,j})] \Big)$$

$$\prod_{\substack{(s-t)/\Delta t + 1 \in [(1-t)/\Delta t] \\ j \in [M]}} \Big( I_{\mathrm{dimvec}[T]} + (\Delta t/2) M^{-1} \sum_{j=1}^M \nabla_{\mathrm{vec}[T]} \mathrm{vec}[h(\widehat{T}_\Theta(H, s + \Delta t/2), w_{s,j})] \Big) \Big\}_{DN+d+1,:}$$

$$C_2 := \exp\Big( \int_t^1 \int_\beta \nabla_{\mathrm{vec}[T]} \mathrm{vec}[g(T_\rho(H,t), \beta)] \rho(\beta, t) d\beta dt \Big)_{DN+d+1,:}$$

The universal boundedness of $\|A_1\|$ and $\|A_2\|$ is implied by Assumption 2 (ii), while the universal boundedness of $|B_1|$ and $|B_2|$ is implied by Assumption 1. Additionally, $\|C_1\|$ and $\|C_2\|$ can be

bounded via Assumption 2 (iii), and one can refer to the proofs of Lemmas C.3 and C.6 for detailed explanations.

From (C.14) and (C.16), we could rewrite $J := \left\| 2(\widehat{G}_f(\theta, \Theta, t) - G_f(\theta, \rho, t)) \right\|$ as

$$
\begin{aligned}
J = \|A_1 B_1 C_1 - A_2 B_2 C_2\| &\leq \|(A_1 - A_2) B_2 C_2\| + \|A_1(B_1 - B_2)C_2\| + \|A_1 B_1(C_1 - C_2)\| \\
&\leq \|A_1 - A_2\|\|B_2\|\|C_2\| + \|A_1\|\|B_1 - B_2\|\|C_2\| + \|A_1\|\|B_1\|\|C_1 - C_2\| \\
&\lesssim \|A_1 - A_2\| + \|B_1 - B_2\| + \|C_1 - C_2\|.
\end{aligned}
$$

We claim that to obtain the result, it suffices to show that

     i. $\|A_1 - A_2\| \lesssim L^{-1} + \sqrt{\frac{\delta + \log(L+1)}{M}}$ under event $E$.

     ii. $\|B_1 - B_2\| \lesssim L^{-1} + \sqrt{\frac{\delta + \log(L+1)}{M}}$ under event $E$.

     iii. There exists some event $E_2$ with $\mathbb{P}(E_2) \geq 1 - \exp(-\delta)$ such that $\|C_1 - C_2\| \lesssim L^{-1} + \sqrt{\frac{\delta + \log(L+1)}{M}}$ under $E \cap E_2$.

This is because if we can establish the above statements, then under the event $E \cap E_2$ with $\mathbb{P}(E \cap E_2) \geq 1 - 2\exp(-\delta)$, we obtain $J = \left\| 2(\widehat{G}_f(\theta, \Theta, t) - G_f(\theta, \rho, t)) \right\| \lesssim L^{-1} + \sqrt{\frac{\delta + \log(L+1)}{M}}$. Given the similarity in proof for $\left\| 2(\widehat{G}_h(w, \Theta, t) - G_h(w, \rho, t)) \right\|$, we deduce that with probability at least $1 - 4\exp(-\delta)$ with respect to the parameter initialization $\Theta^{(0)}$, we have $\left\| \widehat{G}(\beta, \Theta, t) - G(\beta, \rho, t) \right\| \lesssim L^{-1} + \sqrt{\frac{\delta + \log(L+1)}{M}}$. The remainder of the proof focuses on bounding the quantities in statements (i)-(iii).

**Proof for statement (i):** By Assumption 3 (iv), we have $\|A_1 - A_2\| \leq \phi_{TT}(N, D, \sqrt{N+1}B_T, r)\|\widehat{T}_\Theta(H, t) - T_\rho(H, t)\|_F \lesssim L^{-1} + \sqrt{\frac{\delta + \log(L+1)}{M}}$ under event $E$.

**Proof for statement (ii):** Under event $E$, it is obvious to see $\|B_1 - B_2\| \leq \|\widehat{T}_\Theta(H, t) - T_\rho(H, t)\|_F \lesssim L^{-1} + \sqrt{\frac{\delta + \log(L+1)}{M}}$.

**Proof for statement (iii):** We further define the following quantities

$$D_1 := \prod_{\substack{(s-t)/\Delta t+2\in[(1-t)/\Delta t] \\ j\in[M]}} \left( I_{\mathrm{dimvec}[T]} + (\Delta t/2)M^{-1}\sum_{j=1}^{M}\nabla_{\mathrm{vec}[T]}\mathrm{vec}[f(\widehat{T}_\Theta(H,s),\theta_{s,j})] \right)$$

$$\prod_{\substack{(s-t)/\Delta t+1\in[(1-t)/\Delta t] \\ j\in[M]}} \left( I_{\mathrm{dimvec}[T]} + (\Delta t/2)M^{-1}\sum_{j=1}^{M}\nabla_{\mathrm{vec}[T]}\mathrm{vec}[h(\widehat{T}_\Theta(H,s+\Delta t/2),w_{s,j})] \right)$$

$$D_2 := \prod_{(s-t)/\Delta t+2\in[(1-t)/\Delta t]} \left( I_{\mathrm{dimvec}[T]} + (\Delta t/2)\int_\beta \nabla_{\mathrm{vec}[T]}\mathrm{vec}[f(\widehat{T}_\Theta(H,s),\theta)]\rho(\beta|s)d\beta \right)$$

$$\prod_{(s-t)/\Delta t+1\in[(1-t)/\Delta t]} \left( I_{\mathrm{dimvec}[T]} + (\Delta t/2)\int_\beta \nabla_{\mathrm{vec}[T]}\mathrm{vec}[h(\widehat{T}_\Theta(H,s+\Delta t/2),w)]\rho(\beta|s)d\beta \right)$$

$$D_3 := \exp\Bigg( \sum_{(s-t)/\Delta t+2\in[(1-t)/\Delta t]} (\Delta t/2)\int_\beta \nabla_{\mathrm{vec}[T]}\mathrm{vec}[f(\widehat{T}_\Theta(H,s),\theta)]\rho(\beta,s)d\beta$$

$$+ \sum_{(s-t)/\Delta t+1\in[(1-t)/\Delta t]} (\Delta t/2)\int_\beta \nabla_{\mathrm{vec}[T]}\mathrm{vec}[h(\widehat{T}_\Theta(H,s+\Delta t/2),w)]\rho(\beta,s)d\beta \Bigg)$$

$$D_4 := \exp\left( \int_t^1 \int_\beta \nabla_{\mathrm{vec}[T]}\mathrm{vec}[g(T_\rho(H,t),\beta)]\rho(\beta,t)d\beta dt \right)$$

Note that $\|C_1 - C_2\| \le \|D_1 - D_2\| + \|D_2 - D_3\| + \|D_3 - D_4\|$. Assumption 2 (iii) indicates that for any $s = 0, \Delta t, \ldots, (L-1)\Delta$ and $j = 1, \ldots, M$, we have

$$\max\{\|\nabla_{\mathrm{vec}[T]}\mathrm{vec}[f(\widehat{T}_\Theta(H,s),\theta_{s,j})]\|, \|\nabla_{\mathrm{vec}[T]}\mathrm{vec}[h(\widehat{T}_\Theta(H,s+\Delta t/2),w_{s,j})]\|\} \le B_J$$

for some constant $B_J$ dependent on the parameters listed in the result. This implies that each column of $\nabla_{\mathrm{vec}[T]}\mathrm{vec}[f(\widehat{T}_\Theta(H,s),\theta_{s,j})]$ or $\nabla_{\mathrm{vec}[T]}\mathrm{vec}[h(\widehat{T}_\Theta(H,s+\Delta t/2),w_{s,j})]$ has $l_2$ norm upper bounded by $B_J$ as well. Applying Hoeffding's inequality to each column of $\nabla_{\mathrm{vec}[T]}\mathrm{vec}[f(\widehat{T}_\Theta(H,s),\theta_{s,j})]$ and $\nabla_{\mathrm{vec}[T]}\mathrm{vec}[h(\widehat{T}_\Theta(H,s+\Delta t/2),w_{s,j})]$, and subsequently calculating the union bound across all columns yields:

$$\mathbb{P}\left( \left\| M^{-1}\sum_{j=1}^{M}\nabla_{\mathrm{vec}[T]}\mathrm{vec}[f(\widehat{T}_\Theta(H,s),\theta_{s,j})] - \int_\beta \nabla_{\mathrm{vec}[T]}\mathrm{vec}[f(\widehat{T}_\Theta(H,s),\theta_{s,j})]\rho(\beta|s)d\beta \right\| \ge \sqrt{(N+1)D}z \right)$$

$$\le 2(N+1)D\exp\left(-\frac{z^2}{2B_J^2}M\right)$$

$$\tag{F.10}$$

and

$$\mathbb{P}\left( \left\| M^{-1}\sum_{j=1}^{M}\nabla_{\mathrm{vec}[T]}\mathrm{vec}[h(\widehat{T}_\Theta(H,t),w_{s,j})] - \int_\beta \nabla_{\mathrm{vec}[T]}\mathrm{vec}[h(\widehat{T}_\Theta(H,t),w_{s,j})]\rho(\beta|s)d\beta \right\| \ge \sqrt{(N+1)D}z \right)$$

$$\le 2(N+1)D\exp\left(-\frac{z^2}{2B_J^2}M\right)$$

$$\tag{F.11}$$

for any $z > 0$. For (F.10) and (F.11), we further the consider the union bound across all $s = 0, \Delta t, \ldots, (L-1)\Delta$, and let $z = B_J\sqrt{2M(\delta + \log(2(N+1)DL))}$, which implies that with probability at least $1 - \exp(-\delta)$ with respect to the parameter initialization $\Theta^{(0)}$, we have

$$\| M^{-1}\sum_{j=1}^{M}\nabla_{\mathrm{vec}[T]}\mathrm{vec}[f(\widehat{T}_\Theta(H,s),\theta_{s,j})] - \int_\beta \nabla_{\mathrm{vec}[T]}\mathrm{vec}[f(\widehat{T}_\Theta(H,s),\theta_{s,j})]\rho(\beta|s)d\beta \|$$

and

$$\| M^{-1}\sum_{j=1}^{M}\nabla_{\mathrm{vec}[T]}\mathrm{vec}[h(\widehat{T}_\Theta(H,t),w_{s,j})] - \int_\beta \nabla_{\mathrm{vec}[T]}\mathrm{vec}[h(\widehat{T}_\Theta(H,t),w_{s,j})]\rho(\beta|s)d\beta \|$$

bounded by $B_J\sqrt{2M(N+1)D(\delta+\log(2(N+1)DL))} \leq C_J\sqrt{\frac{\delta+\log(L+1)}{M}}$ for any $s = 0, \Delta t, \ldots, (L-1)\Delta$. Here, $C_J$ is some constant that only depends on $N, D, r$ and the parameters of the assumptions. Denote this probability event by $E_2$, and we have $\mathbb{P}(E_2) \geq 1 - \exp(-\delta)$. Under $E_2$, by Lemma C.8, we have

$$\|D_1 - D_2\|$$

$$\leq \frac{1}{2L}\Big( \sum_{(s-t)/\Delta t+2\in[(1-t)/\Delta t]} \Big\| M^{-1}\sum_{j=1}^{M} \nabla_{\text{vec}[T]}\text{vec}[f(\widehat{T}_\Theta(H,s),\theta_{s,j})] - \int_\beta \nabla_{\text{vec}[T]}\text{vec}[f(\widehat{T}_\Theta(H,s),\theta)]\rho(\beta,s)d\beta \Big\|$$

$$+ \sum_{(s-t)/\Delta t+1\in[(1-t)/\Delta t]} \Big\| M^{-1}\sum_{j=1}^{M} \nabla_{\text{vec}[T]}\text{vec}[h(\widehat{T}_\Theta(H,s),w_{s,j})] - \int_\beta \nabla_{\text{vec}[T]}\text{vec}[h(\widehat{T}_\Theta(H,s+\Delta t/2),w)]\rho(\beta,s)d\beta \Big\| \Big)$$

$$\leq C_J\sqrt{\frac{\delta+\log(L+1)}{M}}.$$

(F.12)

For any $s = 0, \Delta t, \ldots, (L-1)\Delta$, we define

$$A_{s,j} = (\Delta t/2)\int_\beta \nabla_{\text{vec}[T]}\text{vec}[f(\widehat{T}_\Theta(H,s),\theta)]\rho(\beta|s)d\beta$$

and

$$B_{s,j} = (\Delta t/2)\int_\beta \nabla_{\text{vec}[T]}\text{vec}[h(\widehat{T}_\Theta(H,s+\Delta t/2),w)]\rho(\beta|s)d\beta.$$

Since Assumption 2 (iii) indicates that $\max\{\|A_{s,j}\|, \|B_{s,j}\|\} \lesssim \Delta t = L^{-1}$, we have

$$\|\exp(A_{s,j}) - I - A_{s,j}\| \lesssim \|A_{s,j}\|^2, \quad \|\exp(B_{s,j}) - I - B_{s,j}\| \lesssim \|B_{s,j}\|^2.$$

Applying Lemma C.8 once more, we have

$$\|D_2 - D_3\| \leq \sum_{(s-t)/\Delta t+2\in[(1-t)/\Delta t]} \|A_{s,j}\|^2 + \sum_{(s-t)/\Delta t+1\in[(1-t)/\Delta t]} \|B_{s,j}\|^2 \lesssim L^{-1}. \quad \text{(F.13)}$$

Since Assumption 2 (iii) ensures the boundedness of $\|D_4\|$, we have

$$\|D_3 - D_4\| \lesssim \Big\| \exp\Big( \sum_{(s-t)/\Delta t+2\in[(1-t)/\Delta t]} (\Delta t/2)\int_\beta \nabla_{\text{vec}[T]}\text{vec}[f(\widehat{T}_\Theta(H,s),\theta)]\rho(\beta,s)d\beta$$

$$+ \sum_{(s-t)/\Delta t+1\in[(1-t)/\Delta t]} (\Delta t/2)\int_\beta \nabla_{\text{vec}[T]}\text{vec}[h(\widehat{T}_\Theta(H,s+\Delta t/2),w)]\rho(\beta,s)d\beta$$

$$- \int_t^1\int_\beta \nabla_{\text{vec}[T]}\text{vec}[g(T_\rho(H,t),\beta)]\rho(\beta,t)d\beta dt\Big) - 1 \Big\|.$$

(F.14)

Therefore, to show that $\|D_3 - D_4\| \lesssim L^{-1} + \sqrt{\frac{\delta+\log(L+1)}{M}}$, it suffices to show that

$$J_{34} := \Big\| \sum_{(s-t)/\Delta t+2\in[(1-t)/\Delta t]} (\Delta t/2)\int_\beta \nabla_{\text{vec}[T]}\text{vec}[f(\widehat{T}_\Theta(H,s),\theta)]\rho(\beta,s)d\beta$$

$$+ \sum_{(s-t)/\Delta t+1\in[(1-t)/\Delta t]} (\Delta t/2)\int_\beta \nabla_{\text{vec}[T]}\text{vec}[h(\widehat{T}_\Theta(H,s+\Delta t/2),w)]\rho(\beta,s)d\beta$$

$$- \int_t^1\int_\beta \nabla_{\text{vec}[T]}\text{vec}[g(T_\rho(H,t),\beta)]\rho(\beta,t)d\beta dt \Big\| \lesssim L^{-1} + \sqrt{\frac{\delta+\log(L+1)}{M}}.$$

By Assumption 3 (iv) and Lemma D.6, we have

$$\Big\| \nabla_{\text{vec}[T]}\text{vec}[f(\widehat{T}_\Theta(H,s),\theta)] - \nabla_{\text{vec}[T]}\text{vec}[f(T_\rho(H,s),\theta)] \Big\| \lesssim \|\widehat{T}_\Theta(H,s) - T_\rho(H,s)\|_F \lesssim L^{-1} + \sqrt{\frac{\delta+\log(L+1)}{M}},$$

and

$$\left\|\nabla_{\text{vec}[T]}\text{vec}[h(\widehat{T}_\Theta(H, s + \Delta t/2), w)] - \nabla_{\text{vec}[T]}\text{vec}[h(T_\rho(H, s), w)]\right\| \lesssim \|\widehat{T}_\Theta(H, s + \Delta t/2) - T_\rho(H, s)\|_F$$

$$\lesssim \|\widehat{T}_\Theta(H, s + \Delta t/2) - \widehat{T}_\Theta(H, s)\|_F + \|\widehat{T}_\Theta(H, s) - T_\rho(H, s)\|_F$$

$$\lesssim L^{-1} + \sqrt{\frac{\delta + \log(L+1)}{M}} + L^{-1}\|M^{-1}\sum_{j=1}^{M} h(\widehat{T}_\Theta(H, s), w_{s,j})\|_F$$

$$\lesssim L^{-1} + \sqrt{\frac{\delta + \log(L+1)}{M}},$$

where the last inequality employs Assumption 2 (i). Therefore, we conclude that

$$\|D_3 - D_4\| \lesssim J_{34} \lesssim L^{-1} + \sqrt{\frac{\delta + \log(L+1)}{M}} + \|(\Delta t/2)\int_\beta \nabla_{\text{vec}[T]}\text{vec}[f(T_\rho(H, t), \theta)]\rho(\beta, t)d\beta\|$$

$$\left\|\sum_{(s-t)/\Delta t+1 \in [(1-t)/\Delta t]} (\Delta t/2)\int_\beta \nabla_{\text{vec}[T]}\text{vec}[f(T_\rho(H, s), \theta)]\rho(\beta, s)d\beta\right.$$

$$+ \sum_{(s-t)/\Delta t+1 \in [(1-t)/\Delta t]} (\Delta t/2)\int_\beta \nabla_{\text{vec}[T]}\text{vec}[h(T_\rho(H, s)(H, s), w)]\rho(\beta, s)d\beta$$

$$\left. - \int_t^1 \int_\beta \nabla_{\text{vec}[T]}\text{vec}[g(T_\rho(H, t), \beta)]\rho(\beta, t)d\beta dt\right\|$$

$$\lesssim L^{-1} + \sqrt{\frac{\delta + \log(L+1)}{M}} + \sup_{|s_1-s_2|\leq\Delta t}\left\|\nabla_{\text{vec}[T]}\text{vec}[g(T_\rho(H, s_1), \beta)] - \nabla_{\text{vec}[T]}\text{vec}[g(T_\rho(H, s_2), \beta)]\right\|$$

$$\lesssim L^{-1} + \sqrt{\frac{\delta + \log(L+1)}{M}} + \sup_{|s_1-s_2|\leq\Delta t}\|T_\rho(H, s_1) - T_\rho(H, s_2)\|_F$$

$$\lesssim L^{-1} + \sqrt{\frac{\delta + \log(L+1)}{M}}$$

(F.15)

where the third inequality uses Assumption 3 (iv), and the last inequality relies on the Lipschitz continuity as demonstrated in Proposition C.1. Combining (F.12), (F.13), and (F.15) yields $\|C_1 - C_2\| \leq \|D_1 - D_2\| + \|D_2 - D_3\| + \|D_3 - D_4\| \lesssim L^{-1} + \sqrt{\frac{\delta + \log(L+1)}{M}}.$ $\qquad\square$

## F.4 Proof of Lemma D.4

*Proof.* Define $F_\rho(T_{\bar\rho}, t) = \int_\beta \mathrm{vec}[g(T_{\bar\rho}(H,t),\beta)]\rho(\beta,t)d\beta$ for any $\rho, \bar\rho \in \mathcal{P}^2$. From Taylor's expansion, we have

$$
\begin{aligned}
\mathrm{vec}[\dot T_{\rho_\eta}(H,t) - \dot T_\rho(H,t)] &= F_\rho(T_{\rho_\eta},t) - F_\rho(T_\rho,t) + F_{\rho_\eta}(T_{\rho_\eta},t) - F_\rho(T_{\rho_\eta},t)\\
&= \left(\int_\beta \nabla_{\mathrm{vec}[T]}\mathrm{vec}[g(T_\rho(H,t),\beta)]\rho(\beta,t)d\beta\right)\left(\mathrm{vec}[T_{\rho_\eta}(H,t) - T_\rho(H,t)]\right)\\
&\quad + \eta\int_\beta \mathrm{vec}[g(T_{\rho_\eta}(H,t),\beta)](\nu-\rho)(\beta,t)d\beta + o(\eta)\\
&= \left(\int_\beta \nabla_{\mathrm{vec}[T]}\mathrm{vec}[g(T_\rho(H,t),\beta)]\rho(\beta,t)d\beta\right)\left(\mathrm{vec}[T_{\rho_\eta}(H,t) - T_\rho(H,t)]\right)\\
&\quad + \eta\int_\beta \mathrm{vec}[g(T_\rho(H,t),\beta)](\nu-\rho)(\beta,t)d\beta\\
&\quad + \eta\left(\int_\beta \nabla_{\mathrm{vec}[T]}\mathrm{vec}[g(T_\rho(H,t),\beta)](\nu-\rho)(\beta,t)d\beta\right)\left(\mathrm{vec}[T_{\rho_\eta}(H,t) - T_\rho(H,t)]\right) + o(\eta)\\
&= \left(\int_\beta \nabla_{\mathrm{vec}[T]}\mathrm{vec}[g(T_\rho(H,t),\beta)]\rho(\beta,t)d\beta\right)\left(\mathrm{vec}[T_{\rho_\eta}(H,t) - T_\rho(H,t)]\right)\\
&\quad + \eta\int_\beta \mathrm{vec}[g(T_\rho(H,t),\beta)](\nu-\rho)(\beta,t)d\beta + o(\eta),
\end{aligned}
$$

of which the last equality holds as Lemma C.2 shows that $\|T_{\rho_\eta}(H,t) - T_\rho(H,t)\|_F = O(W_2(\rho_\eta,\rho)) = O(\eta)$, where we hide the constant dependence on $B, K, N, r$. Therefore, we have

$$
\begin{aligned}
&\frac{d}{dt}\left\{\exp\left(-\int_0^\top\int_\beta \nabla_{\mathrm{vec}[T]}\mathrm{vec}[g(T_\rho(H,s),\beta)]\rho(\beta,s)d\beta\right)\left(\mathrm{vec}[T_{\rho_\eta}(H,t) - T_\rho(H,t)]\right)\right\}\\
&= \exp\left(-\int_0^\top\int_\beta \nabla_{\mathrm{vec}[T]}\mathrm{vec}[g(T_\rho(H,s),\beta)]\rho(\beta,s)d\beta\right)\\
&\quad\left\{\mathrm{vec}[\dot T_{\rho_\eta}(H,t) - \dot T_\rho(H,t)] - \left(\int_\beta \nabla_{\mathrm{vec}[T]}\mathrm{vec}[g(T_\rho(H,t),\beta)]\rho(\beta,t)d\beta\right)\left(\mathrm{vec}[T_{\rho_\eta}(H,t) - T_\rho(H,t)]\right)\right\}\\
&= \exp\left(-\int_0^\top\int_\beta \nabla_{\mathrm{vec}[T]}\mathrm{vec}[g(T_\rho(H,s),\beta)]\rho(\beta,s)d\beta\right)\left\{\eta\int_\beta \mathrm{vec}[g(T_\rho(H,t),\beta)](\nu-\rho)(\beta,t)d\beta + o(\eta)\right\},
\end{aligned}
$$

$$\tag{F.16}$$

which leads to (D.18). $\square$

## F.5 Proof of Lemma D.5

*Proof.* Fix $(\beta,t) \in P_r$. Lemma C.2 implies that

$$
\|p_\rho(H,1) - p_\nu(H,1)\|_F = |\mathrm{Read}(T_\rho(H,1) - \mathrm{Read}(T_\nu(H,1)| \le C_r W_1(\rho,\nu) \le C_r(r+1)\|\rho-\nu\|_1
$$

for some constant $C_r$ dependent on the parameters listed in the result. Our goal is to regulate the difference between $\dot p_\rho(H,t)$ and $\dot p_\nu(H,t)$ to control the the propagation of $\|p_\rho(H,\cdot) - p_\nu(H,\cdot)\|$.

Note that by (C.14) and Assumption 2,

$$
\begin{aligned}
\frac{d}{dt}\|p_\rho(H,t) - p_\nu(H,t)\|_F &\leq \|\dot{p}_\rho(H,t) - \dot{p}_\nu(H,t)\|_F \\
&= \Big\| \mathrm{vec}[p_\rho(H,t)]^\top \int_\beta \nabla_{\mathrm{vec}[T]} \mathrm{vec}[g(T_\rho(H,t),\beta)] \rho(\beta,t) d\beta \\
&\quad - \mathrm{vec}[p_\nu(H,t)]^\top \int_\beta \nabla_{\mathrm{vec}[T]} \mathrm{vec}[g(T_\nu(H,t),\beta)] \nu(\beta,t) d\beta \Big\| \\
&\leq \|p_\rho(H,t) - p_\nu(H,t)\|_F \int_\beta \big\| \nabla_{\mathrm{vec}[T]} \mathrm{vec}[g(T_\rho(H,t),\beta)] \big\| \rho(\beta,t) d\beta \\
&\quad + \|p_\nu(H,t)\|_F \int_\beta \big\| \nabla_{\mathrm{vec}[T]} \mathrm{vec}[g(T_\nu(H,t),\beta)] \big\| (\rho-\nu)(\beta,t) d\beta \\
&\leq L_{r,1}\Big( \|p_\rho(H,t) - p_\nu(H,t)\|_F \int_\beta \rho(\beta,t) d\beta + \|p_\nu(H,t)\|_F \|\rho(\cdot,t) - \nu(\cdot,t)\|_1 \Big) \\
&\leq L_{r,1}\Big( \|p_\rho(H,t) - p_\nu(H,t)\|_F \int_\beta \rho(\beta,t) d\beta + L_{r,2} \|\rho(\cdot,t) - \nu(\cdot,t)\|_1 \Big)
\end{aligned}
\tag{F.17}
$$

where

$$
L_{r,1} = \phi_T(N, D, \sqrt{N+1} B \exp(K(1+r+r^2)))(1+r+r^2)
$$

and

$$
L_{r,2} = (B + B \exp(K(1+r+r^2))) \exp\Big( \phi_T(N, D, \sqrt{N+1} KB \exp(K(1+r+r^2))(1+r+r^2) \Big).
$$

The third inequality of (F.17) uses Lemma C.1 to obtain

$$
\|T_\rho(H,t)\|_F \leq \sqrt{N+1} \|T_\rho(H,t)\|_{2-\mathrm{col}} \leq \sqrt{N+1} B \exp(K(1+r+r^2))
$$

and

$$
\|T_\nu(H,t)\|_F \leq \sqrt{N+1} \|T_\nu(H,t)\|_{2-\mathrm{col}} \leq \sqrt{N+1} B \exp(K(1+r+r^2))
$$

with Assumption 2(iii) to bound the norm of the Jacobian matrix with $L_{r,1}$. The last inequality of (F.17) employs Lemma C.3 to bound $\|p_\nu(H,t)\|_F$ with $L_{r,2}$. Applying the Grönwall's inequality to (F.17), we obtain

$$
\begin{aligned}
\|p_\rho(H,t) - p_\nu(H,t)\|_F &\leq C_r(r+1) \exp(L_{r,1}) \|\rho - \nu\|_1 + \int_0^1 L_{r,1} L_{r,2} \exp\Big(L_{r,1} \int_t^1 \int_\beta \rho(\beta,t) d\beta ds\Big) \|\rho(\cdot,t) - \nu(\cdot,t)\|_1 dt \\
&\leq C_r(r+1) \exp(L_{r,1}) \|\rho - \nu\|_1 + L_{r,1} L_{r,2} \exp(L_{r,1}) \int_0^1 \|\rho(\cdot,t) - \nu(\cdot,t)\|_1 dt \\
&= (C_r + L_{r,1} L_{r,2}) \exp(L_{r,1})) \|\rho - \nu\|_1
\end{aligned}
\tag{F.18}
$$

Since

$$
\int_0^1 \|\rho(\cdot,t) - \nu(\cdot,t)\|_1 dt = \int_0^1 \int_\beta |\rho(\beta,t) - \nu(\beta,t)| d\beta dt = \|\rho - \nu\|_1.
$$

Thus, we complete the proof of the first result.

By Lemma C.3, under Assumption 1 we have

$$
\|p_\rho(H,t)\|_F \leq L_{r,3} := (B + B \exp(K(1+r+r^2))) \exp\Big( \phi_T(N, D, \sqrt{N+1} KB \exp(K(1+r+r^2))(1+r+r^2) \Big).
$$

In addition, by Lemma C.1, under Assumption 2 (i) we have

$$
\|g(T_\nu(H,t),\beta)\|_F \leq \sqrt{N+1} \|g(T_\nu(H,t),\beta)\|_{2-\mathrm{col}} \leq L_{r,4} := KB \exp(K(1+r+r^2))(1+r+r^2).
$$

Therefore, for the gradient function $\frac{\delta Q}{\delta \rho}$, by Lemma C.3 we have

$$
\left| \frac{\delta Q}{\delta \rho} \right|_\rho (\beta, t) - \frac{\delta Q}{\delta \rho} \Big|_\nu (\beta, t) \Big| = \mathbb{E}_\mu[\mathrm{Tr}(g(T_\rho(H,t),\beta)^\top p_\rho(H,t)) - \mathrm{Tr}(g(T_\nu(H,t),\beta)^\top p_\nu(H,t))]
$$

$$
\leq \mathbb{E}_\mu[\|g(T_\rho(H,t),\beta) - g(T_\nu(H,t),\beta)\|_F \|p_\rho(H,t)\|_F]
$$
$$
+ \mathbb{E}_\mu[\|g(T_\nu(H,t),\beta)\|_F \|p_\rho(H,t) - p_\nu(H,t)\|_F]
$$
$$
\leq L_{r,3}\mathbb{E}_\mu[\|g(T_\rho(H,t),\beta) - g(T_\nu(H,t),\beta)\|_F] + L_{r,4}(C_r + L_{r,1}L_{r,2})\|\rho - \nu\|_1
$$
$$
\leq \sqrt{N+1}L_{r,3}\mathbb{E}_\mu[\|g(T_\rho(H,t),\beta) - g(T_\nu(H,t),\beta)\|_{2-\mathrm{col}}] + L_{r,4}(C_r + L_{r,1}L_{r,2})\|\rho - \nu\|_1.
$$
(F.19)

Hence, it suffices to show that for any $H$ such that $\|H\|_{2-\mathrm{col}} \leq B$, we have $\|g(T_\rho(H,t),\beta) - g(T_\nu(H,t),\beta)\|_{2-\mathrm{col}} \leq L_{r,5}\|\rho - \nu\|_1$ for some $L_{r,5} > 0$ in order to obtain the second result of this lemma. By Assumption 2 (iii), we see that

$$
\|g(T_\rho(H,t),\beta) - g(T_\nu(H,t),\beta)\|_{2-\mathrm{col}} \leq \phi_T(N, D, \max\{\|T_\rho(H,t)\|_F, \|T_\nu(H,t)\|_F\})(1 + r + r^2)\|T_\rho(H,t) - T_\rho(H,t)\|_2
$$
$$
\leq \phi_T(N, D, \sqrt{N+1}KB\exp(K(1+r+r^2)))(1+r+r^2)C_r W_1(\rho, \nu)
$$
$$
\leq \phi_T(N, D, \sqrt{N+1}KB\exp(K(1+r+r^2)))(1+r+r^2)C_r(1+r)\|\rho - \nu\|_1,
$$
(F.20)

where the second inequality again uses Lemma (C.2). Combining (F.19) and F.20 completes the proof of the second result. $\qquad\square$

## F.6  Proof of Lemma D.6

*Proof.* Denote the empirical distribution of $\bar{T}_\Theta$ and $\widetilde{T}_\rho$ by

$$
\bar{\rho} = \frac{1}{ML}\sum_t \sum_{j=1}^M \delta(\beta_{t,j}, t), \quad \widetilde{\rho} = \frac{1}{L}\sum_t \delta_t(t)\rho(\beta|t),
$$

respectively. It's straightforward to verify that $\bar{\rho}$ and $\widetilde{\rho}$ meet the conditions outlined in Lemma C.1, and $T_{\bar{\rho}} = \bar{T}_\Theta$ and $T_{\widetilde{\rho}} = \widetilde{T}_\rho$. Hence, Lemmas C.1 and C.4 indicate that $\max\{\|\widehat{T}_\Theta(H,t)\|_{2-\mathrm{col}}, \|\bar{T}_\Theta(H,t)\|_{2-\mathrm{col}}, \|\widetilde{T}_\rho(H,t)\|_{2-\mathrm{col}}\} \leq B\exp(K(1+r+r^2))$ for any $H$ and $t \in [0,1]$. We then define $B_T := B\exp(K(1+r+r^2))$.

The following decomposition equation holds:

$$
\|\widehat{T}_\Theta(H,t) - T_\rho(H,t)\|_F \leq \underbrace{\|\widehat{T}_\Theta(H,t) - \bar{T}_\Theta(H,t)\|_F}_{J_1} + \underbrace{\|\bar{T}_\Theta(H,t) - \widetilde{T}_\rho(H,t)\|_F}_{J_2} + \underbrace{\|\widetilde{T}_\rho(H,t) - T_\rho(H,t)\|_F}_{J_3},
$$
(F.21)

Our proof will bound $J_1$, $J_2$ and $J_3$, possibly in a probabilistic manner, to obtain the desired result.

**Bounding $J_1$:** Note that according to Assumption 2 (i),

$$
\|\widehat{T}_\Theta(H,t) + (\Delta t/2)M^{-1}\sum_{j=1}^M f(\widehat{T}_\Theta(H,t), \theta_{t,j})\|_{2-\mathrm{col}}
$$

$$
\leq \|\widehat{T}_\Theta(H,t)\|_{2-\mathrm{col}} + (\Delta t/2)M^{-1}\sum_{j=1}^M \|f(\widehat{T}_\Theta(H,t), \theta_{t,j})\|_{2-\mathrm{col}}
$$

$$
\leq \|\widehat{T}_\Theta(H,t)\|_{2-\mathrm{col}} + (K\Delta t/2)M^{-1}\sum_{j=1}^M \|\widehat{T}_\Theta(H,t)\|_{2-\mathrm{col}}(1 + \|\theta_{t,j}\| + \|\theta_{t,j}\|^2)
$$

$$
\leq B_T(1 + (K\Delta t/2)(1 + r + r^2))
$$
$$
\leq B_T(1 + (K/2)(1 + r + r^2))
$$

Denote $B_T(1 + (K/2)(1 + r + r^2))$ by $\bar{B}_T$. Combining the two equations in (2.4) gives us

$$\widehat{T}_\Theta(H, t + \Delta t) = \text{MLP}_{w_{t,1},\ldots,w_{t,M}}\left(\text{Attn}_{\theta_{t,1},\ldots,\theta_{t,M}}(\widehat{T}_\Theta(H, t), \Delta t/2), \Delta t/2\right)$$

$$=\text{Attn}_{\theta_{t,1},\ldots,\theta_{t,M}}(\widehat{T}_\Theta(H, t), \Delta t/2) + (\Delta t/2)M^{-1}\sum_{j=1}^M h\left(\text{Attn}_{\theta_{t,1},\ldots,\theta_{t,M}}(\widehat{T}_\Theta(H, t), \Delta t/2), w_{t,j}\right)$$

$$=\widehat{T}_\Theta(H, t) + (\Delta t/2)M^{-1}\sum_{j=1}^M f(\widehat{T}_\Theta(H, t), \theta_{t,j}) + (\Delta t/2)M^{-1}\sum_{j=1}^M h\Big(\widehat{T}_\Theta(H, t)$$

$$+(\Delta t/2)M^{-1}\sum_{j=1}^M f(\widehat{T}_\Theta(H, t), \theta_{t,j}), w_{t,j}\Big)$$

$$\text{(F.22)}$$

Then, from the formula of (F.22) and Assumption 2 (iii), we see that for any $t = 0, \Delta t, \ldots, (L-1)\Delta$,

$$\|\widehat{T}_\Theta(H, t + \Delta t) - \bar{T}_\Theta(H, t + \Delta t)\|_F$$

$$\leq\|\widehat{T}_\Theta(H, t) - \bar{T}_\Theta(H, t)\|_F\left(1 + (\Delta t/2)M^{-1}\sum_{j=1}^M \phi_T(N, D, \sqrt{N+1}B_T)(1 + \|\theta_{t,j}\| + \|\theta_{t,j}\|^2)\right)$$

$$+(\Delta t/2)M^{-1}\sum_{j=1}^M \phi_T(N, D, \sqrt{N+1}\bar{B}_T)(1 + \|w_{t,j}\| + \|w_{t,j}\|^2)$$

$$\|\widehat{T}_\Theta(H, t) + (\Delta t/2)M^{-1}\sum_{j=1}^M f(\widehat{T}_\Theta(H, t), \theta_{t,j}) - \bar{T}_\Theta(H, t)\|_F$$

$$\leq\|\widehat{T}_\Theta(H, t) - \bar{T}_\Theta(H, t)\|_F\left(1 + \Delta t M^{-1}\sum_{j=1}^M \phi_T(N, D, \sqrt{N+1}\bar{B}_T)(1 + \|\beta_{t,j}\| + \|\beta_{t,j}\|^2)\right)$$

$$+\sqrt{N+1}B_T(K\Delta t/2)(1 + r + r^2)(\Delta t/2)M^{-1}\sum_{j=1}^M \phi_T(N, D, \sqrt{N+1}\bar{B}_T)(1 + \|w_{t,j}\| + \|w_{t,j}\|^2)$$

$$\leq\|\widehat{T}_\Theta(H, t) - \bar{T}_\Theta(H, t)\|_F\left(1 + \Delta t \phi_T(N, D, \sqrt{N+1}\bar{B}_T)(1 + r + r^2)\right)$$

$$+\sqrt{N+1}\phi_T(N, D, \sqrt{N+1}\bar{B}_T)B_T(K\Delta t^2/4)(1 + r + r^2)^2$$

$$\text{(F.23)}$$

Therefore, applying (F.23), we deduce that for any $t = 0, \Delta t, \ldots, (L-1)\Delta t, 1$, we have:

$$\|\widehat{T}_\Theta(H, t) - \bar{T}_\Theta(H, t)\|_{2-\text{col}} \leq \sqrt{N+1}B_T(K\Delta t^2/4)(1+r+r^2)\frac{\exp(\phi_T(N, D, \sqrt{N+1}\bar{B}_T)(1+r+r^2))}{\Delta t}C_1 L^{-1}$$

$$\text{(F.24)}$$

where $C_1 := \sqrt{N+1}B_T K(1 + r + r^2)\exp(\phi_T(N, D, \sqrt{N+1}\bar{B}_T)(1 + r + r^2))/4$.

**Bounding $J_2$:** For any $t = 0, \Delta t, \ldots, (L-1)\Delta, i \in [N+1], j \in [M]$, we have $\|g(\bar{T}_\Theta(H, t), \beta_{t,j})_{:,i}\| \leq B_T$. Hence, by the Hoeffding's inequality, for any $z > 0$ we have

$$\mathbb{P}(\|M^{-1}\sum_{j=1}^M g(\bar{T}_\Theta(H, t), \beta_{t,j})_{:,i} - \int_\beta g(\bar{T}_\Theta(H, t), \beta_{t,j})_{:,i}\rho(\beta|t)d\beta\| \geq z) \leq 2\exp(-\frac{z^2}{2B_T^2}M).$$

By the union bound over $i \in [N+1]$ and $t = 0, \Delta t, \ldots, (L-1)\Delta$, the above inequality implies

$$\mathbb{P}(\sup_t\|M^{-1}\sum_{j=1}^M g(\bar{T}_\Theta(H, t), \beta_{t,j}) - \int_\beta g(\bar{T}_\Theta(H, t), \beta_{t,j})\rho(\beta|t)d\beta\|_{2-\text{col}} \geq z) \leq 2(N+1)L\exp(-\frac{z^2}{2B_T^2}M).$$

$$\text{(F.25)}$$

We let $z = B_T\sqrt{2M^{-1}(\delta + \log((N+1)L))}$. Then, (F.25) turns into

$$\mathbb{P}(\sup_t\|M^{-1}\sum_{j=1}^M g(\bar{T}_\Theta(H, t), \beta_{t,j}) - \int_\beta g(\bar{T}_\Theta(H, t), \beta_{t,j})\rho(\beta|t)d\beta\|_{2-\text{col}} \geq B_T\sqrt{2M^{-1}(\delta + \log((N+1)L))} \leq \exp(-\delta).$$

$$\text{(F.26)}$$

Denote the event such that

$$\sup_t \|M^{-1}\sum_{j=1}^{M} g(\bar{T}_\Theta(H,t),\beta_{t,j})-\int_\beta g(\bar{T}_\Theta(H,t),\beta_{t,j})\rho(\beta|t)d\beta\|_{2-\text{col}} \le B_T\sqrt{2M^{-1}(\delta+\log((N+1)L))}$$

by $E_\delta$. (F.26) directly indicates $\mathbb{P}(E_\delta) \ge 1 - \exp(-\delta)$.

Suppose that the high probability event $E_\delta$ occurs. Let's denote $B_T\sqrt{2M^{-1}(\delta+\log((N+1)L))}$ by $B_\delta$ for brevity. From Assumption 2 (iii), it follows that for any $t = 0, \Delta t, \dots, (L-1)\Delta$,

$$
\begin{aligned}
\|\bar{T}_\Theta(H,t+\Delta t) - \widetilde{T}_\rho(H,t+\Delta t)\|_F \le &\|\bar{T}_\Theta(H,t) - \widetilde{T}_\rho(H,t)\|_F \\
&+\Delta t\|M^{-1}\sum_{j=1}^{M} g(\bar{T}_\Theta(H,t),\beta_{t,j}) - \int_\beta g(\bar{T}_\Theta(H,t),\beta_{t,j})\rho(\beta|t)d\beta\|_F \\
&+\Delta t\|\int_\beta (g(\bar{T}_\Theta(H,t),\beta_{t,j}) - g(\widetilde{T}_\rho(H,t),\beta_{t,j}))\rho(\beta|t)d\beta\|_F \\
\le &\|\bar{T}_\Theta(H,t) - \widetilde{T}_\rho(H,t)\|_F + \Delta t\sqrt{N+1}B_\delta+ \\
&+\Delta t\int_\beta \|g(\bar{T}_\Theta(H,t),\beta_{t,j}) - g(\widetilde{T}_\rho(H,t),\beta_{t,j})\|_F\rho(\beta|t)d\beta \\
\le &\|\bar{T}_\Theta(H,t) - \widetilde{T}_\rho(H,t)\|_F(1 + \Delta t\phi_T(N,D,\sqrt{N+1}B_T)(1+r+r^2)) \\
&+\Delta t\sqrt{N+1}B_\delta.
\end{aligned}
$$
(F.27)

Repeatedly applying Equation (F.27) yields

$$
\begin{aligned}
\|\widehat{T}_\Theta(H,t) - \bar{T}_\Theta(H,t)\|_F \le &\frac{\sqrt{N+1}B_\delta}{\phi_T(N,D,\sqrt{N+1}B_T)(1+r+r^2)}\exp(\phi_T(N,D,\sqrt{N+1}B_T)(1+r+r^2)) \\
=&C_2\sqrt{M^{-1}(\delta+\log((N+1)L))}.
\end{aligned}
$$
(F.28)

for some constant $C_2 > 0$ dependent on the parameters listed in the result.

**Bounding $J_3$:** It's worth noting that the convergence proof with a convergence rate of $O(\Delta t) = O(\frac{1}{L})$ for $\widetilde{T}_\rho(H,t)$, the first-order Euler method for $T_\rho(H,t)$, is non-standard. This departure from convention arises because we do not assume the boundedness of the second-order derivative $\frac{d^2 T_\rho(H,t)}{dt^2}$, instead relying on the continuity of $\rho(\cdot,t)$ with respect to the depth index $t$. In this proof, $O(\cdot)$ hides dependencies on $N$, $D$, $r$, and the parameters of the assumptions.

From the definition of $\widetilde{T}_\rho$ and $T_\rho$, we have

$$
\begin{aligned}
\|\widetilde{T}_\rho(H,t+\Delta t) - T_\rho(H,t+\Delta t)\|_F \le &\|\widetilde{T}_\rho(H,t) - T_\rho(H,t)\|_F \\
&+ \underbrace{\|T_\rho(H,t+\Delta t) - T_\rho(H,t) - \Delta t\int_\beta g(T_\rho(H,t),\beta)\rho(\beta|t)d\beta\|_F}_{I_1} \\
&+ \underbrace{\Delta t\|\int_\beta (g(T_\rho(H,t),\beta) - g(\widetilde{T}_\rho(H,t),\beta))\rho(\beta|t)d\beta\|_F}_{I_2}.
\end{aligned}
$$
(F.29)

To bound $I_1$, we use (3.1) to get

$$
\begin{aligned}
I_1 \leq &\| \int_t^{t+\Delta t} \int_\beta g(T_\rho(H,s),\beta)\rho(\beta|s)d\beta ds - \Delta t \int_\beta g(T_\rho(H,t),\beta)\rho(\beta|t)d\beta \|_F \\
\leq &\Delta t \sup_{s\in[t,t+\Delta t]} \| \int_\beta g(T_\rho(H,s),\beta)\rho(\beta|s)d\beta - \int_\beta g(T_\rho(H,t),\beta)\rho(\beta|t)d\beta \|_F \\
\leq &\Delta t \sup_{s\in[t,t+\Delta t]} \| \int_\beta (g(T_\rho(H,s),\beta) - g(T_\rho(H,t),\beta))\rho(\beta|s)d\beta \|_F \\
&+ \Delta t \sup_{s\in[t,t+\Delta t]} \| \int_\beta g(T_\rho(H,t),\beta)(\rho(\beta|s) - \rho(\beta|t))d\beta \|_F \\
\leq &\Delta t \sup_{s\in[t,t+\Delta t]} \int_\beta \|g(T_\rho(H,s),\beta) - g(T_\rho(H,t),\beta)\|_F \rho(\beta|s)d\beta \\
&+ \Delta t \sup_{s\in[t,t+\Delta t]} \| \int_\beta g(T_\rho(H,t),\beta)(\rho(\beta|s) - \rho(\beta|t))d\beta \|_F
\end{aligned}
\tag{F.30}
$$

Given that Proposition C.1 establishes the Lipschitz continuity of $T_\rho(H,t)$ with respect to $t$ under the condition that $\rho \in \mathcal{P}^2$ has a bounded support, we can conclude:

$$
\sup_{s\in[t,t+\Delta t]} \|g(T_\rho(H,s),\beta) - g(T_\rho(H,t),\beta)\|_F \leq C_{3,1}|t-s| \leq C_{3,1}\Delta t
\tag{F.31}
$$

for some constant $C_{3,1} > 0$ dependent on the parameters listed in the result. Furthermore, Lemma C.1 and Proposition C.1 demonstrate that $g(T_\rho(H,t),\beta)$ is both bounded and Lipschitz continuous with respect to $\beta$. Thus, we have

$$
\begin{aligned}
\sup_{s\in[t,t+\Delta t]} \| \int_\beta g(T_\rho(H,t),\beta)(\rho(\beta|s) - \rho(\beta|t))d\beta \|_F = &\sup_{s\in[t,t+\Delta t]} \| \int_\beta g(T_\rho(H,t),\beta)(\rho(\beta,s) - \rho(\beta,t))d\beta \|_F \\
\leq &\sup_{s\in[t,t+\Delta t]} C_{3,2}\|\rho(\cdot,s) - \rho(\cdot,t)\|_{\mathrm{BL}} \\
\leq &C_{3,2}C_\rho\Delta t
\end{aligned}
\tag{F.32}
$$

for some constant $C_{3,2} > 0$ dependent on the parameters listed in the result. Substituting (F.31) and (F.32) into (F.30), we find that there exists a constant $C_{3,5}$ dependent on the parameters listed in the result such that $I_1 \leq C_{3,5}\Delta t^2$.

Additionally, Assumption 2 (iii) implies that

$$
\begin{aligned}
I_2 \leq &\int_\beta \|g(T_\rho(H,t),\beta) - g(\widetilde{T}_\rho(H,t),\beta)\|_F \rho(\beta|t)d\beta \\
\leq &\phi_T(N,D,\sqrt{N+1}B_T)(1+r+r^2)\|T_\rho(H,t),\beta) - \widetilde{T}_\rho(H,t),\beta)\|_F
\end{aligned}
\tag{F.33}
$$

Therefore, by bounding $I_1 + I_2$, we obtain the following inequality

$$
\|T_\rho(H,t+\Delta t),\beta) - \widetilde{T}_\rho(H,t+\Delta t),\beta)\|_F \leq C_{3,5}\Delta t^2 + (1+C_{3,6}\Delta t)\|T_\rho(H,t),\beta) - \widetilde{T}_\rho(H,t),\beta)\|_F,
\tag{F.34}
$$

which implies, after being used multiple times, that

$$
\|T_\rho(H,t),\beta) - \widetilde{T}_\rho(H,t),\beta)\|_F \leq C_{3,5}\Delta t^2 \frac{(1+C_{3,6}\Delta t)^{L+1} - 1}{C_{3,6}\Delta t} \leq C_3 L^{-1}
\tag{F.35}
$$

for any $t = 0, \Delta t, \ldots, (L-1)\Delta t, 1$ and some constant $C_3$ dependent on the parameters listed in the result. Combining (F.24), (F.28), and (F.35) yields the desired result. $\square$

### F.7 Proof of Lemma E.1

*Proof.* Let $\widetilde{\mu}(t)$ be the measure induced by $T_\rho(H,t)$ with $H \sim \mu$, and $\widetilde{\mu}$ be $\widetilde{\mu}(t^*)$. By verifying Lemma C.1, we establish that $\|T_\rho(H,t)\|_{2-\mathrm{col}} \leq B_T := B\exp(K(1+r+r^2))$ for any $H$ and

$t \in [0,1]$. Consequently, $\mathrm{supp}(\widetilde{\mu}(t)) \subset \{T : \|T\|_{2-\mathrm{col}} \leq B_T\}$ for any $t \in [0,1]$. The remainder of the proof involves four steps:

**Step I: Show that $p_\rho(H, t)$ is Lipschitz continuous with respect to $(H, t)$**

In the proof of this proposition, when referring to the Lipschitz continuity of a function, we imply its Lipschitz continuity for $H$ within the support of $\mu$. Recall that we have shown in Lemma C.3 that $p_\rho(H, t)$ is universally bounded and Lipschitz continuous for $\|\cdot\|_F$ and any $t \in [0,1]$.

From the formula of $p_\rho$ in (C.14), for any $H, H'$, we have

$$\|p_\rho(H, t) - p_\rho(H', t)\|_F = \|\mathrm{vec}[p_\rho(H, t)] - \mathrm{vec}[p_\rho(H', t)]\|$$

$$\leq \underbrace{|\mathrm{Read}[T_\rho(H, 1) - T_\rho(H', 1)] - (y(H) - y(H'))|}_{I_1} \underbrace{\left\| \exp \left( \int_t^1 \int_\beta \nabla_{\mathrm{vec}[T]} \mathrm{vec}[g(T_\rho(H, s), \beta)] \rho(\beta, s) d\beta ds \right) \right\|}_{I_2}$$

$$+ \underbrace{|\mathrm{Read}[T_\rho(H', 1)] - y(H')|}_{I_3}$$

$$\cdot \underbrace{\left\| \exp \left( \int_t^1 \int_\beta \nabla_{\mathrm{vec}[T]} \mathrm{vec}[g(T_\rho(H, s), \beta)] \rho(\beta, s) d\beta ds \right) - \exp \left( \int_t^1 \int_\beta \nabla_{\mathrm{vec}[T]} \mathrm{vec}[g(T_\rho(H', s), \beta)] \rho(\beta, s) d\beta ds \right) \right\|}_{I_4}.$$

$$\tag{F.36}$$

From Proposition C.1, we observe that $p_\rho(H, t)$ is Lipschitz continuous. Since $y(\cdot)$ is $K_y$-Lipschitz continuous, as given in Assumption 4 (iii), we obtain that $I_1 \lesssim \|H - H'\|_F$. Moreover, from Assumption 1 and Lemma C.1, we see that $I_3$ is universally bounded. Therefore, to show that $p_\rho(H, t)$ is Lipschitz continuous for $\|\cdot\|_F$, it suffices to demonstrate that $I_2 \lesssim 1$ and $I_4 \lesssim \|H - H'\|_F$.

From Assumption 2 (iii), we have

$$I_2 \leq \exp \left( \int_t^1 \int_\beta \left\| \nabla_{\mathrm{vec}[T]} \mathrm{vec}[g(T_\rho(H, s), \beta)] \right\| \rho(\beta, s) d\beta ds \right) \leq \exp \left( \phi_T(N, D, \sqrt{N+1}B_T)(1+r+r^2) \right).$$

Thus, dividing $I_4$ by the uniformly bounded part $I_2$, we obtain

$$I_4 \leq \left\| \exp \left( \int_t^1 \int_\beta \nabla_{\mathrm{vec}[T]} \left( \mathrm{vec}[g(T_\rho(H', s), \beta)] - \nabla_{\mathrm{vec}[T]} \mathrm{vec}[g(T_\rho(H, s), \beta)] \right) \rho(\beta, s) d\beta ds \right) - I_{\mathrm{divvec}[T]} \right\|$$

$$\leq \exp \left( \int_t^1 \int_\beta \left\| \nabla_{\mathrm{vec}[T]} \left( \mathrm{vec}[g(T_\rho(H', s), \beta)] - \nabla_{\mathrm{vec}[T]} \mathrm{vec}[g(T_\rho(H, s), \beta)] \right) \right\| \rho(\beta, s) d\beta ds \right) - 1$$

$$\leq \exp \left( \phi_{TT}(N, D, \sqrt{N+1}B_T, r) \sup_{t \in [0,1]} \|T_\rho(H, t) - T_\rho(H', t)\|_F \right) - 1,$$

$$\tag{F.37}$$

where the final inequality holds by Assumption 3 (iv).

By the Lipschitz continuity of $T_\rho(H, t)$ for $\|\cdot\|_F$ as stated in Proposition C.1, we have $\sup_{t \in [0,1]} \|T_\rho(H, t) - T_\rho(H', t)\|_F \lesssim \|H - H'\|_F$. Hence, utilizing the universal boundedness of the last equation in (F.37), we derive $I_4 \lesssim \|H - H'\|_F$.

Considering all assertions regarding $I_1, I_2, I_3$ and $I_4$, we conclude that $p_\rho(H, t)$ is $C_p$-Lipschitz continuous with respect to $H$ for some constant $C_p$ dependent on the parameters listed in the result that is sufficiently large. The Lipschitz continuity of $p_\rho(H, t)$ with respect to $t$ could be easily derived from the boundedness of the Jacobian matrix, as asserted in Assumption 2 (iii). Moreover, since $p_\rho(H, t)$ is universally bounded shown in Lemma C.3, we conclude that $p_\rho(H, t)$ is Lipschitz continuous with respect to $(H, t)$ for some universal Lipschitz constant $C_p$ dependent on the parameters listed in the result that is sufficiently large.

**Step II: Prepare bounds related to $p_\rho$ for later use**

(C.14) implies that $p_\rho$ solves the adjoint equation

$$\mathrm{vec}[\dot{p}_\rho(H, t)]^\top = -\mathrm{vec}[p_\rho(H, t)]^\top \int_\beta \nabla_{\mathrm{vec}[T]} \mathrm{vec}[g(T_\rho(H, t), \beta)] \rho(\beta, t) d\beta \tag{F.38}$$

with

$$\left\| \int_\beta \nabla_{\text{vec}[T]} \text{vec}[g(T_\rho(H,t),\beta)]\rho(\beta,t)d\beta \right\| \leq (1+C_\rho/2)\phi_T(N,D,\sqrt{N+1}B_T)(1+r+r^2) =: C_0$$

implied by Assumption 2 (iii). Therefore, the Grönwall's inequality directly indicates that

$$\mathbb{E}_\mu \|p_\rho(H,t)\|_F^2 \geq \exp(-2C_0t)\mathbb{E}_\mu \|p_\rho(H,1)\|_F^2 \geq \exp(-2C_0)\mathbb{E}_\mu[|\text{Read}[T_\rho(H,1)]-y(H)|^2] \geq 2\exp(-2C_0)R(\rho),$$

$$\mathbb{E}_\mu \|p_\rho(H,t)\|_F^2 \leq \exp(2C_0t)\mathbb{E}_\mu \|p_\rho(H,1)\|_F^2 \leq \exp(2C_0)\mathbb{E}_\mu[|\text{Read}[T_\rho(H,1)]-y(H)|^2] \leq 2\exp(2C_0)R(\rho),$$

for any $t \in [0,1]$.

**Step III: Construct the descent direction $\nu$**

By the well-posedness of the ODE solution to (3.1) as shown in Proposition C.1, the solution map $T_\rho(H,\cdot)$ is invertible. Hence, for any $\rho \in \mathcal{P}^2$, there exists a continuous inverse map $T_t^{-1}$ such that $T_t^{-1}(T_\rho(H,t)) = H$ for any $H$ and $t \in [0,1]$. Let's define the following function $\bar{F}(T)$ to approximate

$$\bar{F}(T) = -p_\rho(T_{t^*}^{-1}(T),t^*) + \int_\beta g(T,\beta)\rho(\beta|t^*)d\beta - \frac{h(T,w_0)}{2},$$

The function $\bar{F}(T)$, arising from the composition of $p_\rho(\cdot,t)$ and $T_{t^*}^{-1}(\cdot)$, exhibits continuity over $T \in \text{supp}(\widetilde{\mu})$ owing to the continuous nature of $p_\rho(\cdot,t)$ and $T_{t^*}^{-1}(\cdot)$. Therefore, Assumption 4 (ii) could be applied to $\bar{F}$. Since $f(\cdot,\theta)$ is a universal kernel constrained on $\theta \in \mathbb{R}^{\dim\theta_1} \times \mathcal{K}$ [52], and $\bar{F}(T)$ is continuous with respect to $T$, there exists a sequence $\{(c_k,\theta^k)\}_{k\geq 0} \subset (\mathbb{R} \times \mathbb{R}^{\dim\theta_1} \times \mathcal{K})^{\mathbb{N}}$ such that

$$\left\| \bar{F}(T) - \sum_{k=1}^\infty c_k f(T,\theta^k) \right\|_{\max} \leq \epsilon/3 \tag{F.39}$$

given some $\epsilon > 0$ and any $T$ such that $\|T\|_{2-\text{col}} \leq B_T$. Notably, since $f(T,\theta_1^k,\theta_2^k) = -f(T,-\theta_1^k,\theta_2^k)$, we could assume without loss of generality that $c^k \geq 0$ for any $k$. Furthermore, there exists a constant $k_\epsilon$ such that

$$\left\| \bar{F}(T) - \sum_{k=1}^{k_\epsilon} c_k f(T,\theta^k) \right\|_{\max} \leq 2\epsilon/3. \tag{F.40}$$

We define $C(\epsilon) := \sum_{k=1}^{k_\epsilon} c_k$, and $\bar{\nu}(\beta) \in \mathcal{P}(\mathbb{R}^{\dim\beta})$ as the probability distribution such that, given $\bar{\beta} \sim \bar{\nu}$ and for any $k \geq 0$, $\bar{\beta}$ has probability $c^k/C(\epsilon)$ of being $(2C(\epsilon)\theta_1^k,\theta_2,0)$. Then, (F.40) transforms into

$$\left\| F(T) - \int_\beta g(T,\beta)\bar{\nu}(\beta)d\beta \right\|_{\max} \leq 2\epsilon/3, \tag{F.41}$$

where

$$F(T) == -p_\rho(T_{t^*}^{-1}(T),t^*) + \int_\beta g(T,\beta)\rho(\beta|t^*)d\beta.$$

From (F.41), we claim that there exists some $R(\epsilon)$ such that

- $\bar{\nu}(\{\beta : 1/R(\epsilon) \leq \|\beta\| \leq R(\epsilon)\}) \geq 1/2$,

- $\left\| \int_{\|\beta\|>R(\epsilon)} g(T,\beta)\bar{\nu}(\beta)d\beta \right\|_{\max} \leq \epsilon/3$.

We are now in the position to define the descent direction $\nu$. By defining $\nu \in \mathcal{P}(\mathbb{R}^{\dim\beta})$ as the measure obtained by truncating any part outside $1/R(\epsilon) \leq \|\beta\| \leq R(\epsilon)$ from $\bar{\nu}$, and scaling the measure function by $1/\bar{\nu}(\{\beta : 1/R(\epsilon) \leq \|\beta\| \leq R(\epsilon)\}) \leq 2$, we can establish that

$$\left\| F(T) - \int_\beta g(T,\beta)\nu(\beta)d\beta \right\|_{\max} \leq \epsilon. \tag{F.42}$$

for any $T$ such that $\|T\|_{2-\text{col}} \leq B_T$. A straightforward deduction from (F.42) is that for any $T$ such that $\|T\|_{2-\text{col}} \leq B_T$, we have $\|F(T) - \int_\beta g(T,\beta)\nu(\beta)d\beta\|_F \leq \epsilon\sqrt{(N+1)D}$. It is clear that $\nu$ has

a bounded support as $\{\beta : 1/R(\epsilon) \le \|\beta\| \le R(\epsilon)\}$. We will determine the value of $\epsilon$ later, ensuring it based only on $N$, $D$, $r$, and the parameters of the assumptions.

**Step IV: Upper bound $\int_\beta \frac{\delta Q}{\delta \rho}(\beta, t^*)\Big(\nu(\beta) - \rho(\beta|t^*)\Big)d\beta$ to complete the proof**

Utilizing the gradient definition in (3.4) and $\int_\beta g(T, \beta)\rho(\beta|t^*)d\beta = F(T) + p_\rho(T_{t^*}^{-1}(T), t^*)$, we obtain

$$\int_\beta \frac{\delta Q}{\delta \rho}(\beta, t^*)\Big(\nu(\beta) - \rho(\beta|t^*)\Big)d\beta$$

$$= \mathbb{E}_\mu \int_\beta \mathrm{Tr}\Big[g(T_\rho(H, t^*), \beta)^\top p_\rho(H, t^*)\Big]\Big(\nu(\beta) - \rho(\beta|t^*)\Big)d\beta + \frac{\lambda}{2}\int_\beta \|\beta\|^2 \Big(\nu(\beta) - \rho(\beta|t^*)\Big)d\beta$$

$$\le \mathbb{E}_{T \sim \widetilde{\mu}(t)} \mathrm{Tr}\Big[g(T, \beta)^\top \Big(\widetilde{\nu}(\beta) - \rho(\beta|t^*)\Big)p_\rho(T_{t^*}^{-1}(T), t^*)\Big] + \frac{\lambda}{2}(R(\epsilon) + r)$$

$$= \underbrace{\mathbb{E}_{T \sim \widetilde{\mu}(t)} \mathrm{Tr}\Big[\Big(F(T) - \int_\beta g(T, \beta)\nu(\beta)d\beta\Big)^\top p_\rho(T_{t^*}^{-1}(T), t^*)\Big]}_{J_1}$$

$$- \underbrace{\mathbb{E}_{T \sim \widetilde{\mu}(t)} \mathrm{Tr}\Big[p_\rho(T_{t^*}^{-1}(T), t^*)^\top p_\rho(T_{t^*}^{-1}(T), t^*)\Big]}_{J_2} + \frac{\lambda}{2}(R(\epsilon) + r)$$

For $\rho \in \mathcal{P}^2$ concentrated on $P_r$, we observe

$$R(\rho) \le \frac{1}{2}\mathbb{E}_\mu\Big[\Big(|\mathrm{Read}[T_\rho(H, 1)]| + |y(H)|\Big)^2\Big] \le \frac{1}{2}(B_T + B)^2.$$

Hence, to bound $J_1$, we have

$$\begin{aligned}
J_1 &\le \mathbb{E}_{T \sim \widetilde{\mu}(t)} \|F(T) - \int_\beta g(T, \beta)\nu(\beta)d\beta\|_F \cdot \|p_\rho(T_{t^*}^{-1}(T), t^*)\|_F dt \\
&\le \sqrt{(N+1)D}\epsilon \mathbb{E}_{T \sim \widetilde{\mu}(t)} \|p_\rho(T_{t^*}^{-1}(T), t^*)\|_F dt \\
&\le (N+1)D\epsilon \Big(\mathbb{E}_{T \sim \widetilde{\mu}(t)} \|p_\rho(T_{t^*}^{-1}(T), t^*)\|_F^2\Big)^{1/2} dt \\
&= (N+1)D\epsilon \Big(\mathbb{E}_\mu \|p_\rho(H, t)\|_F^2\Big)^{1/2} dt \\
&\le \sqrt{2}(N+1)D\exp(C_0)R(\rho)^{1/2}\epsilon \\
&\le (N+1)(B_T + B)(N+1)D\exp(C_0)R(\rho)\epsilon \\
&= C_3 R(\rho)\epsilon,
\end{aligned}$$
(F.43)

where $C_3 := (N+1)(B_T + B)(N+1)D\exp(C_0)$.

To bound $J_2$, we have

$$\begin{aligned}
J_2 &= \mathbb{E}_{T \sim \widetilde{\mu}(t)} \|p_\rho(T_t^{-1}(T), t)\|_F^2 dt \\
&= \mathbb{E}_\mu \|p_\rho(H, t)\|_F^2 dt \\
&\ge \mathbb{E}_\mu \exp(-2C_0)\|p_\rho(H, 1)\|_F^2 dt \\
&\ge \frac{1}{2}\exp(-2C_0)R(\rho).
\end{aligned}$$
(F.44)

Combining (F.43) and (F.44), by choosing $\epsilon = \frac{1}{4}\exp(-2C_0)/C_3$, which only depends on $N$, $D$, $r$, and the parameters of the assumptions, we have

$$\int_\beta \frac{\delta Q}{\delta \rho}(\beta, t^*)\Big(\nu(\beta) - \rho(\beta|t^*)\Big)d\beta \le -\frac{1}{4}\exp(-2C_0)R(\rho) + \frac{\lambda}{2}\Big(R(\frac{1}{4}\exp(-2C_0)/C_3) + r\Big).$$

Setting $B_r = R(\frac{1}{4}\exp(-2C_0)/C_3)$, $C_1 = \Big(R(\frac{1}{4}\exp(-2C_0)/C_3) + r\Big)/2$, and $C_2 = \frac{1}{4}\exp(-2C_0)$ completes the proof. $\qquad\square$

## F.8   Proof of Lemma C.1

*Proof.* By applying the Cauchy-Schwarz inequality, we trivially obtain $\int_0^1 \int_\beta ||\beta||_2 \rho(\beta, t) d\beta dt \le A$. Thus, leveraging (3.1) and Assumption 2 (i), we can infer

$$\frac{d}{dt} \|T_\rho(H, t)\|_{2-\text{col}} \le \|\dot{T}_\rho(H, t)\|_{2-\text{col}} = \|\int_\beta g(T_\rho(H, t), \beta) \rho(\beta, t) d\beta\|_{2-\text{col}}$$

$$\le \int_\beta \|g(T_\rho(H, t), \beta)\|_{2-\text{col}} \rho(\beta, t) d\beta$$

$$\le \int_\beta K(1 + ||\beta||_2 + ||\beta||_2^2) \rho(\beta, t) \|T_\rho(H, t)\|_{2-\text{col}} d\beta.$$

(F.45)

Therefore, by the Grönwall's inequality, we have

$$\|T_\rho(H, t)\|_{2-\text{col}} \le \|T_\rho(H, 0)\|_{2-\text{col}} \exp\left(\int_0^1 \int_\beta K(1 + ||\beta||_2 + ||\beta||_2^2) \rho(\beta, t) d\beta dt\right)$$

$$\le \|H\|_{2-\text{col}} \exp(K(1 + A + A^2)).$$

$\square$

## F.9   Proof of Lemma C.2

*Proof.* As per Lemma C.1, the boundedness of $\|T_\nu(H, t)\|_{2-\text{col}}$ and $\|T_\rho(H, t)\|_{2-\text{col}}$ is established by a constant $C := \|H\|_{2-\text{col}} \exp(K(1 + A + A^2)) > 0$ for all $t \in [0, 1]$. Consequently, from (3.1), this implies

$$\|T_\nu(H, t_1) - T_\nu(H, t_2)\|_{2-\text{col}} \le \int_{t_1}^{t_2} \|\dot{T}_\nu(H, t)\|_{2-\text{col}}$$

$$\le \int_{t_1}^{t_2} \int_\beta \|g(T_\nu(H, t), \beta\|_{2-\text{col}} \nu(\beta, t) d\beta dt$$

(F.46)

$$\le (1 + C_\rho/2) KC(1 + r + r^2)(t_2 - t_1).$$

for any $t_1, t_2 \in [0, 1]$. Therefore, $T_\nu(H, t)$ is $(1 + C_\rho/2) KC(1 + r + r^2)$-Lipschitz with respect to $t$ for $\|\cdot\|_{2-\text{col}}$, and thus $\sqrt{N+1}(1 + C_\rho/2) KC(1 + r + r^2)$-Lipschitz with respect to $t$ for $\|\cdot\|_F$. Note that by (3.1),

$$\Delta(H, t) := \|T_\rho(H, t) - T_\nu(H, t)\|_F$$

$$= \|\int_0^\top \dot{T}_\rho(H, s) - \dot{T}_\nu(H, s) ds\|_F$$

$$= \|\int_0^\top \int_\beta g(T_\rho(H, s), \beta) \rho(\beta, t) d\beta ds - \int_0^\top \int_\beta g(T_\nu(H, s), \beta) \nu(\beta, s) d\beta ds\|_F$$

$$\le \underbrace{\int_0^\top \int_\beta \|g(T_\rho(H, s), \beta) - g(T_\nu(H, s), \beta)\|_F \rho(\beta, s) d\beta ds}_{J_1}$$

(F.47)

$$+ \underbrace{\|\int_0^\top \int_\beta g(T_\nu(H, s), \beta)\left(\rho - \nu\right)(\beta, s) d\beta ds\|_F}_{J_2}$$

We then bound $J_1$ and $J_2$ using the following two lemmas separately. Firstly, since $\|T_\rho\|_F \le \sqrt{N+1}C$ and $\|T_\nu\|_F \le \sqrt{N+1}C$, we have by Assumption 2 that

$$\|g(T_\rho(H, s), \beta) - g(T_\nu(H, s), \beta)\|_F \le \left(\sup_{\|T\|_F \le \sqrt{N+1}C} \|\nabla_{\text{vec}[T]} \text{vec}[g(T, \beta)]\|_2\right) \|T_\rho(H, s) - T_\nu(H, s)\|_F$$

$$\le \phi_T(N, D, \sqrt{N+1}C)(1 + r + r^2)\Delta(H, s)$$

(F.48)

Therefore, by (F.48) we have

$$J_1 \leq \phi_T(N, D, \sqrt{N+1}C)(1 + C_\rho/2)(1 + r + r^2) \int_0^\top \Delta(H, s)ds. \tag{F.49}$$

Secondly, we aim to bound the integral $J_2$ given Assumption 2 and $\|T_\nu(H, t)\|_{2-\text{col}} \leq C$ on $t \in [0, 1]$. Again by Assumption 2 we have

$$\|\nabla_\beta \text{vec}[g(T_\nu(H, s), \beta)]\|_F \leq \sqrt{\sum_{i=1}^{N+1} \|\nabla_\beta g(T_\nu(H, s), \beta)_{:,i}\|_{2-\text{col}}^2} \leq \sqrt{N+1}\phi_P(C)(1+r) \tag{F.50}$$

$$\|\nabla_T \text{vec}[g(T_\nu(H, s), \beta)]\|_F \leq \phi_T(N, D, \sqrt{N+1}C)(1 + r + r^2) \tag{F.51}$$

Since $T_\nu(H, s)$ is $\sqrt{N+1}(1+C_\rho/2)KC(1+r+r^2)$-Lipschitz with respect to $s$ for $\|\cdot\|_F$ (as shown in (F.46)), by (F.51), we obtain that $g(T_\nu(H, s), \beta)$ is $\sqrt{N+1}(1+C_\rho/2)KC\phi_T(N, D, \sqrt{N+1}C)(1 + r + r^2)^2$-Lipschitz with respect to $s$. Thus, $g(T_\nu(H, s), \beta)$ is $C'$-Lipschitz with respect to $(s, \beta)$, where $C' = \sqrt{N+1}(1+C_\rho/2)KC\phi_T(N, D, \sqrt{N+1}C)(1+r+r^2)^2 + \sqrt{N+1}\phi_P(C)(1+r)$. This indicates, by the Kantorovich-Rubinstein Theorem (see Theorem 5.10 of [68], for example), that

$$J_2 = \|\int_0^1 \int_D g(T_\nu(H, s), \beta)(\rho - \nu)(\beta, s)d\beta ds\|_F \leq C'W_1(\rho, \nu). \tag{F.52}$$

Define $C^* := \max\{C', \phi_T(N, D, \sqrt{N+1}C)(1 + r + r^2)\}$. By combining (F.49) and (F.52), we have

$$\Delta(H, t) \leq C^* \int_0^\top \Delta(H, s)ds + C^*W_1(\rho, \nu). \tag{F.53}$$

Applying the Grönwall's inequality then shows

$$\Delta(H, t) \leq C^* \exp(C^*t)W_1(\rho, \nu). \tag{F.54}$$

Specifically, we have $\|T_\rho(H, 1) - T_\nu(H, 1)\|_F \leq C^* \exp(C^*)W_1(\rho, \nu)$. $\qquad\square$

## F.10   Proof of Lemma C.3

*Proof.* Let's consider a fixed $(\beta, t)$ pair within $P_r$. Given Lemma C.1, we establish $\|T_\rho(H, t)\|_{2-\text{col}} \leq B \exp(K(1 + A + A^2))$. Consequently, under Assumption 1, we deduce

$$\begin{aligned}\|g(T_\rho(H, t), \beta)\|_F &\leq \sqrt{N+1}\|g(T_\rho(H, t), \beta)\|_{2-\text{col}} \\ &\leq \sqrt{N+1}K\|T_\rho(H, t)\|_{2-\text{col}}(1 + r + r^2) \\ &\leq \sqrt{N+1}KB \exp(K(1 + A + A^2))(1 + r + r^2).\end{aligned} \tag{F.55}$$

On the other hand, from (C.14), we have

$$\|p_\rho(H, t)\|_F \leq |\text{Read}(T_\rho(H, 1)) - y(H)| \cdot \left\|\exp\left(\int_t^1 \int_\beta \nabla_{\text{vec}[T]}\text{vec}[g(T_\rho(H, s), \beta)\rho(\beta, s)d\beta ds\right)\right\|_2$$

$$\leq (B + B \exp(K(1 + A + A^2))) \exp\left(\int_t^1 \int_\beta \|\nabla_{\text{vec}[T]}\text{vec}[g(T_\rho(H, s), \beta)]\| \rho(\beta, s)d\beta ds\right)$$

$$\leq (B + B \exp(K(1 + A + A^2))) \exp\left(\int_t^1 \int_\beta \|\nabla_{\text{vec}[T]}\text{vec}[g(T_\rho(H, s), \beta)]\| \rho(\beta, s)d\beta ds\right)$$

$$\leq (B + B \exp(K(1 + A + A^2))) \exp\left(\int_0^1 \int_\beta \phi_T(N, D, \|T_\rho(H, s)\|_F)(1 + \|\beta\| + \|\beta\|^2)\rho(\beta, s)d\beta ds\right)$$

$$\leq (B + B \exp(K(1 + A + A^2))) \exp\left(\phi_T(N, D, \sqrt{N+1}KB \exp(K(1 + A + A^2))(1 + A + A^2)\right) \tag{F.56}$$

Combining (F.55) and (F.56), we could obtain

$$\begin{aligned}\left\|\frac{\delta Q}{\delta\rho}(\beta, t)\right\| &\leq \mathbb{E}_\mu\|g(T_\rho(H, t), \beta)\|_F\|p_\rho(H, t)\|_F + \frac{\lambda}{2}r^2 \\ &\leq \sqrt{N+1}KB \exp(K(1 + A + A^2))(1 + r + r^2)(B + B \exp(K(1 + A + A^2))) \\ &\quad \exp\left(\phi_T(N, D, \sqrt{N+1}KB \exp(K(1 + A + A^2))(1 + A + A^2)\right) + \frac{\lambda}{2}r^2.\end{aligned}$$

$$\square$$

## F.11 Proof of Lemma C.4

*Proof.* By employing the Cauchy-Schwarz inequality, we readily observe that $\frac{1}{ML}\sum_t\sum_{j=1}^M\|\beta\| \leq A$. Consequently, leveraging (2.4) and Assumption 2, we ascertain that for any $t = 0, \Delta t, \ldots, (L-1)\Delta t$, we obtain:

$$\|\widehat{T}_\Theta(H, t + \Delta t/2)\|_{2-\mathrm{col}} \leq \|\widehat{T}_\Theta(H, t)\|_{2-\mathrm{col}}\Big\{1 + \frac{1}{2ML}\sum_{j=1}^M K(1 + \|\theta_{t,j}\| + \|\theta_{t,j}\|^2)\Big\}$$

$$\|\widehat{T}_\Theta(H, t + \Delta t)\|_{2-\mathrm{col}} \leq \|\widehat{T}_\Theta(H, t + \Delta t/2)\|_{2-\mathrm{col}}\Big\{1 + \frac{1}{2ML}\sum_{j=1}^M K(1 + \|w_{t,j}\| + \|w_{t,j}\|^2)\Big\}.$$

(F.57)

Therefore, by applying (F.57) multiple times, we obtain that for any $t = 0, \Delta t/2, \ldots, (L-1/2)\Delta t, 1$,

$$\|\widehat{T}_\Theta(H, t)\|_{2-\mathrm{col}} \leq \|\widehat{T}_\Theta(H, 0)\|_{2-\mathrm{col}}\prod_t\Big\{1 + \frac{1}{2ML}\sum_{j=1}^M K(1 + \|\theta_{t,j}\| + \|\theta_{t,j}\|^2)\Big\}\Big\{1 + \frac{1}{2ML}\sum_{j=1}^M K(1 + \|w_{t,j}\| + \|w_{t,j}\|^2)$$

$$\leq \|H\|_{2-\mathrm{col}}\exp\Big(\frac{K}{ML}\sum_t\sum_{j=1}^M 1 + \|\beta_{t,j}\| + \|\beta_{t,j}\|^2\Big)$$

$$\leq \|H\|_{2-\mathrm{col}}\exp\Big(K(1 + A + A^2)\Big).$$

$\square$

## F.12 Proof of Lemma C.5

*Proof.* Lemma C.4 shows that $\|\widetilde{T}_\Theta(H, t)\|_{2-\mathrm{col}}$ and $\|\widetilde{T}_{\widetilde\Theta}(H, t)\|_{2-\mathrm{col}}$ are bounded by $B_T := B\exp(K(1 + A + A^2))$ for any $H$ and $t \in [0, 1]$. From (2.4), for any $t = 0, \ldots, (L-1)\Delta t$, from Assumption 2 (ii) and (iii) we have

$$\|\widehat{T}_\Theta(H, t + \Delta t/2) - \widehat{T}_{\widetilde\Theta}(H, t + \Delta t/2)\|_F$$

$$\leq \|\widehat{T}_\Theta(H, t) - \widehat{T}_{\widetilde\Theta}(H, t)\|_F + \frac{\Delta t/2}{M}\sum_{j=1}^M\|f(\widehat{T}_\Theta(H, t), \theta_{t,j}) - f(\widehat{T}_{\widetilde\Theta}(H, t), \widetilde\theta_{t,j})\|_F$$

$$\leq \|\widehat{T}_\Theta(H, t) - \widehat{T}_{\widetilde\Theta}(H, t)\|_F + (\Delta t/2)\sqrt{N+1}\phi_P(B_T)(1 + r)M^{-1}\sum_{j=1}^M\|\theta_{t,j} - \widetilde\theta_{t,j}\| \quad \text{(F.58)}$$

$$+ (\Delta t/2)\phi_T(N, D, \sqrt{N+1}B_T)(1 + r + r^2)\|\widehat{T}_\Theta(H, t) - \widehat{T}_{\widetilde\Theta}(H, t)\|_F$$

$$\leq \|\widehat{T}_\Theta(H, t) - \widehat{T}_{\widetilde\Theta}(H, t)\|_F(1 + C_1(\Delta t/2)) + (\Delta t/2)C_2 M^{-1}\sum_{j=1}^M\|\theta_{t,j} - \widetilde\theta_{t,j}\|.$$

for some constant $C_1$ and $C_2$ depending only $N, D, r$ and assumptions. Similarly, we have

$$\|\widehat{T}_\Theta(H, t + \Delta t) - \widehat{T}_{\widetilde\Theta}(H, t + \Delta t)\|_F \leq \|\widehat{T}_\Theta(H, t + \Delta t/2) - \widehat{T}_{\widetilde\Theta}(H, t + \Delta t/2)\|_F(1 + C_1(\Delta t/2))$$

$$+ (\Delta t/2)C_2 M^{-1}\sum_{j=1}^M\|w_{t,j} - \widetilde w_{t,j}\|.$$

(F.59)

Combining (F.58) and (F.59), we derive

$$\|\widehat{T}_\Theta(H, t + \Delta t) - \widehat{T}_{\widetilde\Theta}(H, t + \Delta t)\|_F \leq \|\widehat{T}_\Theta(H, t) - \widehat{T}_{\widetilde\Theta}(H, t)\|_F(1 + C_3\Delta t) + \Delta t C_3 M^{-1}\sum_{j=1}^M\|\beta_{t,j} - \widetilde\beta_{t,j}\|.$$

(F.60)

where $C_3$ is a constant depending solely on $N, D, r$, and the parameters of the assumptions. Iterating (F.60) multiple times yields

$$\|\widehat{T}_\Theta(H, t) - \widehat{T}_{\widetilde\Theta}(H, t)\|_F \leq \exp(C_3)\frac{1}{ML}d(\Theta, \widetilde\Theta)$$

for any $t = 0, \Delta t, \ldots, 1$. $\qquad\qquad\qquad\qquad\qquad\qquad\qquad\qquad\qquad\qquad\qquad\qquad$ □

## F.13 Proof of Lemma C.6

*Proof.* By verifying that $\Theta$ satisfies the conditions outlined in Lemma C.4, we establish $\|\widehat{T}_\Theta(H, t)\|_{2-\mathrm{col}} \leq B \exp(K(1 + A + A^2))$. The first two results stem from Assumption 2 (i) and (ii), with recognition that $\|T\|_F \leq \sqrt{N+1}\|T\|_{2-\mathrm{col}}$ for any $T \in \mathbb{R}^{D \times (N+1)}$. As for the third result, consider $t = 0, \Delta t, \ldots, (L-1)\Delta t, 1$, where

$$
\max\{\|\widehat{p}_\Theta(H, t)\|_F, \|\widehat{p}_\Theta(H, t + \Delta t/2)\|_F\}
$$

$$
= \max\{\|\mathrm{vec}[\widehat{p}_\Theta(H, t)]\|, \|\mathrm{vec}[\widehat{p}_\Theta(H, t + \Delta t/2)]\|\}
$$

$$
\leq |\mathrm{Read}[\widehat{T}_\Theta(H, 1)] - y(H)|
$$

$$
\Big\{ \prod_{\substack{(s-t)/\Delta t+1 \in [(1-t)/\Delta t] \\ j \in [M]}} \Big\| I_{\dim\theta} + (\Delta t/2)\nabla_{\mathrm{vec}[T]}\mathrm{vec}[f(\widehat{T}_\Theta(H, s), \theta_{s,j})] \Big\|
$$

$$
\prod_{\substack{(s-t)/\Delta t+1 \in [(1-t)/\Delta t] \\ j \in [M]}} \Big\| I_{\dim w} + (\Delta t/2)\nabla_{\mathrm{vec}[T]}\mathrm{vec}[h(\widehat{T}_\Theta(H, s + \Delta t/2), w_{s,j})] \Big\| \Big\}
$$

$$
\leq |\mathrm{Read}[\widehat{T}_\Theta(H, 1)] - y(H)|
$$

$$
\prod_{\substack{(s-t)/\Delta t+1 \in [(1-t)/\Delta t] \\ j \in [M]}} \Big( 1 + (\Delta t/2)\Big\| \nabla_{\mathrm{vec}[T]}\mathrm{vec}[f(\widehat{T}_\Theta(H, s), \theta_{s,j})] \Big\| \Big)
$$

$$
\Big( 1 + (\Delta t/2)\Big\| \nabla_{\mathrm{vec}[T]}\mathrm{vec}[h(\widehat{T}_\Theta(H, s + \Delta t/2), w_{s,j})] \Big\| \Big) \tag{F.61}
$$

$$
\leq (B + B_T)
$$

$$
\exp\Big( (\Delta t/2) \sum_{(s-t)/\Delta t+1 \in [(1-t)/\Delta t]} M^{-1} \sum_{j=1}^M \Big\| \nabla_{\mathrm{vec}[T]}\mathrm{vec}[f(\widehat{T}_\Theta(H, s), \theta_{s,j})] \Big\|
$$

$$
+ (\Delta t/2) \sum_{(s-t)/\Delta t+1 \in [(1-t)/\Delta t]} M^{-1} \sum_{j=1}^M \Big\| \nabla_{\mathrm{vec}[T]}\mathrm{vec}[h(\widehat{T}_\Theta(H, t), w_{s,j})] \Big\| \Big)
$$

$$
\leq (B + B_T) \exp\Big( \phi_T(N, D, \sqrt{N+1}B_T) \frac{1}{ML} \sum_t \sum_{j=1}^M (1 + \|\beta\| + \|\beta\|^2) \Big)
$$

$$
\leq (B + B_T) \exp\Big( \phi_T(N, D, \sqrt{N+1}KB_T)(1 + A + A^2) \Big),
$$

where the first inequality arises from the fact that the matrix 2-norm is greater equal than the norm of any of its columns, and the fourth inequality follows from Assumption 2 (iii). $\qquad\qquad$ □

## F.14 Proof of Lemma C.7

*Proof.* Note that $\|\beta_{t,j}\| \leq r$ for any $t = 0, \Delta t, \ldots, (L-1)\Delta t$ and $j \in [M]$ with its expectation denoted as $\int_\beta \|\beta_{t,j}\|\rho(\beta, t)d\beta$. Applying Hoeffding's inequality yields, for any $z > 0$ and $t = 0, \Delta t, \ldots, (L-1)\Delta t$

$$
\mathbb{P}(|M^{-1}\sum_{j=1}^M \|\beta_{t,j}\|^2 - \int_\beta \|\beta\|^2 \rho(\beta, t)d\beta| \geq z) \leq 2\exp(-\frac{z^2}{2r^2}M).
$$

By applying the union bound over $t = 0, \Delta t, \ldots, (L-1)\Delta t$, the inequality above implies

$$
\mathbb{P}(|\frac{1}{ML}\sum_t \sum_{j=1}^M \|\beta_{t,j}\|^2 - \frac{1}{L}\sum_t \int_\beta \|\beta\|^2 \rho(\beta, t)d\beta| \geq z) \leq \mathbb{P}(\sup_t |M^{-1}\sum_{j=1}^M \|\beta_{t,j}\|^2 - \int_\beta \|\beta\|^2 \rho(\beta, t)d\beta| \geq z)
$$

$$
\leq 2L\exp(-\frac{z^2}{2r^2}M). \tag{F.62}
$$

In addition, we have

$$|\frac{1}{L}\sum_t \int_\beta \|\beta\|^2 \rho(\beta,t)d\beta - \int_0^1 \int_\beta \|\beta\|^2 \rho(\beta,t)d\beta dt| \leq \sup_{|t-s|\leq \Delta t} |\int_\beta \|\beta\|^2 \rho(\beta,t)d\beta - \int_\beta \|\beta\|^2 \rho(\beta,s)d\beta|$$

$$\leq r^2 L^{-1} \sup_{|t-s|\leq \Delta t} \|\rho(\cdot,t) - \rho(\cdot,s)\|_{\mathrm{BL}}$$

$$\leq r^2 C_\rho L^{-1}.$$

(F.63)

Combining (F.62) and (F.63), and setting $z = r\sqrt{2M^{-1}(\delta + \log(2L))}$ completes the proof. $\quad\square$

### F.15 Proof of Lemma C.8

*Proof.* The proof will be trivial by noting the equality

$$\prod_{l=1}^L A_l - \prod_{l=1}^L B_l = \sum_{l=1}^L \Big(\prod_{s=1}^{l-1} B_s(A_l - B_l) \prod_{s=l+1}^L A_s\Big).$$

Hence, we have

$$\|\prod_{l=1}^L A_l - \prod_{l=1}^L B_l\| \leq \sum_{l=1}^L \|\prod_{s=1}^{l-1} B_s\| \cdot \|A_l - B_l\| \cdot \|\prod_{s=l+1}^L A_s\| \leq C\sum_{l=1}^L \|A_l - B_l\|.$$

$\square$

## G  Assumption verification for a concrete example

In this section, we consider
$$f(Z,\theta) = VZ\mathrm{softmax}(Z^\top WZ) \tag{G.1}$$
with the collection of parameters $\theta = \mathrm{vec}[V,W]$, where softmax denotes the column-wise softmax function. Moreover, consider
$$h(Z,w) = W_2\mathrm{HuberizedReLU}(W_1 H) \tag{G.2}$$
with the collection of parameters $w = \mathrm{vec}[W_1, W_2]$. Here, HuberizedReLU denotes the entry-wise HuberizedReLU activation function defined as

$$\mathrm{HuberizedReLU}(x) = \begin{cases} 0, & \text{if } z \leq 0; \\ z^2/2, & \text{if } z \in [0,1]; \\ z - 1/2, & \text{if } z \geq 1. \end{cases}$$

Then, we can consider a Transformer model defined by equations (2.1), (2.2), and (2.4) in the paper, where the functions $f$ and $h$ are specified above. We suppose that this Transformer model is applied to a learning task with data that satisfies Assumption 1. We have the following proposition.

**Proposition G.1.** *Consider the Transformer model defined by equations* (2.1)*,* (2.2)*, and* (2.4)*, with $f(Z,\theta)$ and $h(Z,w)$ defined in* (G.1) *and* (G.2) *respectively. Then Assumptions 2-4 all hold.*

*Proof.* We omit the detailed derivations for the function $h(Z,w)$, which corresponds to the MLP part, in our verification of Assumptions 2 and 3, as $h(Z,w)$ satisfies Assumptions 2 and 3 is relatively more intuitive, especially given the proofs for $f(Z,\theta)$.

Denote $Z = (z_1,\ldots,z_{N+1}) \in \mathbb{R}^{D\times N+1}$. Then the function $f$ can be rewritten as

$$f(Z,\theta) = VZ\mathrm{softmax}(Z^\top WZ) = (f(Z,\theta)_{:,i})_{1\leq i\leq N+1},$$

where $f(Z,\theta)_{:,i} = \sum_{j=1}^{N+1} P_{ij}Vz_j$ and $P_{i,:} = \mathrm{softmax}(Z^\top Wz_i)$. Next, we calculate the derivatives of $f(Z,\theta)_{:,i}$ with respect to $Z$ and $\theta$ as follows:

**For** $Z$: the Jacobian $J \in \mathbb{R}^{(N+1)D \times (N+1)D}$ is $J = (J_{ij})_{1 \le i,j \le N+1}$, where $J_{ij} = \frac{\partial f_{:,i}}{\partial z_j}(Z, \theta) \in \mathbb{R}^{D \times D}$. After calculation, we obtain

$$J_{ij} = VZQ_i[E_{ji}Z^\top W + Z^\top W^\top \delta_{ij}] + P_{ij}V,$$

where $Q_i := \mathrm{diag}(P_{i:}) - P_{i:}^\top P_{i:}$, $E_{ij}$ is the matrix with zeros everywhere except one the $(i,j)$-th entry, and $\delta_{ij}$ is the Kronecker delta (1 if $i = j$, 0 otherwise).

**For** $\theta$: Define $A_i = Z^\top W z_i$. After calculation, we have

$$\frac{\partial P_{ij}}{\partial A_{kl}} = P_{ij}(\delta_{ik} - P_{il}), \quad \frac{\partial A_{ij}}{\partial W_{kl}} = Z_{ki}Z_{jl} \tag{G.3}$$

Thus, by the chain rule, we have

$$\nabla_{W_{kl}} f(Z, \theta)_{:,i} = \sum_{j=1}^{N+1} Z_{ki}Z_{jl}P_{ij}(\delta_{ik} - P_{il})V z_j. \tag{G.4}$$

Moreover, we have

$$\nabla_{\mathrm{vec}[V]} f(Z, \theta)_{:,i} = \sum_{j=1}^{N+1} P_{ij}(z_j^\top, \ldots, z_j^\top)^\top, \tag{G.5}$$

where $(z_j^\top, \ldots, z_j^\top)^\top$ contains $D$ copies of $z_j$. We then verify the assumptions one by one.

For Assumption 2 (i), we have

$$
\begin{aligned}
\|f(T, \theta)\|_{2-\mathrm{col}} = \|VT\mathrm{softmax}(T^\top WT)\|_{2-\mathrm{col}} &\le \|V\|_2 \cdot \|T\|_2 \cdot \|\mathrm{softmax}(T^\top WT)\|_{2-\mathrm{col}} \\
&\le \|\theta\|_2 \cdot \|T\|_{2-\mathrm{col}} \cdot \|\mathrm{softmax}(T^\top WT)\|_{1-\mathrm{col}} \\
&\le \|\theta\|_2 \cdot \|T\|_{2-\mathrm{col}},
\end{aligned}
$$

where the second-to-the-last inequality follows by the fact that $\ell_2$-norm can be upper bounded by the $\ell_1$-norm, and the last inequality follows by the fact that each column of the softmax output has an $\ell_1$-norm equaling one. Therefore, the first condition in Assumption 2 with $K = 1$ is verified for the function $f$ in (G.1).

For $h$ in (G.2), we have

$$
\begin{aligned}
\|h(T, w)\|_{2-\mathrm{col}} = \|W_2\mathrm{HuberizedReLU}(W_1 T)\|_{2-\mathrm{col}} &\le \|W_2\|_2 \cdot \|\mathrm{HuberizedReLU}(W_1 T)\|_{2-\mathrm{col}} \\
&\le \|W_2\|_2 \cdot \|W_1 T\|_{2-\mathrm{col}} \\
&\le 2 \cdot \|w\|_2^2 \cdot \|T\|_{2-\mathrm{col}},
\end{aligned}
$$

where the second inequality follows by the property of HuberizedReLU that $|\mathrm{HuberizedReLU}(x)| \le |x|$. This demonstrates that Assumption 2 (i) with $K = 1$ holds for $h$ in (G.2) as well.

For Assumption 2 (ii), (G.5) leads to

$$\|\nabla_{\mathrm{vec}[V]} f(T, \theta)_{:,i}\|_2 \le \sum_{j=1}^{N+1} \|P_{ij}T_{:,j}^\top\|_2 \le \sum_{j=1}^{N+1} P_{ij}\|T_{:,j}\|_2 \le \|T\|_{2-\mathrm{col}}.$$

Moreover, (G.4) leads to

$$
\begin{aligned}
\left\|\nabla_{\mathrm{vec}[W]} f(T, \theta)_{:,i}\right\|_2 &\le \sum_{1 \le k,l \le D} \sum_{j=1}^{N+1} \|Z_{ki}Z_{jl}P_{ij}(\delta_{ik} - P_{il})VT_{:,j}\|_2 \\
&\le 2 \sum_{1 \le k,l \le D} \sum_{j=1}^{N+1} P_{ij}\|T_{ki}T_{jl}VT_{:,j}\|_2 \\
&\le 2 \max_{1 \le j \le N+1} \sum_{1 \le k,l \le D} \|T_{ki}T_{jl}VT_{:,j}\|_2 \\
&\le 2 \max_{1 \le j \le N+1} \sum_{1 \le k,l \le D} |T_{ki}T_{jl}|\|V\|_2\|T_{:,j}\|_2 \\
&\le 2\|T\|_{2-\mathrm{col}}^2\|\theta\|_2
\end{aligned}
$$

Combining the two equations above gives Assumption 2 (ii) with $\phi_P(\|T\|_{2-\text{col}}) = \|T\|_{2-\text{col}} + 2\|T\|_{2-\text{col}}^2$.

For Assumption 2 (iii), we have

$$\|J\|_2 \leq \sqrt{\sum_{1 \leq i,j \leq N+1} \|J_{ij}\|_2^2} \leq (N+1) \max_{1 \leq i,j \leq N+1} \|J_{ij}\|_2.$$

For any $1 \leq i,j \leq N+1$, we have

$$\begin{aligned}
\|J_{ij}\|_2 &\leq P_{ij}\|V\|_2 + \|T\|_2\|Q_i\|_2\|E_{ji}T^\top W + T^\top W^\top \delta_{ij}\|_2\|V\|_2 \\
&\leq \|V\|_2\left(1 + 2\|T\|_2^2\|W\|_2\right) \\
&\leq \|\theta\|_2 + 2\|T\|_F^2\|\theta\|_2^2 \\
&\leq (1 + 2\|T\|_F^2)(1 + \|\theta\|_2 + \|\theta\|_2^2).
\end{aligned}$$

Hence, we have

$$\|J\|_2 \leq 2N\|T\|_F^2 \cdot (N+1)(1 + 2\|T\|_F^2)(1 + \|\theta\|_2 + \|\theta\|_2^2).$$

The above equation demonstrates that for $f$, Assumption 2 (iii) holds with $\phi_T(N, D, \|T\|_F) = (N+1)(1 + 2\|T\|_F^2)$. We have verified Assumption 2 for the attention layer encoder $f$. The verification for $h$ is similar and easier.

Next, we verify Assumption 3. Given that we are currently considering the example where the encoder employs a smooth univariate activation function, we can prove stronger results by removing the expectation $\mathbb{E}_\mu$.

(i) and (iii): Given the calculation of derivatives in (G.4) and (G.5) we have presented above, we first show that
$$P_{ij} = \text{softmax}(Z^\top W z_i)$$
is locally Lipschitz continuous with respect to $Z$ and $\theta$. By (G.3) and the chain rule, we can derive that

$$\nabla_{W_{kl}} P_{ij} = \sum_{j=1}^{N+1} Z_{ki}Z_{jl}P_{ij}(\delta_{ik} - P_{il}), \quad \frac{\partial P_{ij}}{\partial z_j} = Q_i[E_{ji}z_j^\top W + z_j^\top W^\top \delta_{ij}].$$

and the local Lipschitz continuity is then obvious given the boundedness of $\nabla_{W_{kl}} P_{ij}$ and $\frac{\partial P_{ij}}{\partial z_j}$ with respect to necessary parameters.

As we prove that $P_{ij}$ is locally Lipschitz continuous, given that then each component in (G.4) and (G.5) is locally Lipschitz continuity with respect to both $Z$ and $\theta$, and is obviously bounded by an increasing function of $N, D, \|\theta\|, K_T, L_T$. Then the local Lipschitz continuity is straightforward as they are all sufficiently smooth.

(ii) and (iv): Because the norm of the difference of two Jacobian matrices $\|J^1 - J^2\|_2$ is bounded by $\sqrt{\sum_{1 \leq i,j \leq N+1} \|J_{ij}^1 - J_{ij}^2\|_2^2}$, it suffices to show that $J_{ij}$ is locally Lipschitz continuous with respect to both $\theta$ and $Z$. Again each component of $J_{ij}$ that depends on $Z$ or $\theta$, i.e. $Z, Q_i, W, P_{ij}$, is bounded by an increasing function of $N, D, K_P, L_T, K_T, \|\theta\|$, and is locally Lipschitz continuous given sufficient smoothness. Hence, (ii) and (iv) also hold.

For Assumption 4, we consider the pair $(g, \alpha) = (h, w)$, and the partition $\alpha = (\alpha_1, \alpha_2)$ with $\alpha_1 = W_2, \alpha_2 = W_1$. We also let a compact set $\mathcal{K} = \{W_1 : \|W_1\| \leq 1\}$. Then Assumption 4 (i) on the partial 1-homogeneity property straightforwardly holds:

$$h(T, W_1, c \cdot W_2) = c \cdot W_2\text{HuberizedReLU}(W_1 H) = c \cdot h(T, W_1, W_2).$$

Regarding Assumption 4 (ii) on the universal kernel property, we first note that according to the choice $(g, \alpha) = (h, w)$, this assumption is purely an assumption on the MLP part of the Transformer. Here we give the detailed proof as follows.

First of all, according to the classic universal approximation theory (see the wiki page of "universal approximation theorem" and [34, 18, 57] for more details), we know that two-layer fully-connected

networks with non-polynomial activation functions and without any constraints on its parameters are universal approximates.

Therefore, we know that the function class $\mathrm{span}\{W_2\mathrm{ReLU}^2(W_1T) : W_1 \in \mathbb{R}^{\dim(W_1)}, W_2 \in \mathbb{R}^{\dim(W_2)}\}$ is dense in $\mathcal{C}(\|T\|_{2-\mathrm{col}} \leq B, \mathbb{R}^{D \times (N+1)})$. Moreover, by the definition of Huberize-dReLU, for any $B > 0$ and any $\widehat{W}_1, \widehat{W}_2$, there exist small constant $c$ such that $c \cdot \widehat{W}_1 \in \mathcal{K}$, $c \cdot \|\widehat{W}\|_1 \leq B^{-1}$, and

$$
\begin{aligned}
c^{-2} \cdot \widehat{W}_2 \mathrm{HyberizedReLU}(c \cdot \widehat{W}_1 T) &= c^{-2} \cdot \widehat{W}_2 \mathrm{ReLU}^2(c \cdot \widehat{W}_1 T) \\
&= c^2 \cdot c^{-2} \cdot \widehat{W}_2 \mathrm{ReLU}^2(\widehat{W}_1 T) \\
&= \widehat{W}_2 \mathrm{ReLU}^2(\widehat{W}_1 T),
\end{aligned}
$$

where the second equation follows by the positive 2-homogeneity of $\mathrm{ReLU}^2$ activation. This implies that

$$\{W_2\mathrm{ReLU}^2(W_1T) : W_1 \in \mathbb{R}^{\dim(W_1)}, W_2 \in \mathbb{R}^{\dim(W_2)}\} \subseteq \{W_2\mathrm{HyberizedReLU}(W_1T) : W_2 \in \mathbb{R}^{\dim(W_2)} \times \mathcal{K}\}.$$

Therefore, we conclude that $\mathrm{span}\{W_2\mathrm{HyberizedReLU}(W_1T) : W_2 \in \mathbb{R}^{\dim(W_2)} \times \mathcal{K}\}$ is dense in $\mathcal{C}(\|T\|_{2-\mathrm{col}} \leq B, \mathbb{R}^{D \times (N+1)})$. This finishes the validation of Assumption 4. $\qquad\square$

## H  Experiments

As discussed in Sections 3 and 4, our mean-field approximation results and global convergence results are asymptotic guarantees requiring exponentially large number of heads $M$ and number of layers $L$. Such results are due to the nature of mean-field type analysis. In practice, we frequently observe that global convergence can be achieved by Transformer models of reasonable sizes. In this section, we run simple experiments on training Vision Transformers (ViT) [24] on the CIFAR-10 datasets to demonstrate global convergence in practical applications.

We train Vision Transformers with different numbers of heads and layers. In all our experiments, we split each CIFAR-10 image into four patches and then pass the patches into Vision Transformer models. We keep the dimension of each attention head to be 128. The output of each self-attention layer is passed through a single-hidden-layer feedforward component with 128 hidden neurons and GeLU activation. Both the self-attention and feedforward components include skip connections. We implement dropout in the self-attention layers as well as the feedforward layers with a dropout probability of 0.1. The model is attached to a linear classifier.

In all experiments, we train the ViT models using Adam for 200 epochs with a mini-batch size 512. We set the initial learning rate to be $1e - 4$, and implement a cosine annealing learning rate schedule. We do not use any data augmentation or explicit regularization techniques, so that global convergence for large enough models implies close-to-zero training loss and close to $100\%$ training accuracy.

In the first set of experiments, we fix the depth of the ViT to 6 layers (i.e., there are six self-attention layers, each followed by a single-hidden-layer feedforward component). We train such Vision Transformers with the numbers of heads per layer ranging from 4 to 40, and record the training loss and training accuracy throughout training. The results are given in Figure 1. Based on the results, it is clear that for ViT models with more than 20 heads can achieve close-to-zero training loss and close to $100\%$ training accuracy, demonstrating global convergence on the CIFAR-10 training data.

In the second set of experiments, we fix number of heads in the ViT model per layer to 8. We train such Vision Transformers with depths ranging from 2 to 20, and record the training loss and training accuracy throughout training. The results are given in Figure 2. Again, the results indicate that ViT models with more than 16 layers can achieve close-to-zero training loss and close to $100\%$ training accuracy on the CIFAR-10 dataset, implying global convergence.

We note that all these experiments are conducted on a standard GPU card. We can observe clear global convergence when the Vision Transformer is sufficiently wide or deep, but still within reasonable scales. This indicates that, although our theoretical guarantees require extremely large numbers of heads and layers due to the limitations of the mean-field technical tools, global convergence can be achieved by Transformers of reasonable sizes in practice.

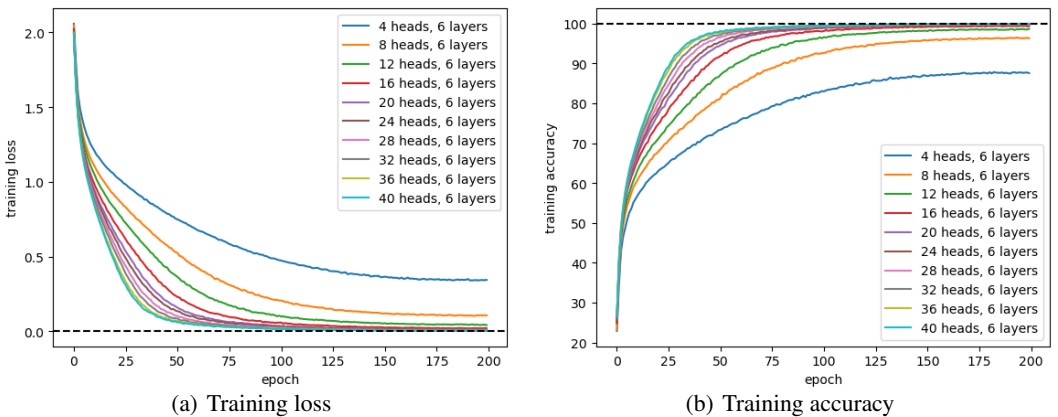

Figure 1: Training loss and training accuracy of Vision Transformers with different numbers of heads. (a) gives the curves of training loss, while (b) gives the curves of training accuracy.

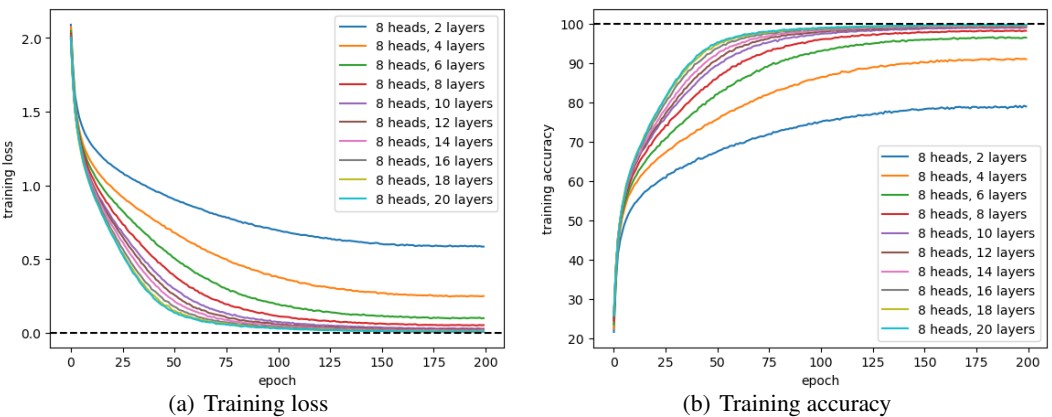

Figure 2: Training loss and training accuracy of Vision Transformers with different depths. (a) gives the curves of training loss, while (b) gives the curves of training accuracy.

