# OpenReview forum: "Global Convergence in Training Large-Scale Transformers"
_NeurIPS.cc/2024/Conference — NeurIPS 2024 poster_

### Official Review · Reviewer_mtEE · 2024-06-26

**Soundness:** 4
**Presentation:** 2
**Contribution:** 2
**Rating:** 5
**Confidence:** 3

**Summary:**

The paper considers the theoretical mean-field limit of Transformers where width and depth go to infinity and studies the approximation error and convergence properties of gradient flow with weight decay. Both a residual self-attention layer and a residual feedforward layer are approximated by an ODE which averages the two encoders, whose solution models distribution of the parameters of both blocks throughout the depth of the Transformer. It is shown that the dynamics of the discretized model approximates the Wasserstein gradient flow under some  regularity assumptions. Moreover, it is shown under partial homogeneity and universal kernel assumptions that if the Wasserstein flow weakly converges to a stationary distribution, then the discrete model must also converge with arbitrarily small risk.

**Strengths:**

* The paper is a nontrivial extension of existing mean-field analyses of two-layer neural networks and ResNets to Transformers, which is challenging due to the existence of both feedforward and attention layers, the latter of which is new in the literature.
* The assumptions for the networks and input data are quite general and can encompass e.g. the softmax mechanism and in-context learning settings.
* The analyses and techniques are sufficiently novel and detailed and serve as an initial characterization of the dynamics of large scale Transformers, in particular the global convergence analysis of Section 4.

**Weaknesses:**

* The obtained error bounds (Theorem 3.1) are so large that they likely have little practical relevance beyond the initial moments of training. In particular, the hidden constants are in the worst case super-super-exponential with respect to the time horizon. Specifically, the bound is exponential in $\phi_T(N,D,C\times B_\tau)$ where $B_\tau$ is exponential in $R_\tau$, which in turn is exponential in $\tau$. While bounds exponentially diverging in time are common in the literature (which is a priori expected from an ODE discretization argument) as mentioned in  the paper, the dependency in Theorem 3.1 is much worse and some effort needs to be made to at least justify this. For example, one exponentation seems to be removable if e.g. $\phi_T$ is bounded. Currently the dependency in the hidden constants are not made obvious without going through the proofs in the Appendix, which I feel should be addressed more transparently.
* The convergence analysis (Theorem 4.1) also suffers from the same problem. The analysis requires a time horizon $\tau_0$ large enough so that $W_2(\rho^{(\tau)}, \rho_\infty)$ is exponentially small (w.r.t. $R_\infty$) compared to the desired error $\epsilon$. This horizon is then fed into the approximation bound of Theorem 3.1, again yielding very large constants unless convergence is achieved in the very early stages of training. (This issue seems less critical since it only affects the $C_1$ term, although the $C_2\lambda$ term is still super-exponential in $R_\infty$.)
* The recurring rate $L^{-1}+\sqrt{\log L/M}$ is not shown to be tight, leading me to further question the utility of the provided bounds. Why is the scaling for $L,M$ different? How does the rate compare with the analysis for ResNets or other ODE systems with 'width' & 'depth' dimensions?
* A nitpick: the terms 'universal constant' or 'universal bound' are used quite frequently, however they should be reserved for constants that do not depend on *any* problem parameters.

**Questions:**

* Can the dependency on time horizon be alleviated with stronger model assumptions?
* Is the $L^{-1}+\sqrt{\log L/M}$ rate tight/expected? (see Weakness)
* The widths of both the feedforward layer and the attention layer are both set equal to $M$, however the number of heads cannot typically be expected to be very large compared to the width of feedforward layers. Does the analysis generalize to when they are different?
* Some recent papers [1,2] have also studied mean-field limits of Transformers from different perspectives, it would be nice to add a comparison in the related works section (although the latter is a contemporary work).

[1] https://openreview.net/forum?id=xm2lU7tteQ

[2] https://arxiv.org/abs/2405.15712

**Limitations:**

See Weaknesses.

---

> ### Author Rebuttal · Authors · 2024-08-07
>
> Thank you for your insightful comments and suggestions. They are extremely helpful in improving our work and presentation. Below, we address your concerns point by point.
>
> **Q1**: The error bounds in Theorem 3.1 are so large they may be impractical beyond initial training moments, with hidden constants possibly being super-super-exponential over time. The hidden constants' dependency is unclear without reviewing the Appendix proofs and should be addressed more transparently. Additionally, can stronger model assumptions alleviate the time horizon dependency? One exponentiation might be removable if $\phi_T$ is bounded.
>
> **A1**: We thank the reviewer for this insightful suggestion and careful clarification on this point. We agree that the hidden constants in Theorem 3.1 have sensitive dependencies on the time horizon, which is partially due to the structural complexities of the Transformers. Since Assumptions 2-3 are mild, allowing additional parameter norm factors in the bound, it is inevitable that the constant dependency would accumulate, thus easily causing super-exponential constants. Our goal is to take the first theoretical step in analyzing Transformers using mean-field tools. We can always make $L$ and $M$ sufficiently large to ensure vanishing approximation error. Though the result becomes asymptotic, our work lays the groundwork for future mean-field theory for Transformers.
>
> Yes, we could alleviate the dependency with stronger model assumptions, as $\phi_T$ may not depend on the Transformer output magnitude. We have noticed a lot of interest in identifying the optimal choice of $\phi_T$, i.e., the Lipschitz constant of the Jacobian matrix of the self-attention term. This is a particularly challenging frontier question. For instance, [R1] suggests that $\phi_T$ can be bounded by $\sqrt{N/D} + \mathrm{poly}(\|T\|_F)$, where $\mathrm{poly}(\cdot)$ denotes a polynomial function. In the context of $l_2$ self-attention, [R2] finds $\phi_T$ to be $\sqrt{N \log N / D}$, which notably does not depend on $\|T\|_F$, and [R3] demonstrates that for $l_1$ distance metrics in attention layers, $\phi_T$ could be $\sqrt{D \log N}$. We will include this illustration if our submission is accepted.
>
> [R1] Dasoulas, G., Scaman, K., & Virmaux, A. (2021). Lipschitz normalization for self-attention layers with application to graph neural networks. ICML
>
> [R2] Kim, H., Papamakarios, G., & Mnih, A. (2021). The lipschitz constant of self-attention. ICML.
>
> [R3] Vuckovic, J., Baratin, A., & Combes, R. T. D. (2020). A mathematical theory of attention. arXiv preprint arXiv:2007.02876.
>
> **Q2**: The convergence analysis in Theorem 4.1 also has issues with large constants. It requires a large time horizon $\tau_0$ to make $W_2(\rho^{(\tau)},\rho_\infty)$ exponentially small relative to the error $\epsilon$. This leads to large constants in Theorem 3.1's approximation bound unless early-stage convergence occurs. While this mainly affects the $C_1$ term, $C_2$ remains super-exponential in $R_\infty$.
>
> **A2**: We thank the reviewer for the crucial clarification on this point. We agree that $C_1$ heavily depends on the choice of $\tau_0$, and $\tau_0$ heavily depends on the prefixed $\epsilon$. Nevertheless, we can always make $L$ and $M$ sufficiently large (independent of the $\tau_0$ choice) to ensure the risk is asymptotically bounded. We will focus more on refining these constants in future work after current initial steps towards extending the architecture into the realm of Transformers.
>
> **Q3**: The rate $L^{-1} + \sqrt{\log L/M}$ is not shown to be tight. Why is the scaling for $L$ and $M$ different? How does this rate compare to the analysis for ResNets or other ODE systems with 'width' and 'depth' dimensions?
>
> **A3**: The different scaling for $L$ and $M$ also appears in the analysis for ResNets (see [20] Theorem 9), where $L$ depends linearly and $M$ depends quadratically on $\epsilon$. We think such rates are indeed expected. Intuitively, the different scaling is from the source of the approximation difference: $L^{-1}$ arises from the discretization of the ODE into $L$ small steps, and $M^{-1/2}$ comes from the implicit use of the Hoeffding’s Inequality considering the average of the outputs of $M$ nodes/heads. Furthermore, since our result in Theorem 3.1 considers the maximum difference across the entire time interval, we add an additional $\log L$ term as we apply the probability union bounds.
>
> **Q4**: The terms 'universal constant' or 'universal bound' are used quite frequently, however they should be reserved for constants that do not depend on any problem parameters.
>
> **A4**: We thank the reviewer for the careful reading. We will fix this issue in the revised version.
>
> **Q5**: The widths of both the feedforward layer and the attention layer are both set equal to, however the number of heads cannot typically be expected to be very large compared to the width of feedforward layers. Does the analysis generalize to when they are different?
>
> **A5**: We thank the reviewer for the great question. The results can be easily extended when the widths of the feedforward layer ($M_1$) and the attention layer ($M_2$) both go to infinity, and we conjecture that the main results still hold by replacing $M$ with $\min\{M_1, M_2\}$. However, by the nature of the mean-field analysis, we do need $M_2$ to be sufficiently large to achieve the theoretical convergence. The shift from discrete parameters to parameter distributions requires a large number of heads to provide a close approximation for the "average".
>
> **Q6**: Two recent papers referenced by the reviewer have also studied mean-field limits of Transformers from different perspectives. It would be nice to add a comparison.
>
> **A6**: We thank the reviewer for pointing out the related works. We will cite them and add discussions in the revision.

---

> > ### Comment · Reviewer_mtEE · 2024-08-07
> >
> > Thank you for the detailed reply. I will maintain my score as I still feel the technical issue of Q1 ends up weakening the relevance of the paper, however I personally do think the approach and ideas are very interesting.

---

### Official Review · Reviewer_accV · 2024-07-06

**Soundness:** 4
**Presentation:** 2
**Contribution:** 4
**Rating:** 7
**Confidence:** 4

**Summary:**

The authors theoretically investigate the mean field limit of residual transformer models. As depth and width go to infinity, thanks to the residual structure, the forward pass can be modeled as an ordinary differential equation, and the training gradient flow converges to a Wasserstein gradient flow. The author show well-posedness of this PDE, provide non-asymptotic time uniform bounds on quantities of interest and show global convergence to the global minimum of the empirical risk as $L,m/log(L) \to+\infty$ and $\lambda \to 0$.

**Strengths:**

The tackled problem is extremely relevant and the derived results are sound. Given the size of the work, the authors did a good job overall in organizing the contents.

**Weaknesses:**

1. The content is intense and the presentation doesn't help a novel reader. I believe polishing of the manuscript would improve the quality.
More concretely, I would add a small paragraph in the main manuscript summarizing the main ideas behind the novelties. In the appendix 			instead it is really easy to get lost in a long proof, loosing track of the main goal. I suggest to include a small paragraph at the beginning of 		each section with the main steps required for the final proof, to guide the reader and make even the technical part more accessible 			(something on the line of what the authors did in D.3 would be perfect).

2. While this did not and will not impact my score, for manuscripts of this size I believe at least a really small experimental section (even in the appendix) to test the claims would make the results more sound.

**Questions:**

On the technical side:
1. While I understand the need of assumption i) in theorem 4.1, it is not obvious to me how stringent it is. Already for an extremely simple problem like $Q(\rho) = \int \rho \log(\rho) + \lambda \int x^2 d\rho$ the minimizer is a Gaussian, that has unbounded support. 	Given the last example, the heuristic justification given by the authors seems to not be sufficient. Do the authors see a way to transfer this condition on the growth rate of $R(\Theta)$ at infinity?
2. The immediate extension of the deterministic case would be to consider the full Fokker-Plank, as it may emerge as a model of stochastic optimization. Already in the simplest case, the PDE under consideration would be $\partial_\tau \rho^{(\tau)} = div(\rho^{(\tau)} \nabla U) + \sigma \Delta \rho^{(\tau)}$. The stationary distribution in this case would be $\rho \propto e^{-U/\sigma}$, that has unbounded support for $U$ defined on all parameter space. This suggest that already in the simplest case (just pure isotropic diffusion added), the extension of the results may not be straightforward. I would appreciate the authors' comment on this, and I would like to know if they see an easy way to weaken that hypothesis as it's something one has no control on.
3. While mentioned in the proofs, it is not clear in the main manuscript what $\delta$ is. For example, in theorem 4.1 the statement is "with high probability", but it is not mentioned on what. I would add two words in the theorems/lemmas that are in high probability to specify with respect to what.
4. It sounds strange to me that theorem 3.1 gives a bound on a compact set, while theorem 4.1 on an unbonded time horizon, with what appears to be exactly the same constant. Is the constant in theorem 4.1 really the same of the one in 3.1 or it includes other terms?

**Limitations:**

The authors correctly assessed the limitations in the manuscript, I have no further suggestions.

---

> ### Author Rebuttal · Authors · 2024-08-07
>
> We are grateful for your supportive comments, which are greatly helpful for us to improve our work. We address your questions as follows.
>
> **Q1**: The content is intense and the reviewer suggests polishing the manuscript. Adding a summary of the main ideas in the manuscript and brief introductory paragraphs in each appendix section are helpful.
>
> **A1**: We thank the reviewer for this great suggestion. We agree that it would be helpful to briefly summarize the main ideas in the manuscript and the main proof ideas in the appendix. (If we can efficiently utilize the additional page for the camera-ready version, we may even add a brief proof sketch if space permits.) We will revise the paper following your suggestions.
>
> **Q2**: The reviewer believes that at least a really small experimental section (even in the appendix) to test the claims would make the results more sound.
>
> **A2**: In practice, we observe that using vision Transformers to fit simple datasets such as MNIST can easily achieve near-zero training loss, indicating convergence to a global minima. This observation reinforces our confidence that our results are valid and that our Transformer structure can minimize the training loss to zero as $L$ and $M$ increase. We will also conduct experiments investigating the relation between discrete transformers and their continuous limits and include the results in the revision.
>
> **Q3**: Assumption i) in Theorem 4.1 has ambiguity on how stringent it is without sufficient heuristic justification. A related example: the minimizer of $Q(\rho)=\int \rho\log(\rho) +\lambda x^2d\rho$ is a Gaussian that has unbounded support. Do the authors see a way to transfer this condition to the growth rate at infinity? The immediate extension of the deterministic case would be to consider the full Fokker-Planck, whose stationary point of the PDE has unbounded support.
>
> **A3**: We thank the reviewer for the insightful comments about the extra entropy and diffusion terms. Since these two questions are related, we have combined them into one and provided a comprehensive response.
>
> Yes, several papers [13,24,45] in the mean-field analysis literature study the same terms you mentioned, considering noisy gradient descent (NGD). By introducing an additional random Gaussian noise term $N(0,\lambda_2)$ into the gradient flow/descent formula, the PDE evolution of $\rho^{(\tau)}(\theta,w)$ becomes a diffusion process with an additional term $\nabla^2\rho^{(\tau)}(\theta,w)$ based on the Fokker-Planck equation. Consequently, the stationary point of the PDE is the local minimum of the regularized risk function with a regularization term $\int \rho\log(\rho) +\lambda x^2d\rho$. In this case, we note that the minimizer of $\tilde{Q}(\rho)$ will not have compact support. In comparison, (noiseless) gradient flow minimizes the risk function with $\int \lambda x^2d\rho$ regularization, and it remains possible that the minimizer still has compact support.
>
> While the extension to NGD is very interesting, after careful consideration, we believe this extension does not fit within the framework and assumptions of our paper. The key reason is that our Assumptions 1-3 are much milder than the corresponding assumptions in [13,24,45], as the constant terms in our assumptions also depend on the parameter norms to accommodate Transformer structures. Given the dependence on parameter norms (e.g., $(1+||\theta||)$ in Assumption 2(ii)), we must ensure that the parameter distribution is always bounded within any infinite time $s \in [0,\tau]$ to apply these assumptions. If we consider a simpler structure like residual networks with stronger assumptions, then the extension you mentioned could perfectly fit or even enhance the theoretical proof.
>
> Lastly, we hope to clarify the first assumption in Theorem 4.1 to alleviate concerns about its stringency. Although this assumption is uncheckable from the given assumptions, the weight decay regularization can penalize the parameter norms, potentially leading to a $\rho^{(\tau)}$ with compact support. Additionally, this assumption is more likely to hold for simpler learning tasks: if the true label generation process is defined by a $\rho^*$ with compact support, then a bounded solution suffices to minimize the loss for such a simple learning task.
>
> **Q4**: It is not clear in the main manuscript what $\delta$ is. The reviewer would add two words in the Theorems/lemmas that are in high probability to specify with respect to what in Theorem 4.1.
>
> **A4**: We thank the reviewer for the careful reading and the great suggestion. The probability is with respect to the parameter initialization $\Theta^{(0)}=\\{\theta^{(0)}\_{t,j},w^{(0)}\_{t,j}\\}_{t,j}$. We will clarify it in the revision.
>
> **Q5**: It sounds strange that Theorem 3.1 gives a bound on a compact set, while Theorem 4.1 on an unbonded time horizon, with what appears to be exactly the same constant. Is the constant in Theorem 4.1 really the same of the one in 3.1 or does it include other terms?
>
> **A5**: We thank the reviewer for the question and careful reading. The constant in Theorem 4.1 is indeed the same as that in Theorem 3.1, but with $\tau$ is fixed as $\tau_0$. To clarify how we obtain the constant in Theorem 4.1, we first select a large time horizon $\tau_0$ and apply Theorem 3.1 with respect to this specific choice to bound $|\widehat{R}^{(\tau_0)} - R(\rho^{(\tau_0)})|$. The results follow if we can show $\sup_{\tau \geq \tau_0} R(\rho^{(\tau)})$ is asymptotically smaller than $\epsilon + \lambda$. This is why $C_1$ is dependent on $\tau_0$ in Theorem 4.1. Further illustration can be found in the proof steps of Theorem 4.1 (pages 34-36), where $\tau $ being large means exactly $\tau \geq \tau_0$.

---

> > ### Comment · Reviewer_accV · 2024-08-07
> >
> > First of all, I would like to thank the authors for their thorough rebuttal.
> >
> > I am satisfied with the answers and I would like to keep my score, conditionally on the authors addressing Q1 in the revised version.

---

### Official Review · Reviewer_eZmg · 2024-07-09

**Soundness:** 3
**Presentation:** 3
**Contribution:** 3
**Rating:** 6
**Confidence:** 1

**Summary:**

This paper analyzes gradient flow on Transformer networks. It is shown that for wide and deep Transformers, gradient flow converges to the Wasserstein gradient flow and reaches a global minimum.

**Strengths:**

This is a well-written paper and it seems the results are strong and clean. However, I have very little background in this area and cannot verify the correctness of the statements. I hope the AC can find another qualified reviewer and delete this review if possible.

**Weaknesses:**

N/A

**Questions:**

N/A

---

> ### Author Rebuttal · Authors · 2024-08-07
>
> Thank you for your recognition of our paper. Your comments provide valuable guidance on our presentation. In this work, we aim to take the initial steps towards studying the theoretical optimization guarantees of Transformer models and, for the first time, prove the global convergence property via gradient descent-based methods.

---

### Official Review · Reviewer_oct4 · 2024-07-18

**Soundness:** 2
**Presentation:** 3
**Contribution:** 2
**Rating:** 5
**Confidence:** 3

**Summary:**

This paper studies transformers in their mean field limit. They use a ResNet architecture with infinite depth. The residual blocks are made of two steps: one transformer step and one standard MLP. They also use infinite width for both steps.
Then, they show that the gradient flow in this limit is well posed, by using the notion of Wasserstein gradient flows since the mean field model is defined on space of measures. They also show consistency with the original discrete transformer model.
Then, they study the global convergence properties of the gradient flow: they prove a sort of global convergence result, which requires some hypotheses.

**Strengths:**

The writing of the paper is quite clear.

Proving mean field convergence results has been studied a lot in the literature on theoretical deep learning. The result of the authors is interesting because they deal with the transformer architecture which ends up being a sum of two maps in the mean field representation. The authors can prove similar results in this context, which are new to the best of my knowledge of the literature.

**Weaknesses:**

One of the main weakness for a submission to NeurIPS is the format of the paper. The 9 pages of the main paper are devoted to a (well done) explanation of the main results. There is no concrete ideas of proof in the main paper. More than 40 pages of technicalities are necessary to validate their result.

The novelty of the results is the fact that transformer architecture is taken into account in the mean-field limit. However, the stated results do not show to what extent this architecture helps in obtaining the results. In fact, the transformer architecture appears more as something that hinders the standard proof on global convergence.
The paper does not shed light on the interest/peculiarity of the transformer architecture.
I would have appreciated if some simple experiments could be done to assess the range of validity of the results and in particular testing the importance of the transformer architecture.

The hypothesis in section 4 to obtain a global convergence result seems not checkable in practice. In my opinion, it is not a global convergence result but rather an "if" theorem. Although I know that this assumption has been put forward in other previous papers, it is still an very demanding hypothesis.

**Questions:**

What are specific features of the transformer architecture that are necessary to make the results valid? In other words, can you extend your results to more general architecture than transformers?

The justification of the hypothesis on the separability property in C5 is difficult to understand. Can you elaborate more on this assumption and try to report precisely on the progress of the literature on this assumption?

I do think it is not fair to claim global convergence for such kind of "if" theorem with a hypothesis that is uncheckable. I suggest the authors to rephrase their contributions.

**Limitations:**

Not applicable.

---

> ### Author Rebuttal · Authors · 2024-08-07
>
> Thank you for your valuable suggestions. In the following, we give point-by-point responses to your questions.
>
> **Q1**: One of the main weaknesses for a submission to NeurIPS is the format. The main paper is devoted to a (well done) explanation of main results, but insufficient explanations of the proof ideas are given in the main paper.
>
> **A1**: Thank you for your comment. Since our work is novel in the context of Transformer models, we believe comprehensive explanations of the main results are necessary to clearly convey the key theoretical messages to readers. Consequently, there is limited space for including the concrete ideas of proofs due to their highly technical nature.
>
> We have made efforts to address the lack of proof sketches. In the appendix, we included proof ideas for the most important results before presenting the rigorous proofs. For example, we discussed the idea of obtaining the adjoint ODE solution of $p_\rho$ in Appendix C.4, and the proof idea of Theorem 3.1 at the beginning of Appendix D. For the proofs of the main results, we also explicitly listed the goal of each proof step to enhance understanding.
>
> We agree that it would be helpful to explain the ideas of proof in the main paper. If this paper is accepted, we will revise the paper structure and also utilize the one additional page for the camera-ready version to add a proof sketch.
>
> **Q2**: The novelty of the results is to take the transformer architecture into account in the mean-field limit. However, the results do not show how this architecture helps in obtaining the results. Can you extend your results to more general architecture than transformers? In addition, some simple experiments could be helpful to assess the range of validity of the results and in particular testing the importance of the transformer architecture.
>
> **A2**: We thank the reviewer for the insightful question. The purpose of our paper is to enable mean-field analysis on Transformers, so that  global convergence guarantees can be established for Transformers. The complexity of the Transformer structure makes achieving such convergence results particularly challenging, and existing mean-field tools cannot be directly applied. Therefore, our paper's contribution lies in overcoming these difficulties.
>
> While our focus is on Transformers, our result can indeed cover (or be easily extended to cover) more general architectures. Importantly, Assumption 4 only requires partial 1-homogeneity. This can potentially enable mean-field analysis of very general architectures and activation functions
>
> Regarding experiments, in practice, we observe that Vision Transformers can fit simple datasets such as MNIST and achieve a near-zero training loss, indicating convergence to a global minimum. We will include experimental results on validating the mean-field limit of transformers in the camera-ready.
>
> **Q3**: Assumptions in Theorem 4.1 are not checkable in practice. It is rather an "if" Theorem. Although these assumptions appear in other previous papers, they are still demanding.
>
> **A3**: We thank the reviewer for the question and the acknowledgement about the common issue in literature. Indeed, many other papers in the mean-field literature make global convergence claims using similar assumptions. Although our paper retains these untestable assumptions to prove global convergence, we emphasize that we have already made significant contributions towards for practical assumptions by:
> 1. Weakening these “if” assumptions compared to [41], as we do not require full homogeneity.
> 2. Weakening Assumptions 1-3 compared to [19, 20, 41], whose assumptions cannot be extended to Transformer structures.
> Therefore, while validity concerns about the “if” Theorem still exist, we believe that weakening these assumptions to a milder version is a significant step towards the ultimate goal of achieving “checkable” assumptions.
>
> **Q4**: Discussions about the hypothesis on the separability property in Appendix C.5 are difficult to understand. Can you elaborate more, and report precisely on the progress of the literature on this assumption?
>
> **A4**: The first assumption in Theorem 4.1 requires that the parameter distribution $\rho^{(\tau)}$ is concentrated in a bounded region across all time. Though this assumption is uncheckable, the introduction of the regularization parameter $\lambda$ penalizes the parameter norms, which may lead to a $\rho^{(\tau)}$ with compact support. Additionally, this assumption is more likely to hold on simpler learning tasks: if the true label generation process is defined by a $\rho^*$ with compact support, then a bounded solution suffices to minimize the loss for such a simple learning task.
>
> The second assumption concerns the separation property. It requires that the support of the convergence point $\rho_\infty$ “separates” a small and a big sphere at any local point $\theta_0 (w_0)$ regarding the 1-homogeneous parameter part, and the support always “spans the full disk” $\mathcal{K}$ used for universal approximation. A mild case that satisfies the “separation part” of this assumption is that the original point $0_{\text{dim}\theta+\text{dim}w}$ is an interior point of the support for some $t^*$. This condition is relatively mild and is generally satisfied, though no paper has rigorously shown it when considering deep networks where $L$ tends to infinity. The challenge arises more in verifying the “spanning full disk” part, i.e., $\text{supp}(\rho_\infty(\cdot,t))$ extends to encompass the entire space $\mathcal{K}$. While no paper can prove this property for the limit $\rho_\infty$, [20] with a similar theoretical setting shows that if the initial parameter distribution $\rho_0(\cdot,t^*)$ spans $\mathcal{K}$, then $\rho^{(\tau)}(\cdot,t^*)$ spans $\mathcal{K}$, i.e., this expansive support property is maintained at any finite time. We will revise Appendix C.5 with clearer statements and more detailed discussions.

---

> > ### Comment · Reviewer_oct4 · 2024-08-13
> >
> > I thank the authors for their answers.
> >
> > About Q2/A2: I do not think that an experiment on the MNIST dataset and just claiming global convergence would be a valuable addition to illustrate the claims in the paper. Studying the size of parameters to obtain this global convergence would be more interesting.
> >
> > In any case, having read the answers to Q3 and Q4, I reckon this work appears as an improvement over the current literature and could motivate some further progress. This motivates me to increase my score.

---

> > > ### Author Response · Authors · 2024-08-14
> > >
> > > Thank you for your suggestions on the experiments and for increasing your score! We will make sure to add experiments to study the size of parameters that can guarantee to global convergence.

---

### Official Review · Reviewer_iBME · 2024-07-29

**Soundness:** 2
**Presentation:** 3
**Contribution:** 3
**Rating:** 5
**Confidence:** 4

**Summary:**

The paper presents a rigorous theoretical analysis of the global convergence properties of gradient flow in training Transformers in the setting of in-context learning. By constructing the mean-field limit, the authors show that as the model's width and depth increase to infinity, the gradient flow converges to a Wasserstein gradient flow. This convergence ensures that the training process reaches a global minimum when the weight decay regularization parameter is sufficiently small. The paper introduces techniques adapted to Transformers, leveraging partial homogeneity and local Lipschitz smoothness to demonstrate the close approximation between discrete and continuous models and to establish the global convergence of gradient flow.

**Strengths:**

Strength:

1. The paper studies the Transformer model in the context of in-context learning. Although the mean-field approach has been applied to infinite width and depth ResNet models before, the Transformer model is currently the most widely used and thus worthy of detailed study. This paper extends the analysis by incorporating two distinct encoders, filling the gap in understanding the Transformer model in infinite limits.

2. The paper introduces new techniques by extending previous approaches used for ResNet models. The authors refine the analysis by assuming only partial homogeneity and local Lipschitz smoothness, which is a key extension. The idea of considering the continuous dynamics as a function of the average behavior of f and h is new and interesting.

3. The final result has a mild requirement for the dependence of depth and width, where one would only need depth $L = \Theta(\epsilon^{-1})$ and the number of Transformer block $M = \Theta(\epsilon^{-2} \log (\epsilon^{-1}))$ to achieve a loss as small as $\epsilon$.  (However, the reviewer has some questions about the result, so this strength can still be questionable.)

4. The paper is well-organized and clearly written. The authors effectively communicate the ideas, contributions, approaches, and limitations of their work, making it accessible and comprehensible to the reader.

**Weaknesses:**

## Potential fallacy in the claim of Corollary 4.1:

1. The reviewer is convinced by most results presented in the paper, but has concerns about the claim in Corollary 4.1. The main issue arises from the fact that the authors treat many objects as constants, even though they clearly depend on some key quantities. In short, the reviewer thinks that $C_1$ in Theorem 4.1 depends significantly on $\epsilon$, which makes the claim in Corollary 4.1 questionable.
The concern stems from the observation that $C$ in Theorem 3.1 may depend exponentially on $\tau$. Following the proof of Theorem 4.1, to achieve a loss as small as $\epsilon$, one would need $Q(\rho^{(\tau_0)}) \le \epsilon$. This introduces a non-trivial dependence of $\tau_0$ on $\epsilon$. Given that $C$ may depend exponentially on $\tau_0$, $C_1$ in Theorem 4.1 would thus depend exponentially on $\tau_0$, leading to a complex dependence on $\epsilon$. This dependence cannot be ignored, and treating $C_1$ as a simple constant to achieve Corollary 4.1 appears incorrect. (The reviewer believes this is how Corollary 4.1 is currently derived.)
Based on the above, the reviewer thinks that
    1. The current approach in proving Corollary 4.1 is flawed.
    2. A naive application of Theorem 4.1 might only lead to asymptotic bounds for $L$ and $M$, instead of the linear and quadratic terms presented nicely in the current version of Corollary 4.1.

## Limitations of settings and results

Though the reviewer agrees that studying and achieving a good understanding of the case where both width and depth go to infinity is valuable, there are some limitations in the results presented in the paper.

1. The main result, Theorem 4.1, depends on Assumptions 1-4. Although the reviewer considers Assumptions 1-4 to be reasonable and likely to hold under certain settings, the authors do not make it clear when these assumptions would hold. Consequently, the setting of the Transformer model might differ from classical Transformer models (e.g., it might only hold under the reparameterized case (2.3) instead of keeping all value, key, query, and output matrices). The reviewer will address this point more in the Questions section.

2. The main result, Theorem 4.1, relies heavily on a few additional assumptions that are not justified in the paper:
    1. the Wasserstein gradient flow weakly converges to some distribution $\rho_{\infty}$
    2. the uniform boundedness of $\rho^{(\tau)}$ for large time $\tau$
    3. the separation property for $\alpha_1$ with the support expansion of $\alpha_2$ to $K$.

The reviewer acknowledges that some of these assumptions are also adopted in other mean-field approaches and appreciates the authors' efforts to justify them intuitively in Section C.5. However, they are not examined carefully and may harm the validity of the statement. For example, for the second assumption, the reviewer agrees that a large regularization $\lambda$ can implicitly bound the norm of the solution. However, this might require a large $\lambda$, which could make the results less meaningful. Notice that the current bound shown in Proposition 3.2 implies that $R_{\tau}$ may depend exponentially on $\tau$, thus it is not clear that a small $\lambda$ can result in a bounded solution. And based on Theorem 4.1, a small $\lambda$ would be required to achieve a small loss.

3. The result does not provide a non-asymptotic bound for the required training time. In other words, there is no guarantee that $\hat{Q}(\Theta^{(\tau)})$ will be small for any specific $\tau$. This is because there is no control over the convergence rate of $\rho^{(\tau)}$ to $\rho^{\infty}$, which would require a stronger assumption beyond the current weak convergence assumption of $\rho^{(\tau)}$.

4. The loss and training dynamics are only considered for the population risk under $\mu$. There is no consideration of the effect of an empirical dataset. This is a clear limitation, but the reviewer can accept this as an initial step toward studying the Transformer model.

## Minor:

1.  Some parts of the statements are not well organized. For example, Assumption 2 is an assumption about norm bounds of function $f$ and its gradient. The line "define ReLU'(x) = ..." has no direct connection with the assumption. The reviewer thinks that the authors are trying to argue that by using ReLU'(x) as the gradient of the ReLU function, $f$ defined in (2.3) would satisfy Assumption 2. If this is the case, the reviewer suggests the authors separate the statement into 1) what the assumption of $f$ is. and 2) when the assumption would hold. Another minor point is that the Fréchet derivative still acts differently from the gradient, so some careful treatment might be needed if the authors claim that the result in Theorem 4.1 applies to Transformers with ReLU activation.

Typos:
1. Line 111, it should be "h(Z, w) = W_2 \sigma_M(W_1 Z)" instead of $W_1 H$.
2. Line 156, in Assumption 2, $\Psi_j$ in the description should be $\Phi_P$.
3. Line 86: "~~the~~ our deep Transformer model"
4. Line 901, For the line that bounds $R(\Theta)$, there is no $C_{\lambda}$ in the first line, and the transition from the first line to the second line is currently shown as $1/2 \epsilon + ... \le 1/4 \epsilon + ...$

**Questions:**

1. Please address the above concern for Corollary 4.1 carefully. The answer may greatly affect the final rating from the reviewer.

2. Questions related to assumption 1-4:
    1. For the purpose of confirmation, have you shown in which cases Assumption 1-4 are met for the Transformer model? My current understanding is that they would be met when you reparameterize $f$ as in (2.3) but not when you keep the W_V, W_K, W_Q and W_O matrices (at least Assumption 2 won't be satisfied in this case). Is that correct? The review is not against the reparameterization and considers this is still a Transformer model, but want to understand the difference and the claim of the paper more carefully.
    2. As the focus of the paper is on the Transformer model, the main results are based on the structure defined in (2.1) (2.2), (2.4) and Assumptions 1-4 over f and h. The review wonders if it is possible to provide a separate corollary for a commonly used form that satisfy Assumption 1-4, such as (2.3), making it clearer on an example where minimum assumptions are used. Or in other words, are there technical difficulties in showing Assumption 1-4 for (2.3)? Is the universal approximation capabilities the key obstacle?

**Limitations:**

The authors have adequately addressed the limitation of the paper.

---

> ### Author Rebuttal · Authors · 2024-08-07
>
> We are grateful for your constructive and helpful comments and suggestions. We address your concerns as follows.
>
> **Q1**: Corollary 4.1 only implies asymptotic bounds for $L$ and $M$, instead of the linear and quadratic terms in the current version of Corollary 4.1.
>
> **A1**: Thank you for the crucial clarification on this point. You are correct, and we will revise Corollary 4.1 and remove the comment on how $L_0$ and $K_0$ depend on $\epsilon$. Our result only ensures that the risk can be bounded when ​​$L$ and $M$ are sufficiently large.
>
> Our work does not aim to provide a tight bound on $L$ and $M$ with respect to $\tau_0$ or measure the exact reduction in risk. Instead, our goal is to take the first theoretical step in demonstrating that large-scale Transformers can achieve global convergence via gradient descent-based optimization. Even though our result is asymptotic and does not involve an explicit rate, it is the first of its kind and lays the groundwork for future theoretical optimization guarantees for Transformers.
>
> **Q2**: Assumptions 1-4 may rely heavily on the reparameterized form of transformer as in (2.3). The reviewer is not against the reparameterization and considers this is still a Transformer model, but wants to understand the difference and the claim of the paper more carefully. Additionally, is it possible to provide a separate corollary for a commonly used form that satisfies Assumption 1-4, such as (2.3)?
>
> **A2**: Thank you for your suggestion. We indeed mainly focus on the reparameterized form of transformer in (2.3). However, given the theoretical nature of our work, we believe that the form of transformer in (2.3) covered in our paper is already pretty close to the practice, compared with most existing theoretical analyses of transformers.
>
> We believe that Assumptions 1-4 are satisfied when we consider:
> - The form of (2.3) for the attention layer, with $\sigma_A$ being the column-wise softmax.
> - $W_2\sigma_M(W_1H)$ for the MLP layer with the ReLU activation or smooth activations with Lipschitz derivatives.
>
> The first two assumptions are easier to verify by definition. Assumption 3 is straightforward to verify when all activation functions are smooth. For ReLU activation, Assumption 3 can still hold when the data are generated from a smooth distribution – all the properties in Assumptions 3 are in the form of expectations over the data distribution, and such an expectation can have a smoothing effect.
>
> For Assumption 4, the partial 1-homogeneity is clearly met in our paper given the partially linear structure of Transformers. Furthermore, since the universal kernel property applies to either the attention or the MLP layer, we can choose the MLP layer with the ReLU activation, whose universal approximation ability has been discussed in [12].
>
> We will add more discussions in the revision.
>
> **Q3**: Theorem 4.1 relies heavily on a few additional assumptions that are not examined carefully and may harm the validity of the statement. It is not clear that a small $\lambda$ can result in a bounded solution. And based on Theorem 4.1, a small $\lambda$ would be required to achieve a small loss.
>
> **A3**: Thank you for acknowledging the common adoption of these assumptions. We agree that a small $\lambda$ may not always result in a bounded solution. We treat the assumption of “the uniform boundedness of $\rho^{(\tau)}$ for a long time $\tau$” as a data-related assumption. Our heuristic thinking is that if the true data distribution $\mu$ is simple, and the true label generation process $\rho^*$ is bounded, then a bounded solution suffices to minimize the loss for such a simple learning task.
> Likewise, the first and third assumptions are also assumptions on the learning task regarding $\mu$ and $\rho^*$. We believe they should be satisfied when the task is sufficiently simple. However, like most other papers considering mean-field analysis across multiple network layers, we acknowledge that it is challenging to construct a general class that could be verified to meet these assumptions.
>
> **Q4**: The result does not provide a non-asymptotic bound for the required training time.
>
> **A4**: Thank you for your comment. Similar to our response in Q1, we believe that even establishing an asymptotic convergence bound is a novel and significant finding given the technical complexities of the Transformer models. Establishing more concrete convergence rates is an important future work direction.
>
> **Q5**: The loss and training dynamics are only considered for the population risk under $\mu$, but the reviewer can accept this as an initial step toward studying the Transformer model.
>
> **A5**: We thank the reviewer for the careful reading and understanding. We consider the extension to finite sample results with explicit generalization error as the next step of our research.
>
> **Q6**: In Assumption 2, the line "define ReLU'(x) = ..." has no direct connection with the assumption. The authors should separate the statement into
> 1) what the assumption of $f$ is.
> 2) when the assumption would hold. The Fréchet derivative still acts differently from the gradient, so some careful treatment might be needed.
>
> **A6**: We thank the reviewer for the careful reading and the excellent suggestions. We indeed aimed to claim that by using $\text{ReLU}'(x)$ as the gradient of the ReLU function, $f$ defined in (2.3) would satisfy Assumption 2. We will revise the statement following your suggestions.
>
> We believe the formula of the Fréchet derivative holds under ReLU activation (Equation (3.4) and Proposition 3.1), as the Fréchet derivative is more akin to the weak derivative and milder than the standard definition of the derivative. We will add a detailed discussion in the revision.
>
> **Q7**: Some typos.
>
> **A7**: We thank the reviewer again for pointing out the typos, especially with the careful reading of the proof. Line 901 indeed contains an error in the constant term. We will fix them all in the revision.

---

> > ### Comment · Reviewer_iBME · 2024-08-09
> >
> > The reviewer first thanks the authors for their detailed and thoughtful response. The reviewer also appreciates the authors' honesty and transparency in the rebuttal. Unfortunately, the reviewer does not feel that the concerns have been adequately addressed. As a result, the reviewer has decided to change the rating. The reviewer understands how frustrating this can be, especially given the thorough rebuttal provided by the authors in an effort to address all the questions raised. The reviewer struggled with the rating during the initial review phase and provided the rating with the hope that some of these concerns would be resolved. However, after careful and extensive consideration, the reviewer believes that the paper remains incomplete in its current form.
> >
> > Before delving into specific concerns, the reviewer would like to acknowledge the strengths of the paper, as noted in the initial review. The paper is well-organized and clearly written, and it tackles a challenging and important problem: the theoretical understanding of training Transformer models. The reviewer recognizes that theoretical work, particularly in complex areas like Transformer networks, is inherently difficult. The reviewer is open to compromises, such as accepting reasonable assumptions that may be difficult to verify or acknowledging results with suboptimal dependence on problem parameters, including asymptotic results.
> >
> > However, based on the authors' response, the reviewer believes the following deficiencies are significant:
> > 1. Corollary 4.1 is incorrect and requires major revision.
> > 2. The dependence on training time $\tau$, the depth $L$ and the width $M$ is purely asymptotic.
> > 3. The paper lacks clear results for the Transformer network (e.g., in the form of (2.3)) or for functions with ReLU activation.
> >
> > First of all, having a misleading or incorrect conclusion for one of the main claims of the paper is a serious issue, as it undermines the credibility of the rest of the work. This problem also leads to the asymptotic nature of the results, which significantly diminishes the importance of the paper’s contributions from the reviewer’s perspective. To clarify for those who may be interested, the reviewer believes that the logic behind Theorem 4.1 and Corollary 4.1 is as follows: A specific training time is required to ensure that the limiting dynamics reach a small error $Q(\rho^{(\tau_0)})$, and $L$ and $M$ must be chosen large enough so that the discrete training dynamics remain close to the limiting dynamics within the time $\tau_0$. The lack of control over $\tau_0$ to achieve a small loss leads to the asymptotic nature of the time (with additional weak convergence assumption), depth, and width.
> >
> > These first two deficiencies alone prevent the reviewer from giving the paper a high score. However, what is most concerning upon reevaluating the paper is the third point. The paper claims to provide convergence results for large-scale Transformers, yet no specific Transformer model is present in any of the main theorems. A Transformer should be defined in the form of (2.4) or at least in the form of (2.3), rather than as a concatenation of general functions $f$ and $h$. In the rebuttal, the authors used many "believe"s including the "belief" that (2.3) would satisfy Assumptions 1-4 and the "belief" that the results apply to the ReLU function with its Fréchet derivative. While the reviewer agrees with these beliefs on an intuitive level, the absence of a theorem that includes at least the form in (2.3) or the ReLU function makes these "beliefs" shaky and the work incomplete. It is somewhat surprising that the authors claim to achieve results for Transformers, yet provide no specific results related to the attention mechanism (2.4) (or its reparameterized form (2.3)) and the ReLU function. This presentation is therefore also misleading, making it difficult to connect the main results in the paper with a Transformer network.  Given the rebuttal, it seems that Assumptions 1-4 for (2.3) and ReLU activation for Proposition 3.1 are checkable, so including these components would form a more complete story for the convergence of Transformers in the asymptotic regime.

---

> ### Comment · Reviewer_iBME · 2024-08-09
>
> Given the above points, the reviewer considers the paper to be an interesting attempt at tackling important questions. The paper introduces valuable techniques for addressing problems related to Transformer networks and has the potential to become a strong contribution. However, in its current form, the paper is incomplete and would benefit greatly from revision and resubmission to a future venue. The reviewer suggests the following improvements:
> 1. Please rewrite Corollary 4.1 and other theorems, ensuring that the dependencies for all constants are clearly specified.
> 2. Prove that assumptions 1-4 are satisfied for Transformer networks in the form of (2.3).  It would be helpful if some parameters in the assumptions, including $K, K_T, K_P, \Phi_T, \Phi_P, \Phi_{PP}, \Phi_{TP}, \Phi_{TT}$ could be represented in more explicit forms with clear bounds.
> 3. If the authors intend to include ReLU activation beyond smooth activations, they should demonstrate that ReLU activation also satisfies Assumptions 1-4 and Proposition 3.1.
> 4. For the assumptions used in Theorem 4.1, even though it may be difficult to prove when these assumptions hold, it would be beneficial to provide heuristic examples to help readers understand when and why the assumptions are valid. For instance, the example provided by Reviewer accV in question 1 (Q3 in the rebuttal) offers a good illustration that this bounded assumption may not generally hold. Therefore, the authors need to make a stronger effort to justify the assumptions. In the current response to that question, phrases like "remains possible" are not convincing enough.
>
> Again, the reviewer understands that theoretical work in this area is challenging and that compromises are often necessary. For example, the weak convergence assumption is difficult to avoid in the mean-field regime, and previous work on ResNet models [Ding et al. 2021] has also shared the same asymptotic nature as in this paper. This is why the reviewer initially leaned toward accepting the paper, hoping that some of these concerns could be addressed. However, as outlined above, the reviewer does consider the work to be incomplete and believes that the paper does not meet the standard for a conference paper due to its lack of direct application to Transformer models, its weak (asymptotic) results, and its reliance on assumptions that are difficult to verify. The reviewer apologizes for the change in rating, noting that this adjustment reflects a reassessment of the initial rating rather than a direct response to the rebuttal.
>
> [Ding, Zhiyan, et al. "Overparameterization of deep ResNet: zero loss and mean-field analysis." Journal of Machine Learning Research 23.48 (2022): 1-65.]

---

> > ### Author Response · Authors · 2024-08-10
> >
> > We appreciate your careful feedback. We understand the concerns you have raised regarding Corollary 4.1, the asymptotic nature of our results, and the applicability to Transformer models. However, we would like to argue that these weaknesses you have mentioned are essentially the weaknesses of almost all existing mean-field studies of deep neural networks. In fact, our paper contributes to pushing the common assumptions in the mean-field literature towards more practical scenarios. Therefore, we respectfully disagree with your comment that our work is incomplete.
> >
> > Corollary 4.1 should not need major revision, as its mathematical formulation and technical proof are accurate. To ensure rigor, it suffices to delete the sentence in line 302 "Here, $L_0$ scales as $\Omega(\epsilon^{-1})$, and $K_0$ as $\Omega((1+\delta)\epsilon^{-2})$", and remove the paragraph around lines 304-306.
> >
> > Regarding your concern that the dependence on training time $\tau$, the depth $L$ and the width $M$ is asymptotic, we would like to emphasize again that our result remains, to our best knowledge, the state-of-the-art global convergence guarantees for large-scale Transformers. Moreover, we would also like to point out that our work has two main results: (i) connections between the discrete Transformers and their continuous limit, and (ii) global convergence guarantees of Transformers. We feel that it is unfair to deny all of our contributions based solely on that our global convergence guarantees for discrete Transformers require asymptotically large $\tau$, $L$ and $M$.
> >
> > Regarding your concern about the applicability to Transformers, we honestly believe that the Transformer architecture we consider is already among the ones considered in theoretical works that are closest to practical Transformer architectures.
> >
> > We will add concrete examples of Transformer architectures and rigorously prove that Assumptions 1-4 hold for these architectures in the camera-ready version. As clarified, these assumptions easily hold for activation functions that are sufficiently smooth. We will also clarify the case of the ReLU activation function, noting that it may not generally satisfy these assumptions, but it is possible for Assumptions 1-4 to hold when the data follow certain favorable distributions.
> >
> > Once again, we appreciate your detailed and comprehensive feedback. We sincerely hope that you can consider re-evaluating our contributions. Thank you for your time and attention.
> >
> > Best regards,
> >
> > Authors

---

> > > ### Comment · Reviewer_iBME · 2024-08-13
> > >
> > > The reviewer appreciates the authors' further response and acknowledges the effort made in addressing the concerns raised. Here are the reviewer's further comments:
> > >
> > > 1. The reviewer agrees that the term "major revision" used in the previous feedback regarding Corollary 4.1 was not accurate. For the sake of correctness, removing the lines related to the dependencies of $L$ and $M$ should be sufficient. When the reviewer suggested a revision, it was more about improving the clarity of the statement. For instance, as also noted by reviewer mtEE, the authors have treated many quantities as constants even though they still have dependencies on the problem parameters. The reviewer believes that this may have contributed to the inability to identify the incorrect conclusion in the initial paper. Another minor revision could involve removing the $C_{\lambda}$ constraint and instead simply choosing $\lambda \le (2C_2)^{-1}\epsilon$, and then adjusting the proofs on line 901 with the correct constants.
> > >
> > > 2. The reviewer also acknowledges that many current works in the mean-field regime exhibit asymptotic behavior. The reviewer recognizes that this work represents state-of-the-art results in this challenging area. However, the asymptotic nature of the results, particularly regarding the width and depth, still limits their practical applicability.
> > >
> > > 3. As noted in the previous response, the concern regarding the concrete application to the Transformer architecture remains the key issue that leads to a reject score. The reviewer appreciates the authors' willingness to include concrete examples of Transformer architectures and to address the case of ReLU activation. The reviewer agrees that these additions seem feasible, and if incorporated, the reviewer would be willing to raise the score and accept the paper. However, while these additions make sense intuitively and the authors seem confident in their ability to implement them, the reviewer remains skeptical that these theorems can be adequately developed and integrated into the camera-ready version, given the current form of the paper and the volume of work required. Nonetheless, including these components would make the work more convincing regarding the convergence story for Transformer networks.

---

> ### Author Response · Authors · 2024-08-14
>
> Thank you very much for acknowledging that the asymptotic assumptions on $L$ and $M$ are common in the mean-field analysis literature. We also appreciate your suggestion on removing $C\_{\lambda}.$ We will follow your suggestion, choose $\lambda \le (2C\_2)^{-1}\epsilon,$ and adjust the proofs accordingly.
>
> Regarding your concern about the concrete application to the Transformer architecture, we would like to clarify that verifying Assumptions 2-4 (please note that Assumption 1 is irrelevant to the Transformer model, and is only a fairly mild assumption on the data) for concrete examples of Transformer architectures with smooth activations is fairly intuitive and the proof is mainly based on a series of tedious calculations. Below, we give a concrete proposition and its brief proof as follows.
>
>
> ---
>
> Consider
>
> $f(Z,\theta)=VZ \mathrm{softmax} (Z^T W Z ) $ $\qquad $ (Eq 1)
>
> with the collection of parameters $\theta = \mathrm{vec}[V,W],$ where $\mathrm{softmax}$ denotes the column-wise softmax function. Moreover, consider
>
> $h(Z,w)=W\_2\mathrm{HuberizedReLU}(W\_1H)$  $\qquad $ (Eq 2)
>
> with the collection of parameters $w= \mathrm{vec}[W\_1,W\_2].$ Here, $\mathrm{HuberizedReLU}$ denotes the entry-wise HuberizedReLU activation function defined as
>
> $\mathrm{HuberizedReLU}(x) = \left\\{
> \begin{aligned}
> &0, &&\mathrm{if} \ z\leq 0;\\\\
> &z^2/2, &&\mathrm{if} \ z\in [0,1];\\\\
> &z - 1 / 2, &&\mathrm{if} \ z\geq 1.
> \end{aligned}
> \right. $
>
> Then, we can consider a Transformer model defined by equations (2.1), (2.2), and (2.4) in the paper, where the functions $f$ and $h$ are specified above. We suppose that this Transformer model is applied to a learning task with data that satisfies Assumption 1.
> We have the following proposition.
>
>
> **Proposition.** Consider the Transformer model defined by equations (2.1), (2.2) and (2.4), with $f(Z,\theta)$ and $h(Z,w)$ defined in (Eq 1) and (Eq 2) respectively. Then Assumptions 2-4 all hold.
>
> In our following comments, we will present a proof sketch of the proposition above. Due to the large character count of equations, we split the proof into several separate replies. We apologize for the long comments. To simplify our response, we omit the detailed derivations for the function $h(Z,w)$, which corresponds to the MLP part, in our verification of Assumptions 2 and 3. We feel that the fact that $h(Z,w)$ satisfies Assumptions 2 and 3 is relatively more intuitive, especially given the proofs for $f(Z,\theta)$. We will also omit some calculation details to keep our response reasonably simple. We will make sure that all the omitted details will be added to the revised version of our paper.

---

> ### Author Response · Authors · 2024-08-14
>
> Proof. We first introduce some notations. We remind the readers that for a matrix $\mathbf{A}$, we denote by $\\| \mathbf{A} \\|\_2$ the spectral norm of $\mathbf{A}.$ We also denote by $\\| \cdot \\|\_{1-\mathrm{col}}$ the maximum $\ell\_1$-norm across all columns of a matrix.
>
> Denote $Z=(z\_1,\dots,z\_{N+1})\in\mathbb{R}^{D\times N+1}.$ Then the function $f$ can be rewritten as
>
> $f(Z,\theta)=VZ\mathrm{softmax}(Z^TWZ)=(f(Z,\theta)\_{:,i})\_{1\leq i\leq N+1}$,
>
> where
>
> $f(Z,\theta)\_{:,i}=\sum\_{j=1}^{N+1}P\_{ij}z\_j$ and $P\_{i,:}=\mathrm{softmax}(Z^TWz\_i).$ Next, we calculate the derivatives of $f(Z,\theta)\_{:,i}$ with respect to $Z$ and $\theta$ as follows:
>
>
> For $Z$: the Jacobian $J\in\mathbb{R}^{(N+1)D\times(N+1)D}$ is $J=(J\_{ij})\_{1\leq i,j\leq N + 1},$ where $J\_{ij}=\frac{\partial f\_{:,i}}{x\_j}\in\mathbb{R}^{D\times D}.$ After calculation, we obtain
>
> $J\_{ij}=ZQ\_i\big[E\_{ji}Z^TW+Z^TW^T\delta\_{ij}\big]+P\_{ij}I,$
>
> where $Q\_i:=\mathrm{diag}(P\_{i:})-P^T\_{i:}P\_{i:},$ $E\_{ij}$ is the matrix with zeros everywhere except one the $(i,j)$-th entry, and $\delta\_{ij}$ is the Kronecker delta.
>
>
> For $\theta$: Define $A\_i=Z^TWz\_i.$ After calculation, we have
>
> $\nabla\_{\mathrm{vec}[V]} f(Z,\theta)\_{:,i}=\sum\_{j=1}^{N+1}P\_{ij}\mathrm{diag}\Big([B\_{kj}(z\_j)]\_{1\leq k\leq D}\Big),$
>
> $\nabla\_{\mathrm{vec}[W]} f(Z,\theta)\_{:,i}=\sum\_{j=1}^{N+1}P\_{ij} \Big(\mathrm{diag}\Big([z_l^T  B\_{kj}( z\_j)]\_{1\leq k\leq D}\Big)[\frac{\partial\mathrm{softmax}(A\_i)}{\partial A\_i}]\_l\Big)_{1\leq l \leq N+1} Vz\_j,$
>
> where $B\_{kj}(z)$ is the $D\times D$ matrix with zeros everywhere except $z\_j$ for the $k$-th row.
>
> ---
>
> We then verify the assumptions one by one. For Assumption 2 (i), we have
>
>
> $ \\| f(T,\theta) \\|\_{2-\mathrm{col}}  = \\| VT \mathrm{softmax} (T^T W T ) \\|\_{2-\mathrm{col}} \leq \\| V\\|\_2 \cdot \\| T \\|\_2 \cdot \\| \mathrm{softmax} (T^T W T ) \\|\_{2-\mathrm{col}} \leq \\| \theta \\|\_2 \cdot \\| T \\|\_{2-\mathrm{col}}  \cdot \\| \mathrm{softmax} (T^T W T ) \\|\_{1-\mathrm{col}} \leq \\| \theta \\|\_2 \cdot \\| T \\|\_{2-\mathrm{col}},$
>
> where the second-to-the-last inequality follows by the fact that $\ell\_2$-norm can be upper bounded by the $\ell\_1$-norm, and the last inequality follows by the fact that each column of the softmax output has an $\ell\_1$-norm equaling one. Therefore, the first condition in Assumption 2 with $K = 1$ is verified for the function $f$ in (Eq 1).
>
> For $h$ in (Eq 2), we have
>
> $ \\| h(T,w) \\|\_{2-\mathrm{col}} = W\_2\mathrm{HuberizedReLU}(W\_1T) \\|\_{2-\mathrm{col}} \leq \\| W\_2 \\|\_2 \cdot \\|\mathrm{HuberizedReLU}(W\_1T) \\|\_{2-\mathrm{col}}  \leq  \\| W\_2 \\|\_2 \cdot \\|W\_1T \\|\_{2-\mathrm{col}} \leq 2 \cdot \\| w \\|\_2^2 \cdot \\|T \\|\_{2-\mathrm{col}}$,
>
> where the second inequality follows by the property of HuberizedReLU that $|\mathrm{HuberizedReLU}(x)| \leq |x|.$ This demonstrates that Assumption 2 (i) with $K=1$ holds for $h$ in (Eq 2) as well.
>
>
>
> For Assumption2 (ii), we have
>
> $\\| \nabla\_{\mathrm{vec}[V]} f(T,\theta)\_{:,i} \\|\_{2} \leq \sum\_{j=1}\^{N+1} P\_{ij} \\| \mathrm{diag}\Big([B\_{kj}(T\_j)]\_{1\leq k\leq D}\Big) \\|\_{2} \leq \sum\_{j=1}^{N+1} P\_{ij} \\| T\_j \\|\_{2}
> \leq \\|T\\|\_{2-\mathrm{col}}.
> $
>
> Similarly, we have
>
> $\\| \nabla\_{\mathrm{vec}[W]} f(T,\theta)\_{:,i} \\|\_{2} \leq \sum\_{j=1}^{N+1} P\_{ij} \Big\\| \Big(\mathrm{diag}\Big([z_l^T  B\_{kj}( z\_j)]\_{1\leq k\leq D}\Big)[\frac{\partial\mathrm{softmax}(A\_i)}{\partial A\_i}]\_l\Big)_{1\leq l \leq N+1} VT\_j \Big\\|\_{2}
> \leq \sum\_{j=1}^{N+1} P\_{ij} \sum\_{l=1}^{N+1}\\|T\_l^T T\_j \\|\_{2}\cdot  (\frac{\partial\mathrm{softmax}(A\_i)}{\partial A\_i})\_l \cdot \\|V\\|\_{2} \cdot \\| T\_j \\|\_{2}
> \leq \\|T\\|\_{2-\mathrm{col}}^3 \\|\theta\\|\_{2}.$
>
> Combining the two equations above gives Assumption2 (ii) with $\phi\_P( \\|T\\|\_{2-\mathrm{col}} ) = \\|T\\|\_{2-\mathrm{col}} + \\|T\\|\_{2-\mathrm{col}}^3$.
>
> For Assumption2 (iii), we have
>
> $\\| J \\|\_{2}\leq \sqrt{\sum\_{1\leq i,j\leq N+1}\\|J_{ij}\\|^2\_2}\leq (N +1) \max\_{1\leq i,j\leq N+1} \\|J_{ij}\\|\_2.$
>
> For any $1\leq i,j\leq N + 1,$ we have
>
> $\\|J_{ij}\\|\_2\leq P_{ij} + \\|T\\|\_2 \\|Q\_i\\|\_2 \\|E\_{ji}T^TW+T^TW^T\delta\_{ij}\\|\_2
> \leq 1 + 2 \\|T\\|\_2^2 \\|W\\|\_2
> \leq 1 + 2 \\|T\\|\_F^2 \\|\theta\\|\_2.
> $
>
> Hence, we have
>
> $\\| J \\|\_{2}\leq 2N \\|T\\|\_F^2\cdot (1 + \\|\theta\\|\_2).$
>
> The above equation demonstrates that for $f$, Assumption2 (iii) holds with $\phi_T ( N, D, \\| T \\|\_F ) = 2N \\|T\\|\_F^2$.
>
> We have verified Assumption 2 for the attention layer encoder $f$. The verification for $h$ is similar and easier. We omit the derivation for $h$ here to shorten our response, but we will make sure to include the full verification in the revised version of the paper.

---

> ### Author Response · Authors · 2024-08-14
>
> Next, we verify Assumption 3. Given that we are currently considering the example where the encoder employs a smooth univariate activation function, we can prove stronger results by removing the expectation $\mathbb{E}_{\mu}$.
>
> (i) and (iii): Given the calculation of derivatives we have presented above, it suffices to show that $\mathrm{diag}\Big([B\_{kj}(z\_j)]\_{1\leq k\leq D}\Big)$ and $\mathrm{diag}\Big([Z^T  B\_{kj}( z\_j)]\_{1\leq k\leq D}\Big)\frac{\partial\mathrm{softmax}(A\_i)}{\partial A\_i}Vz\_j$ are both locally Lipschitz continuous with respect to $Z$ and $\theta.$
>
> Since each of $\mathrm{diag}\Big([B\_{kj}(z\_j)]\_{1\leq k\leq D}\Big),$ $\mathrm{diag}\Big([Z^T  B\_{kj}( z\_j)]\_{1\leq k\leq D}\Big),$ $\frac{\partial\mathrm{softmax}(A\_i)}{\partial A\_i}$ and $Vz\_j$ is obviously bounded by an increasing function of $N,D,\\|\theta\\|, K_T,L_T,$ it suffices to show that each of them is locally Lipschitz continuity with respect to both $Z$ and $\theta.$ This is straightforward as they are all sufficiently smooth.
>
> (ii) and (iv): Because the norm of the difference of two Jacobian matrices $\\|J^1-J^2\\|\_2$ is bounded by $\sqrt{\sum\_{1\leq i,j\leq N+1} \\|J^1\_{ij}-J^2\_{ij}\\|\_2^2},$ it suffices to show that $J\_{ij}$ is locally Lipschitz continuous with respect to both $\theta$ and $Z.$ Again each component of $J\_{ij}$ that depends on $Z$ or $\theta,$ i.e.
> $Z,Q_i,W,P\_{ij},$ is bounded by an increasing function of  $N,D, K_P,L_T,K_T,\\|\theta\\|,$ and is locally Lipschitz continuous given sufficient smoothness. Hence, (ii) and (iv) also hold.
>
> The proof for the HuberizedReLU MLP encoder is similar to the above and is not conceptually complicated, if not easier. We omit the details here, and save the space for a more detailed discussion about Assumption 4, as we expect that you may be more skeptical about the proof regarding the “universal kernel” assumption in Assumption 4.
>
> ---
>
> For Assumption 4, we consider the pair $(g,\alpha) = (h,w),$ and the partition $\alpha = (\alpha\_1,\alpha\_2)$ with $\alpha\_1 = W\_2,$ $\alpha\_2 = W\_1$. We also let a compact set $\mathcal{K} = \\{ W_1: \\| W_1 \\| \leq 1 \\}$. Then Assumption 4(i) on the partial $1$-homogeneity property straightforwardly holds:
>
> $h(T,W\_1,c\cdot W\_2) = c\cdot W\_2\mathrm{HuberizedReLU}(W\_1H)= c\cdot h(T,W\_1,W\_2).$
>
> Regarding Assumption 4(ii) on the universal kernel property, we first note that according to the choice $(g,\alpha) = (h,w)$, this assumption is purely an assumption on the MLP part of the Transformer. Here we give the detailed proof as follows.
>
> First of all, according to the classic universal approximation theory (see the wiki page of “universal approximation theorem” and [1,2,3] for more details), we know that two-layer fully-connected networks with non-polynomial activation functions and without any constraints on its parameters are universal approximators.
>
> Therefore, we know that the function class
>
> $ \mathrm{span} \\{  W\_2\mathrm{ReLU}^2(W\_1T):  W_1 \in \mathbb{R}^{\mathrm{dim}(W_1)}, W_2 \in \mathbb{R}^{\mathrm{dim}(W_2)}  \\} $
>
> is dense in $\mathcal{C}(\\|T\\|\_{2-\mathrm{col}}\leq B,\mathbb{R}^{D\times(N+1)})$. Moreover, by the definition of HyberizedReLU, for any $B>0$ and any $\hat{W}_1$, $\hat{W}_2$, there exist small constant $c$ such that $c\cdot \hat{W}_1 \in \mathcal{K}$, $c\cdot \\|\hat{W}_1\\| \leq B^{-1}$, and
>
> $ c^{-2} \cdot \hat{W}\_2\mathrm{HyberizedReLU}(c\cdot \hat{W}\_1T) =  c^{-2} \cdot \hat{W}\_2\mathrm{ReLU}^2(c\cdot \hat{W}\_1T) = c^2\cdot c^{-2} \cdot \hat{W}\_2\mathrm{ReLU}^2(\hat{W}\_1T) = \hat{W}\_2\mathrm{ReLU}^2(\hat{W}\_1T)$,
>
> where the second equation follows by the positive $2$-homogeneity of $\mathrm{ReLU}^2$ activation. This implies that
>
> $  \\{  W\_2\mathrm{ReLU}^2(W\_1T):  W_1 \in \mathbb{R}^{\mathrm{dim}(W_1)}, W_2 \in \mathbb{R}^{\mathrm{dim}(W_2)}  \\}  \subseteq \\{W\_2\mathrm{HyberizedReLU}(W\_1T): W_2 \in \mathbb{R}^{\mathrm{dim}(W_2)}\times \mathcal{K}\\} $.
>
> Therefore, we conclude that $ \mathrm{span} \\{W\_2\mathrm{HyberizedReLU}(W\_1T): W_2 \in \mathbb{R}^{\mathrm{dim}(W_2)}\times \mathcal{K}\\}$ is dense in $\mathcal{C}(\\|T\\|\_{2-\mathrm{col}}\leq B,\mathbb{R}^{D\times(N+1)})$. This finishes the validation of Assumption 4.
>
> ---
>
> [1] Funahashi, Ken-Ichi (January 1989). "On the approximate realization of continuous mappings by neural networks". Neural Networks.
>
> [2] Cybenko, G. (1989). "Approximation by superpositions of a sigmoidal function". Mathematics of Control, Signals, and Systems.
>
> [3] Pinkus, Allan (January 1999). "Approximation theory of the MLP model in neural networks". Acta Numerica.

---

> > ### Author Response · Authors · 2024-08-14
> >
> > We hope that by presenting the details that you have requested, we can convince you that our theory can be applied to fairly practical Transformer models, and that the volume of work required should not be an issue. We are dedicated to add additional details on the explanations and verifications in our revised paper. We sincerely hope that you could take our discussion above into consideration and reevaluate our result. We appreciate your time and effort in reviewing our paper.

---

> > > ### Comment · Reviewer_iBME · 2024-08-14
> > >
> > > The reviewer thanks the authors for their further efforts. While the reviewer is not able to examine the full proof within the rebuttal period, the reviewer will take a closer look into it in the coming days. Based on the provided proof, the reviewer believes that it can be incorporated into the revised paper and is therefore willing to increase the score, provided no major concerns arise upon closer inspection. As a side note, the reviewer finds it interesting to explore how the parameters specific to the Transformer architecture ($\phi_P, \phi_T$ and similar quantities in Assumptions 1-4) influence the convergence properties in the final theorem. Once again, the reviewer appreciates the authors' efforts during the rebuttal period to address the concerns raised.

---

### Decision · Program_Chairs · 2024-09-25

**Decision:**

Accept (poster)

**Comment:**

The authors present a rigorous theoretical analysis of global convergence in an in-context learning setting for certain transformer-type neural network architectures. This is accomplished by a mean-field theory approach: one takes width and depth to infinity, then studies the approximation error and convergence properties of gradient flow with weight decay. The authors show that the dynamics of the discretized model approximates the Wasserstein gradient flow under some regularity assumptions, and moreover that under partial homogeneity and universal kernel assumptions, if the Wasserstein flow weakly converges to a stationary distribution, then the discrete model must also converge with arbitrarily small risk.

After robust discussion, reviewers uniformly acknowledge the nontrivial technical contributions of the authors to the mean-field study of neural networks, which so far have concentrated on ResNet architectures. Reviewers also emphasized several drawbacks of the work: the lack of practical insights into the transformer architecture afforded by the theory; certain imprecisions in presentation of the networks to which the authors’ analysis applies, which, after protracted discussion were eventually more-or-less clarified; the difficulty of checking the authors’ strong assumptions on model/data necessary to obtain global convergence results in practice (which, it must be noted, are inherited from prior work on mean-field theory of neural networks). However, on balance, reviewers argued that the paper's contributions to mean-field theory should be seen as outweighing these concerns. The AC concurs with this assessment and recommends acceptance. The authors are encouraged to incorporate all feedback from the reviews on presentation into the revision, especially the outputs of the discussion with `iBME`.